# Mitoribosome structure with cofactors and modifications reveals mechanism of ligand binding and interactions with L1 stalk

Vivek Singh [1,14], Yuzuru Itoh [1,2,14], Samuel Del'Olio [3,14], Asem Hassan [4,5,14], Andreas Naschberger [1,6], Rasmus Kock Flygaard [7], Yuko Nobe [8], Keiichi Izumikawa [9], Shintaro Aibara [1], Juni Andréll [10], Paul C. Whitford [4,5], Antoni Barrientos [3,11,12], Masato Taoka [8] & Alexey Amunts [1,13] ✉

The mitoribosome translates mitochondrial mRNAs and regulates energy conversion that is a signature of aerobic life forms. We present a 2.2 Å resolution structure of human mitoribosome together with validated mitoribosomal RNA (rRNA) modifications, including aminoacylated CP-tRNA[Val]. The structure shows how mitoribosomal proteins stabilise binding of mRNA and tRNA helping to align it in the decoding center, whereas the GDP-bound mS29 stabilizes intersubunit communication. Comparison between different states, with respect to tRNA position, allowed us to characterize a non-canonical L1 stalk, and molecular dynamics simulations revealed how it facilitates tRNA transitions in a way that does not require interactions with rRNA. We also report functionally important polyamines that are depleted when cells are subjected to an antibiotic treatment. The structural, biochemical, and computational data illuminate the principal functional components of the translation mechanism in mitochondria and provide a description of the structure and function of the human mitoribosome.

The human mitoribosome translates at least 13 respiratory chain protein-coding mRNAs from the mitochondrial genome[1]. It consists of three rRNAs and at least 82 mitoribosomal proteins, 36 of which have no corresponding homologs in bacterial and cytosolic ribosomes. The mitochondrial rRNA is reduced compared to its bacterial counterpart (2512 vs 4568 nucleotides in *Escherichia coli*), tRNA[Val] has been

incorporated as an additional constituent, and protein components are extended (~18,075 vs 7536 residues in *E. coli*)[2]. The mitoribosomal proteins might have functional roles in mitochondria-specific aspects of translation, and therefore mutations in the corresponding genes can result in cardio- and encephalomyopathies[3,4]. However, due to the limited resolution and heterogeneity of the characterized complexes,

[1]Science for Life Laboratory, Department of Biochemistry and Biophysics, Stockholm University, 17165 Solna, Sweden. [2]Department of Biological Sciences, Graduate School of Science, University of Tokyo, 113-0033 Tokyo, Japan. [3]Department of Molecular and Cellular Pharmacology, University of Miami Miller School of Medicine, Miami, FL 33136, USA. [4]Department of Physics, Northeastern University, Boston, MA 02115, USA. [5]Center for Theoretical Biological Physics, Northeastern University, Boston, MA 02115, USA. [6]King Abdullah University of Science and Technology, Thuwal 23955, Saudi Arabia. [7]Department of Molecular Biology and Genetics, Danish Research Institute of Translational Neuroscience - DANDRITE, Nordic EMBL Partnership for Molecular Medicine, Aarhus University, 8000 Aarhus C, Denmark. [8]Department of Chemistry, Graduate School of Science, Tokyo Metropolitan University, Minami-osawa 1-1, Hachioji-shi, Tokyo 192-0397, Japan. [9]Department of Molecular and Cellular Biochemistry, Meiji Pharmaceutical University, 2-522-1, Noshio, Kiyose-shi, Tokyo 204-8588, Japan. [10]Department of Medical Biochemistry and Biophysics, Karolinska Institutet, 17177 Stockholm, Sweden. [11]Department of Neurology, University of Miami Miller School of Medicine, Miami, FL 33136, USA. [12]Department of Biochemistry and Molecular Biology, University of Miami Miller School of Medicine, Miami, FL 33136, USA. [13]Present address: Westlake University, Hangzhou, China. [14]These authors contributed equally: Vivek Singh, Yuzuru Itoh, Samuel Del'Olio, Asem Hassan. ✉e-mail: alexey.amunts@gmail.com

the available models are currently incomplete, and many mitoribosomal protein key elements are described as unassigned densities[5–10]. The existing models suggest that mitoribosomal proteins might play a role in binding mRNA, regulatory elements such as the L1 stalk are probably remodeled, and the specific mitoribosomal protein mS29, a putative GTPase is associated with inter-subunit communication. Yet, the lack of experimental data leaves open the questions of a functional involvement.

The mitochondrial rRNA is modified, and dedicated post-transcriptionally activated enzymes provide means for the regulation of gene expression and mitoribosome assembly[11]. Due to the central role of the human mitoribosome in cellular energy production, defects in rRNA modification result in fatal clinical syndromes from birth[12,13]. Some of the bacterial counterpart modifying enzymes are missing from mitochondria, for example methyltransferases RlmA, RlmB, RmsE and pseudouridine synthase RluA, while other mitochondria-specific ones have evolved[14]. Therefore, rRNA modifications are different, and they have been identified by various experimental approaches[11,15–23]. Furthermore recent structural studies visualized individual rRNA modifications on the mitoribosomal large (LSU)[24] and small (SSU) subunits[25,26]. However, the entire set of modifications is yet

to be quantitatively identified and visualized in the presence of bound ligands in order to enable a detailed examination of their roles. To date, the molecular mechanisms compensating for the lack of bacteria-like modifications remain unknown, as well as the modifications and the amino-acylation state of the structural tRNA[Val].

Mitoribosomes are also linked to age-related diseases that are associated with decline in mitochondrial function and biogenesis[27–30]. Aging-related pathological changes can be inhibited by polyamine consumption[31,32], which also has anti-inflammatory effects in mouse models in vivo[33]. Polyamines have been shown to be essential for mammalian cell differentiation and proliferation through their impact on translation dynamics[34,35]. However, no polyamine biosynthetic pathways within mitochondria have been found, and their role in mitochondrial physiology remained unknown. Whether mitoribosomes might be related to these phenomena and associated with cellular senescence via polyamines needs to be explored.

In this work, to address these issues, we combine high resolution structural studies of mitoribosomal complexes with mass spectrometry based quantitative RNA analysis, biochemistry, and molecular dynamics simulations.

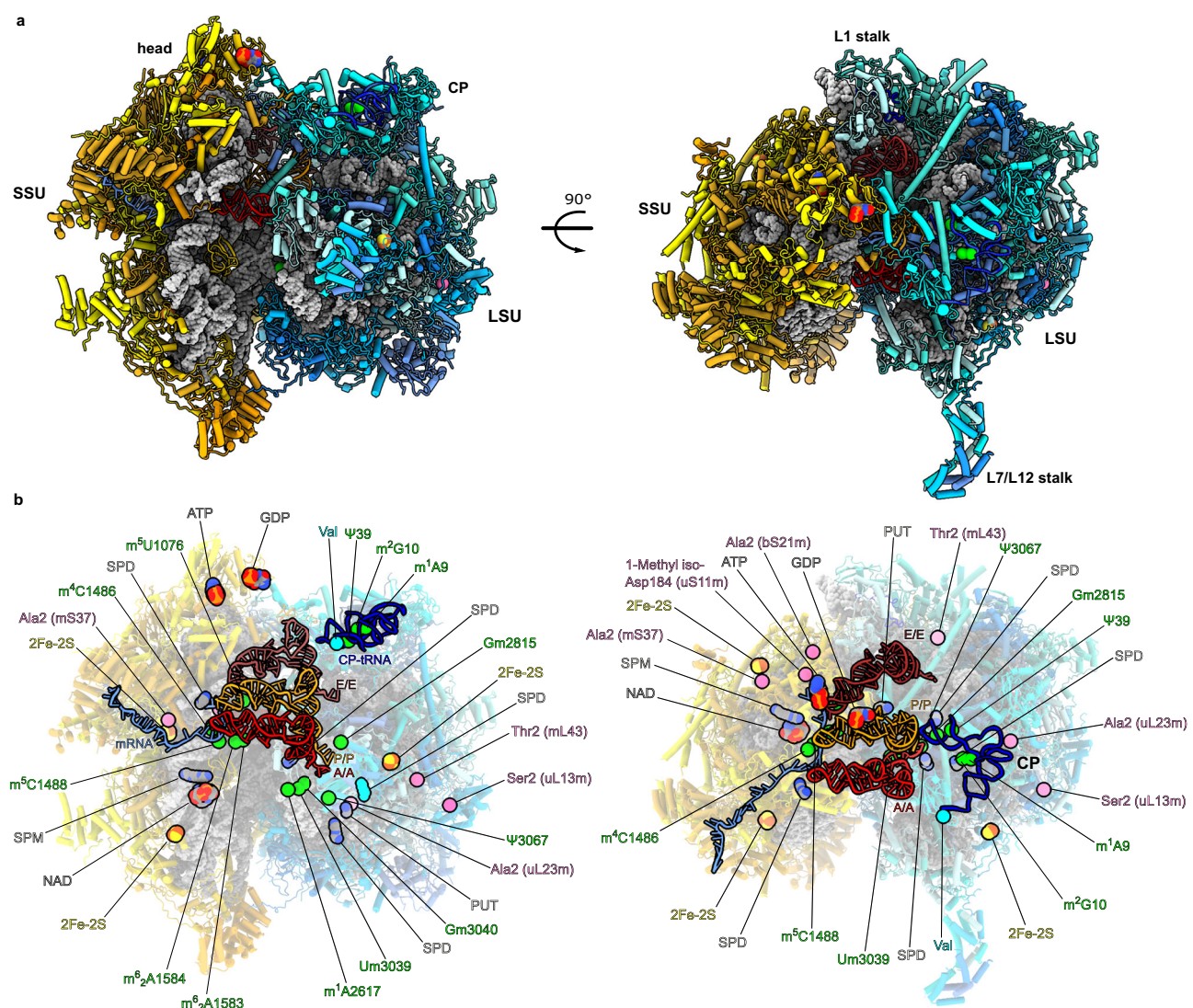

**Fig. 1 | Structure of the human mitoribosome. a** Overview of the 2.2 Å resolution model with mRNA and tRNAs. Proteins are shown as cartoons colored in blue for the LSU and yellow for the SSU. The rRNA is shown as grey surface. Newly identified features are shown as spheres. **b** The newly identified cofactors and modifications are indicated against background of translucent rRNA and proteins with tRNAs and mRNA shown in cartoon. We observe 13 rRNA modifications (green), 6 protein modifications (pink), 3 iron-sulfur clusters (red-yellow), guanosine diphosphate (GDP), adenosine triphosphate (ATP), NAD, SPM, 4 SPDs and PUT.

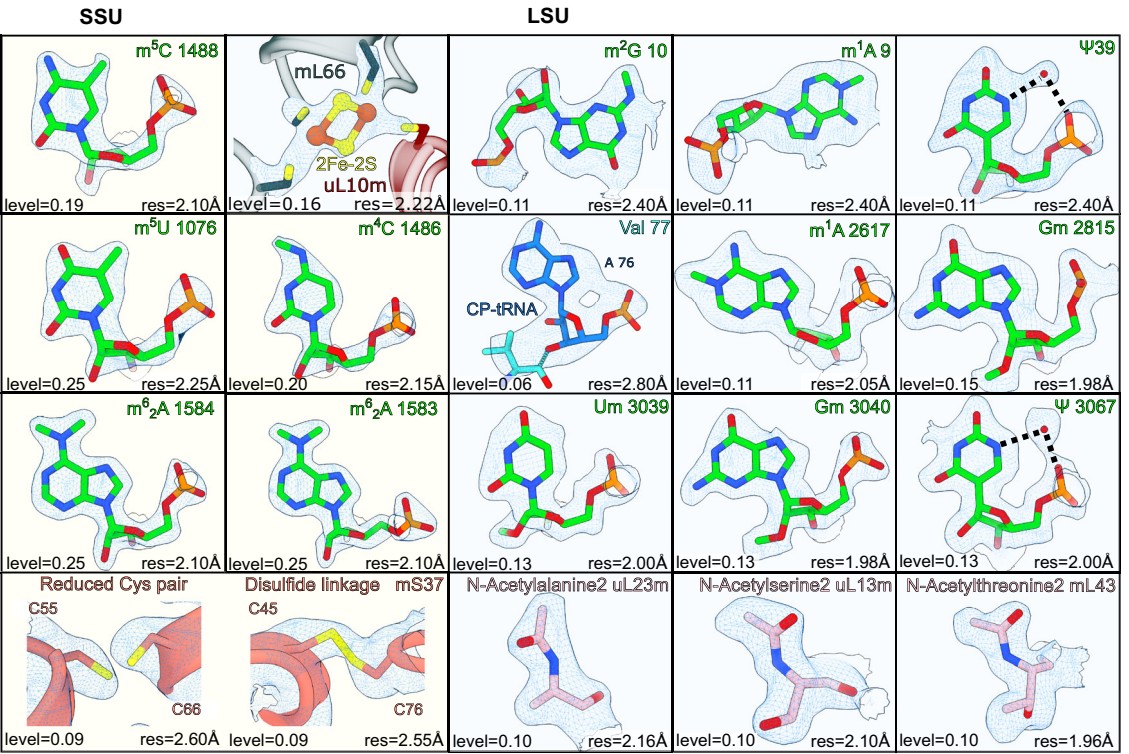

**Fig. 2 | High resolution features.** Modified rRNA residues and disulfide-linked or reduced cysteine residues of mS37 in the SSU (yellow background). Modified rRNA and protein residues, and 2Fe-2S cluster in the LSU (cyan background). Their corresponding densities in the 2.2 Å map are shown as mesh at indicated threshold (bottom left) and local resolution (bottom right) levels.

## Results

### Structure determination

To produce a 2.2 Å overall resolution map of the mitoribosome, we collected five separate cryo-EM datasets. Selected monosome particles were subjected to per-particle defocus, beam-tilt and per-particle astigmatism correction, followed by Bayesian polishing[36,37]. The data was then sorted into 86 optics groups based on acquisition areas followed by beam-tilt, magnification anisotropy and higher order aberration correction[38]. A final set of 509,691 particles yielded a 2.2 Å map. The local resolution was further improved by performing masked refinements around different regions of the monosome. In addition, we obtained a 2.6 Å resolution map in the presence of mRNA and three tRNAs in the classical state, and 3.0 Å resolution map with two tRNAs in the hybrid state (Supplementary Figs. 1 and 2). The improved resolution allowed building the most complete available model of the mitoribosome, including several key protein extensions involved in tRNA binding (Fig. 1a). The presence of mRNA on the mitoribosome further enabled modeling 32 of its residues, which we then used to trace mitochondria-specific elements involved in mRNA binding and put the rRNA modifications in a functional context (Fig. 1b).

To provide a description of the conformational changes associated with tRNA movement, we next focused efforts on modelling the L1 stalk. Thus, we merged particles containing tRNA in the E-site and through partial signal subtraction[39] and obtained a 2.9 Å resolution map of the region, enabling model building of the L1 stalk (Supplementary Fig. 1). We then used this model to perform molecular simulations of the complete ribosome with tRNAs.

### rRNA and protein modifications, aminoacylated tRNA[Val], polyamines

On the rRNA level, the quality of the map allowed the modeling of 13 rRNA modifications, of which 6 are mitochondria specific compared to *E. coli*[21]. All the rRNA modifications were validated by a tailored quantitative method of Stable Isotope-Labeled riboNucleic Acid as an internal Standard (SILNAS)[40] (Supplementary Figs. 3,4). The mitochondria specific modifications include base methylations m[5]U1076, m[5]C1488 in the SSU; m[1]A2617 in the LSU; and m[1]A9, m[2]G10 and ψ39 in the tRNA[Val] incorporated in the central protuberance (CP)[41–43] (Fig. 1b). In the map, methylations of rRNA were identified by systematically examining the density map for the presence of unmodeled densities continuous with the base or sugar moieties of rRNA residues (Fig. 2). Identity of the two high-occupancy pseudo-uridines was established based on the presence of a structured water molecule within hydrogen-bonding distance from the N5 atom.

The tRNA[Val] modifications are consistent with the previous mass-spectrometry data on isolated mitochondrial tRNAs from a general pool[44] (Fig. 3). The density further revealed that the tRNA[Val] incorporated in the CP is amino acylated (Fig. 2a). In the structure, ψ39 of tRNA[Val] contributes to the conformation of the anticodon arm, which contacts uL18m, mL38, mL40 and mL48 (Fig. 3a). In addition, two methylated residues, m[1]A and m[2]G, are observed at the positions 9 and 10, respectively in the D arm (Fig. 3a). m[1]A9 is stacked between G45 and A46 and forms a base triple with the second base pair (C11:G24) of the D stem. On the other hand, m[2]G10 is stacked between C11 and A26 and forms the first base pair (m[2]G10:C25) of the stem, which further interacts with G45 to be a base triple. These methylations facilitate the stacking and hydrogen bonds among bases, and thereby contribute to the conformation of the D arm, which contacts uL18m and mL38 (Fig. 3a–c). The terminal A76 has a base-specific interaction with Q175 of mL46 and together with Valine is stacked with F165 and F215 (Fig. 3d).

A notable feature of the cryo-EM map is the presence of continuous tubular densities associated with rRNA that we assigned as polyamines, justified by the electrostatic and hydrogen-bonding interactions at the amine groups (Supplementary Fig. 5). Thus, in addition to a previously assigned spermine (SPM)[45], we detected one triamine spermidine (SPD) in the SSU, and a diamine putrescine (PUT)

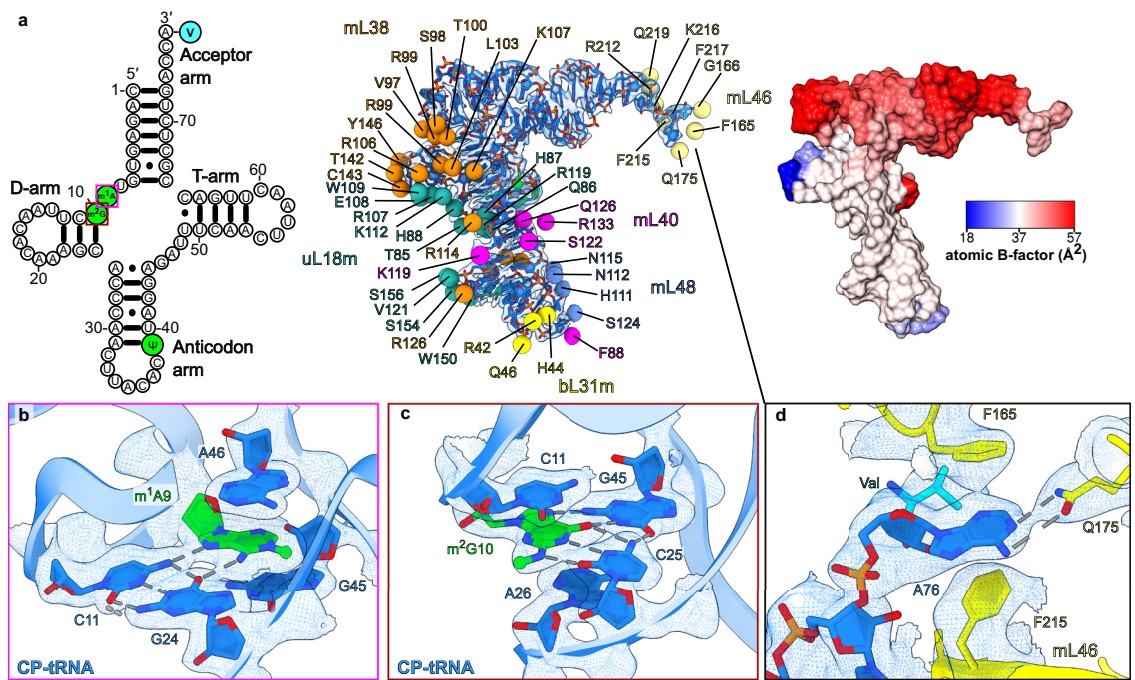

**Fig. 3 | Interactions and post-transcriptional modifications in CP-tRNA of the mitoribosome. a** CP-tRNA$^{Val}$ (left to right) secondary structure diagram with modified nucleotides (green) and valine (cyan) highlighted, protein-contact map, *B*-factor representation. **b**, **c** Interactions of the modified m$^1$A9 and m$^2$G10 nucleotides of CP-tRNA$^{Val}$. **d** Interactions of A76 and V77 of CP-tRNA$^{Val}$ with mL46. The cryo-EM density map is displayed for panels (**b**–**d**) (consensus map) as blue mesh. Panels (**b**, **c**) are related to panel (**a**) on the left by colored boxes.

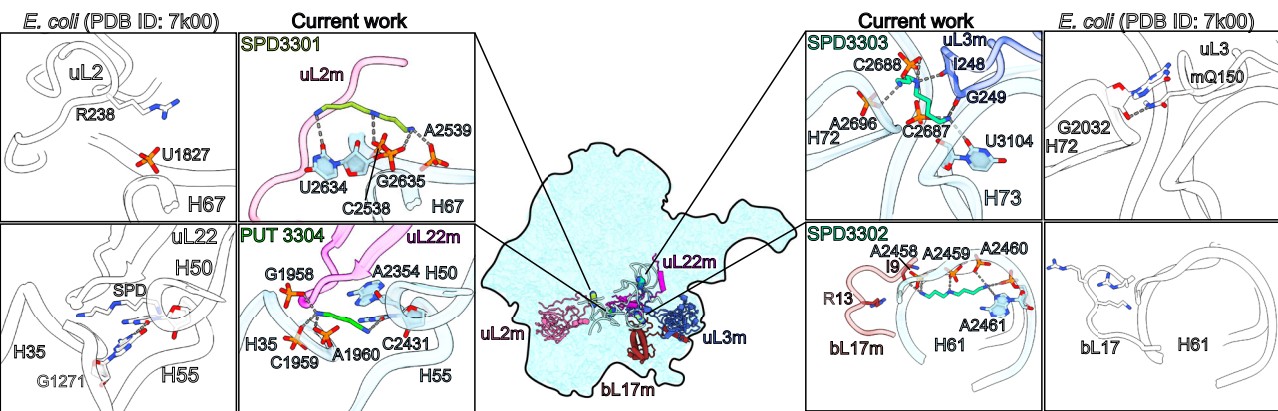

**Fig. 4 | Polyamines functionally compensate for rRNA and protein alterations.** Central inset shows positions of polyamines and interacting rRNA and proteins in the LSU. For each polyamine, its environment and interacting partners are shown in zoom-in panels with *E. coli* counterparts[52] (white) for comparison.

and three SPDs in the LSU. The identified polyamines are bound to negatively charged rRNA where mitochondria specific changes occur, providing required structural support (Fig. 4; Supplementary Fig. 5). For example, SPD3301 in the LSU fills the formed space of a shortened mitoribosomal protein uL2m, which has undergone reduction in the human mitoribosome. SPD3303 is tethered to uL3m, where a bacterial counterpart has a signature Q150 methylation, therefore replacing the post-translational modification. The PUT is found in the exit tunnel upper region of uL22m hairpin.

Finally, the model contains six protein modifications; three iron-sulfur clusters, one of which is coordinated by uL10m and mL66, and a molecule of nicotinamide adenosine dinucleotide (NAD) reported previously[26,45,46] (Fig. 1b, Supplementary Table 1, Supplementary Table 2). We also identified a disulfide bond in one of the two redox active cysteine pairs in mS37 (CHCHD1) (Fig. 2) that is oxidized in the

intermembrane space during mitochondrial import, representing a quality control mechanism[47].

## Initial binding of mRNA involves Y-N-C-Y motif coevolved with mS39

The modulation of a stable mitoribosomal complex allowed us to resolve the mRNA path and investigate mitoribosome-specific components involved in mRNA recognition (Fig. 5, Supplementary Fig. 6, Supplementary Movie 1). It engages seven helical repeats of mS39 with specific contacts within a prominent positively charged groove (Fig. 5a). Particularly, a pyrimidine ring of the mRNA nucleotide in position 25 is stacked between H489 and R524 and interacts with S490 of mS39 (Fig. 5a, b). Hydrogen bonding between the side chain of R377 of mS39 and nucleotide 27 favors cytosine over uracil because the protonated N3 of a

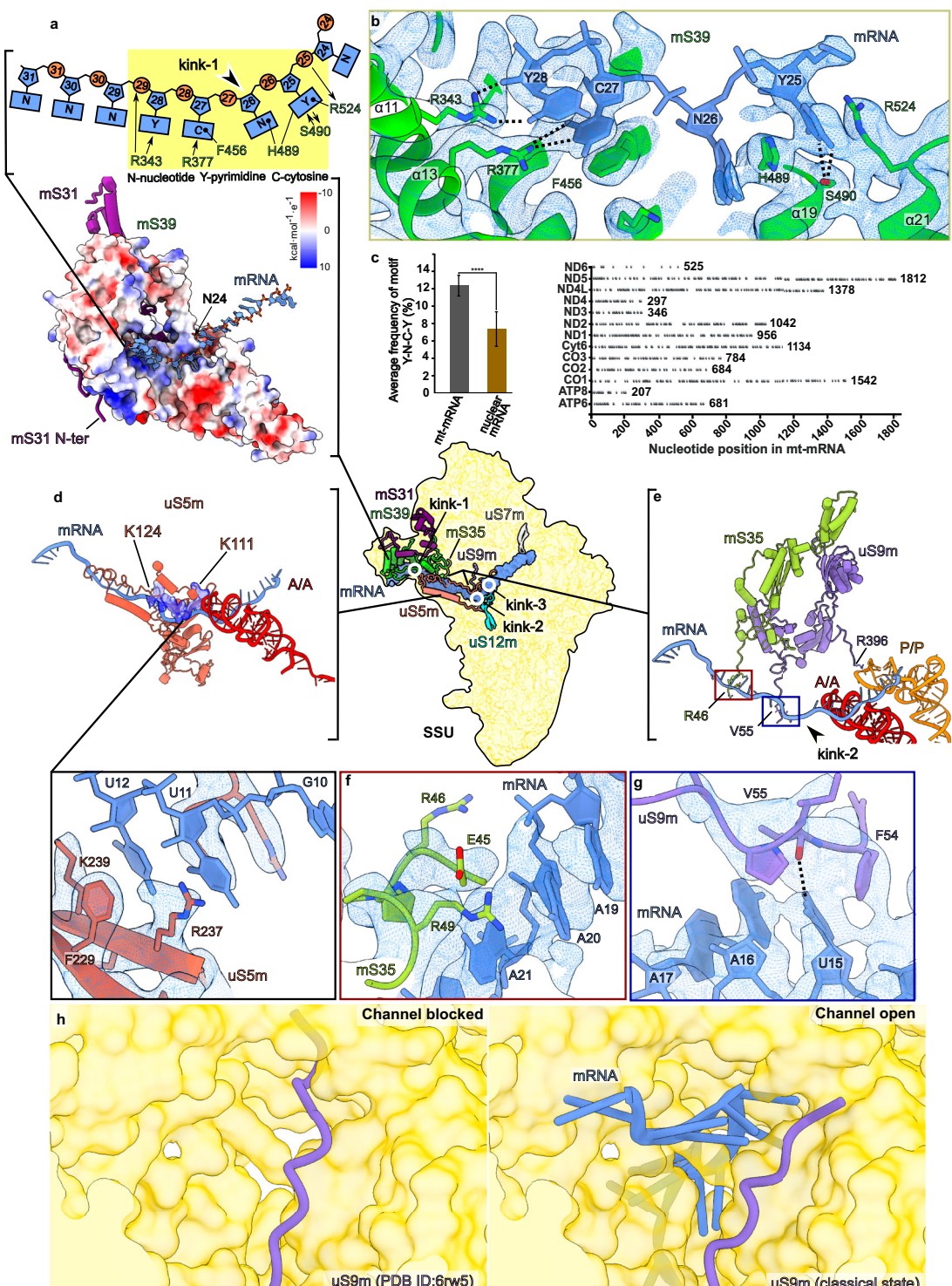

**Fig. 5 | Specific protein elements involved in mRNA binding. a** Electrostatic coulomb potential surface representation of the model shows a positively charged groove (blue) on mS39 that accommodates the mRNA. N-terminal loop of mS31 interacts with mS39. Y-N-C-Y motif and its interactions with mS39 residues are depicted as a schematic. Position of kink-2 is marked. **b** Interactions of the Y-N-C-Y motif with mS39 and density map (mesh). Protein mS31 has been removed for clarity. **c** Average percentage frequency of Y-N-C-Y motif (left) with +/− standard deviation for all mitochondrial mRNAs except ND6 ($n = 12$ biologically independent samples;; dark grey bar) and nuclear mRNAs ($n = 1387$ biologically independent samples; brown bar). *P*-value ('****' indicates $p < 0.0001$) calculated by Welch's unpaired two-tailed t-test. Positions of the Y-N-C-Y motif in the mt-mRNAs are

shown as discs (right) to depict its distribution along the mt-mRNA sequences. The total number of residues for each mt-mRNA is indicated. **d** Overview of uS5m showing a region of contact with mRNA. The polybasic stretch 111–124 along the mRNA is colored by electrostatic potential. Interacting residues between uS5m and mRNA are shown with the local density map. **e** Interactions of mS35 and uS9m with mRNA at the channel entry and the P-site. **f**, R46 and R48 of mS35 potentially interact with the mRNA backbone, and the density suggests flexibility of the side-chains. **g** uS9m forms stacking and hydrogen bonding interactions with mRNA position 15 via F54 and V55, respectively. **h** uS9m N-terminus (purple cartoon) adopts alternative conformations that result in mRNA channel blocked or open states regulating mRNA access to the SSU (gold surface).

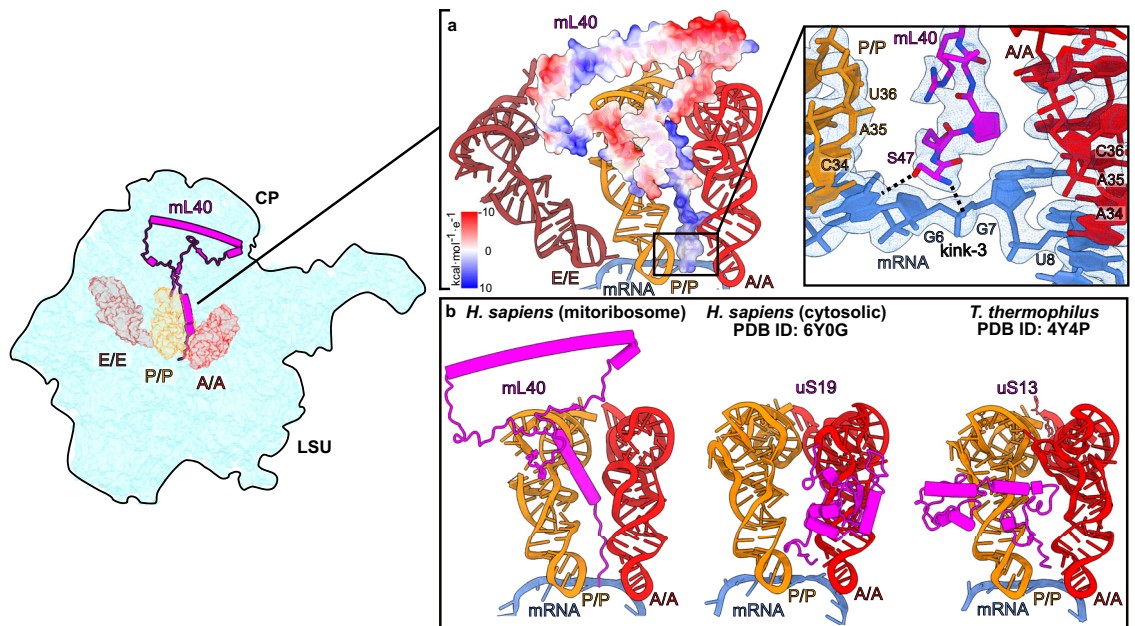

**Fig. 6 | Analysis of mRNA binding in tRNA sites. a** Interactions of mL40 (shown with electrostatic potential of the model) with mRNA between A/A- and P/P-tRNAs. Hydrogen bonds G6 base and G7 phosphate are shown as dashed lines. **b** Comparison of mL40-tRNA interaction (left) with those of uS19 (middle) and uS13 (right) in human cytosolic[54] and *T. thermophilus* ribosome[51], respectively.

uracil would partially clash with the protonated Nη of R377. R343 of mS39 forms a hydrogen bond with the pyrimidine ring of nucleotide 28. Thus, the structure reveals a binding motif: 25 pyrimidine (Y), 27 cytosine (C), 28 pyrimidine (Y) (Fig. 5a, b). Bioinformatic analysis showed that the Y-N-C-Y motif is enriched in mitochondria and distributed over the entire length of each of the 12 open reading frames in the guanine-rich heavy DNA strand, suggesting a co-evolution of the mitochondrial DNA and the mitoribosome (Fig. 5c, Supplementary Fig. 7). The only mammalian protein-coding gene with a reduced frequency of the Y-N-C-Y motif is the *MT-ND6* that is on the light strand and codes for an 18 kDa subunit of complex I. The expression of the ND6 mRNA is unique, as it requires an auxiliary factor FASTK to generate the mature form, suggesting a dedicated regulatory mechanism[48].

The most 5′ contact between mRNA and mS39 is ~50 Å from the channel entrance, and by following the mRNA density toward the decoding center, we identified six specific mitoribosomal proteins that are important for mRNA recruitment, channel opening, and precise alignment of the message for translation in the decoding center. In our structure, the Y-N-C-Y motif forms a ~100° turn (kink-1) in the mRNA midway through mS39 (Fig. 5a and Supplementary Movie 1). After the kink, the N-terminal mitochondria-specific extension of uS5m supports the path of mRNA over a distance of ~60 Å, involving a unique N-terminal polybasic stretch of 14 residues 111-124 (KKGRGKRTKKKKRK) together with R237 and K239 (Fig. 5d, Supplementary Movie 1, Supplementary Fig. 6). From the side of the SSU head, mS35 provides potential stabilizing electrostatic interactions to mRNA with N-terminal residues K46 and K49 (Fig. 5e, f). This mitochondria-specific extension of uS5m structurally replaces bacterial bS4 and uS3-CTD that interact with mRNA (Supplementary Fig. 6) and confer RNA helicase activity on the bacterial ribosome[49]. Just before the channel entry, the binding of mRNA induces a conformational change of the N-terminal extension of uS9m that contacts the mRNA nucleotide at position 15 via the backbone of V55 (Fig. 5e, g; Supplementary Movie 1), which in the absence of mRNA blocks the entrance to the decoding center[50]. Thus, uS9m exists in a closed conformation in the absence of mRNA, and upon mRNA threading, a conformational change in uS9m allows the passage (Fig. 5h).

## mL40 N-terminal tail stabilizes the A- and P-site tRNAs and decoding interactions

We then examined the role of mitoribosomal proteins in the mRNA alignment. The structure shows that to enter the SSU channel, the mRNA must be kinked again (kink-2), and in the A-site the nucleotide at position 9 is stabilized by uS12m (Supplementary Fig. 8). Here, a conserved K72 side chain forms a salt bridge with the phosphate of nucleotide 9 (last nucleotide in the A site codon). At the same time, *cis*-proline P73 (Supplementary Fig. 8) coordinates two potassium ions, one of which directly coordinates 2′O of nucleotide 9 (Supplementary Fig. 8, Supplementary Movie 1) and stabilizes the decoding center. The densities identified as two potassium ions have been observed previously, but modeled as $Mg^{2+}$ (PDB 4V51)[51] or waters (PDB 7K00)[51,52]. However, they have been experimentally identified as $K^+$ by long wavelength X-ray diffraction analysis (PDB 6QNR)[53], which agrees with our assignment (Supplementary Fig. 8) based on distances from coordinating residues (Supplementary Fig. 9)

Next, between the A- and P-site, the map reveals a third kink in the mRNA (kink-3), which is co-localized with a 35 Å long density extending from the CP in between the two tRNAs (Fig. 6a, Supplementary Fig. 10a) that could not be previously interpreted. We identified it as a positively charged N-terminus of mL40 with the terminal backbone amino group of S47 forming a salt bridge with the backbone phosphate of the mRNA at position 7 in the A-site (Fig. 6a, Supplementary Fig. 10b), and the interaction is further stabilized via a magnesium ion coordinated by the phosphate of G1485. In addition, the hydroxyl group of S47 interacts with the mRNA nucleotide base at position 6 in the P-site (Fig. 6a, Supplementary Movie 1), although the contacts are likely to be flexible. Moreover, the base orientation allows any nucleotide to interact with S47 (Supplementary Fig. 10c). The concept is analogous to the cytosolic ribosome that engages uS19 for tRNA stabilization and mRNA interactions in the decoding center[54] (Fig. 6b). The presence of four lysine residues 52-55 (Fig. 6a, Supplementary Fig. 10b), might suggest that mL40 contributes to balancing the charge between the A- and P-tRNAs. Together with R51, which forms salt bridges with A1560 and C1561 in h44, the positively charged patch compensates for the negative RNA charges accumulated in that region (Fig. 6a, Supplementary Fig. 10b, Supplementary Movie 1). Further in

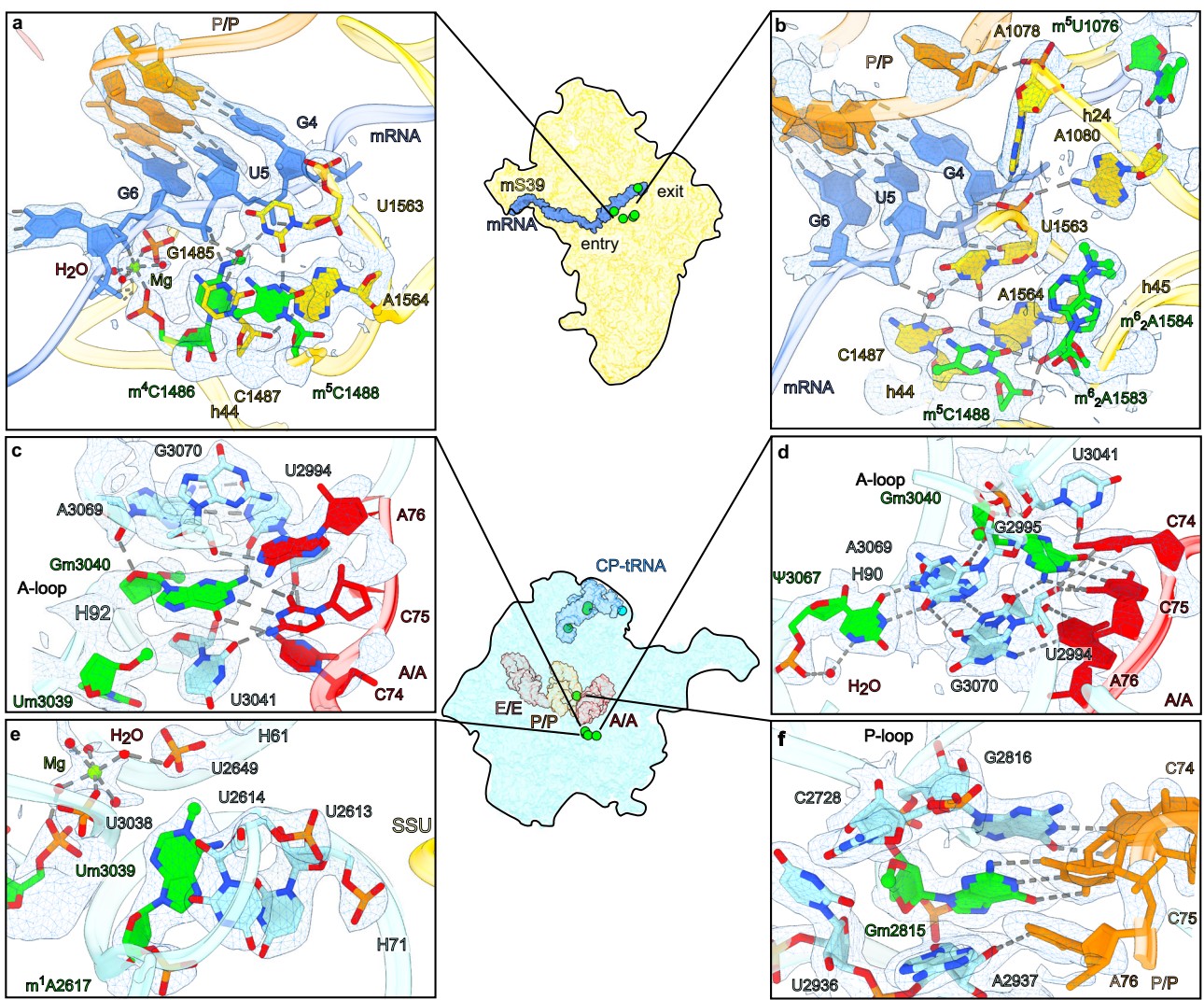

**Fig. 7 | Post-transcriptional modifications in functional centers of the mitoribosome. a** Base-methylated m⁴C1486 and m⁵C1488 and the interactions at the P-site mRNA codon. **b** The dimethylated m⁶₂A1583, m⁶₂A1584 and base-methylated m⁵1076 facilitate interactions with mRNA codon and P-tRNA. **c** 2′-O-methyl modified Um3039 and Gm3040 lead to interactions with A-tRNA. **d** Gm3040 and Ψ3067 contribute to the interface with the A-tRNA. **e** m¹A2617 contributes to a stabilization of inter-subunit bridge and the A-loop. **f** Gm2815 base-pairs with P-tRNA. Cryo-EM density map (A/A P/P E/E state) is displayed as a blue mesh.

the P-site, we observe a well-resolved density showing the conserved C-terminal R396 of uS9m interacting with the anticodon loop of the P-site tRNA (Supplementary Fig. 10d). Finally, in the E-site, uS7m residues G164-165 further contribute to mRNA stabilization (Supplementary Fig. 10e). Thus, mitoribosomal proteins play a role in mRNA binding.

## mRNA alignment in the decoding center and rRNA modifications

In the decoding center, the universally conserved G902 is in *anti* conformation, while A1557 and A1558 are flipped out from h44 to interact with the A-site codon:anti-codon helix in the canonical manner[55,56] (Supplementary Fig. 11). A well-resolved density, combined with the mass-spectrometry SILNAS[40,57] approach further allowed us to validate the complete set of the rRNA modifications, including six mitochondria specific, and propose a role in stabilizing the decoding center (Fig. 5, Supplementary Figs. 4 and 5). Close to the decoding center, the residue G899 was recently shown to interact with streptomycin in human mtSSU[45]. Its methylated to m⁷G527 in *E.coli* (Supplementary Fig. 11) and this modification partly contributes to streptomycin sensitivity[58]. In the P-site, two base-

methylated nucleotides m⁴C1486 and m⁵C1488 flank the base triple C1487:A1564:U1563 by stabilizing the stacking interactions (Fig. 5a, b). The orientation of N⁴-methylated m⁴C1486 towards mRNA is conserved with respect to *E. coli*, where the corresponding residue is also 2′-O methylated (m⁴Cm1402) (Supplementary Fig. 11). m⁴C1486 together with C1487 interacts with the phosphate group of the last nucleotide in the P-site mRNA codon via hydrogen bonds (Fig. 5a, b). Further, phosphate group of m⁴C1486 coordinates a magnesium ion, which stabilizes a conserved kink of the mRNA between A- and P-sites (Fig. 7a). In contrast to the *E. coli* P-site, U1563 does not have an N³-methylation (m³U1498 in *E. coli*) (Supplementary Fig. 11). This lack of N³-methylation allows U1563 together with C1487 to form a water mediated interaction with the P-site codon (Fig. 7a, b). Further, the conserved dimethylated residues, m⁶₂A1583 and m⁶₂A1584 orient the nucleotide U1563 such that the ribose of U1563 forms a hydrogen bond to a phosphate group of the second nucleotide in the P-site mRNA codon, while the phosphate of U1563 forms a hydrogen bond with the ribose on the first codon nucleotide (Fig. 7b). The two dimethylated nucleotides facilitate A1080 positioning for the 2′-OH to hydrogen bond with N3 of m⁵U1076. This together with methylation of m⁵U1076 contributes to the stability of

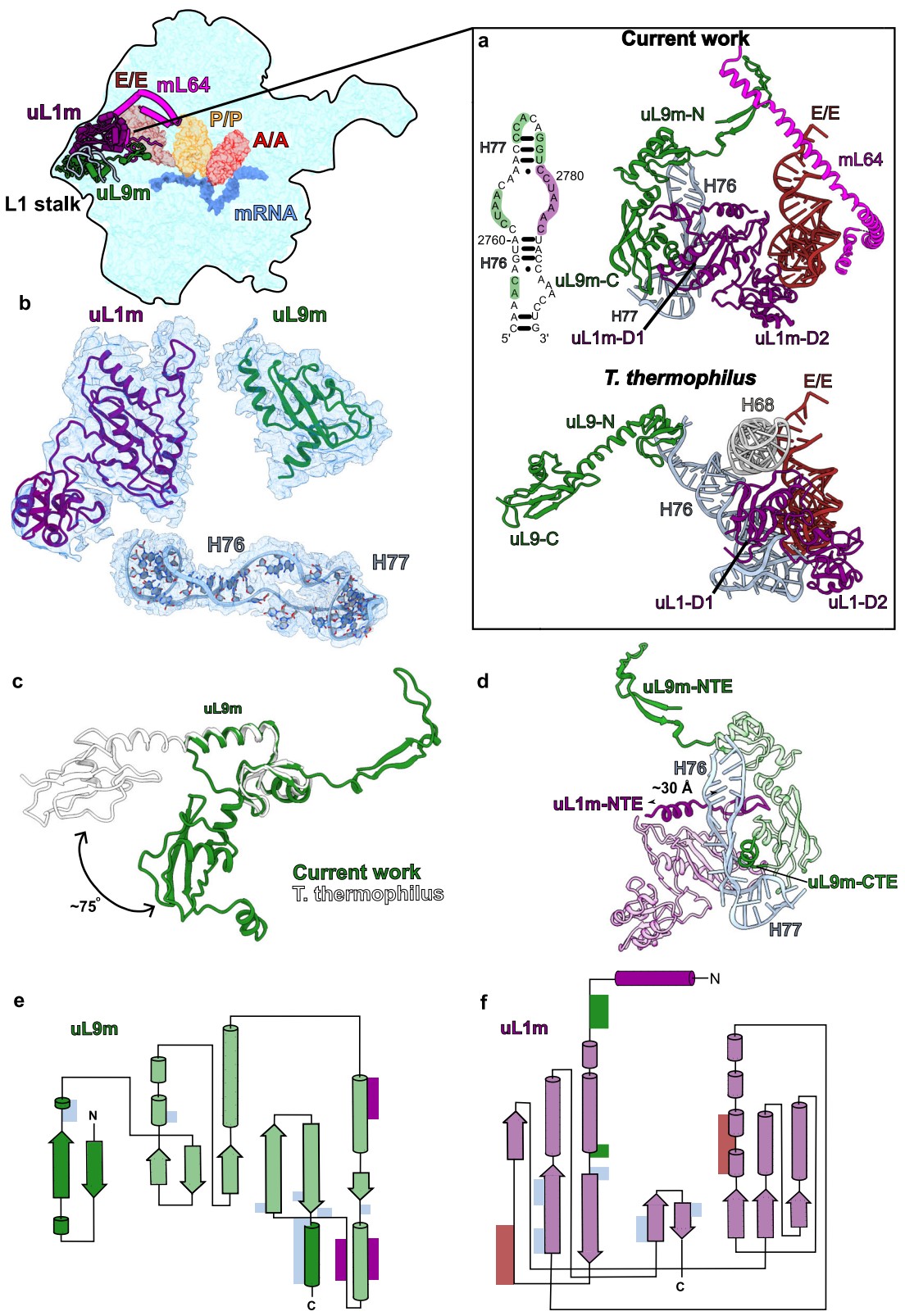

**Fig. 8 | Structural features of the L1 stalk and interactions with tRNA. a** The L1 stalk components uL9m (green), uL1m (purple) rRNA helix H76-77 (residues 2757-2790, light blue), as well as mL64 (magenta) interact with E/E-tRNA. The structure and interactions are different from *T. thermophilus* (PDB ID: 4V51) [https://doi.org/10.2210/pdb4V51/pdb][51] **b** Components of the L1 stalk modeled into their respective densities (shown as mesh). **c** Comparison of uL9m

conformation with that of *T. thermophilus* uL9 (white PDB ID 4V51). **d** Model of uL1m stalk with uL9m in green and uL1m in purple, highlighting mitochondria-specific protein elements. **e** 2D diagram of uL9m showing interactions with E/E-tRNA (brown bars), rRNA (light-blue bars), uL1m (dark-blue bars). Conserved regions are shown in light and mitochondria-specific in dark colors. **f** 2D diagram of uL1m color-coded as in panel d.

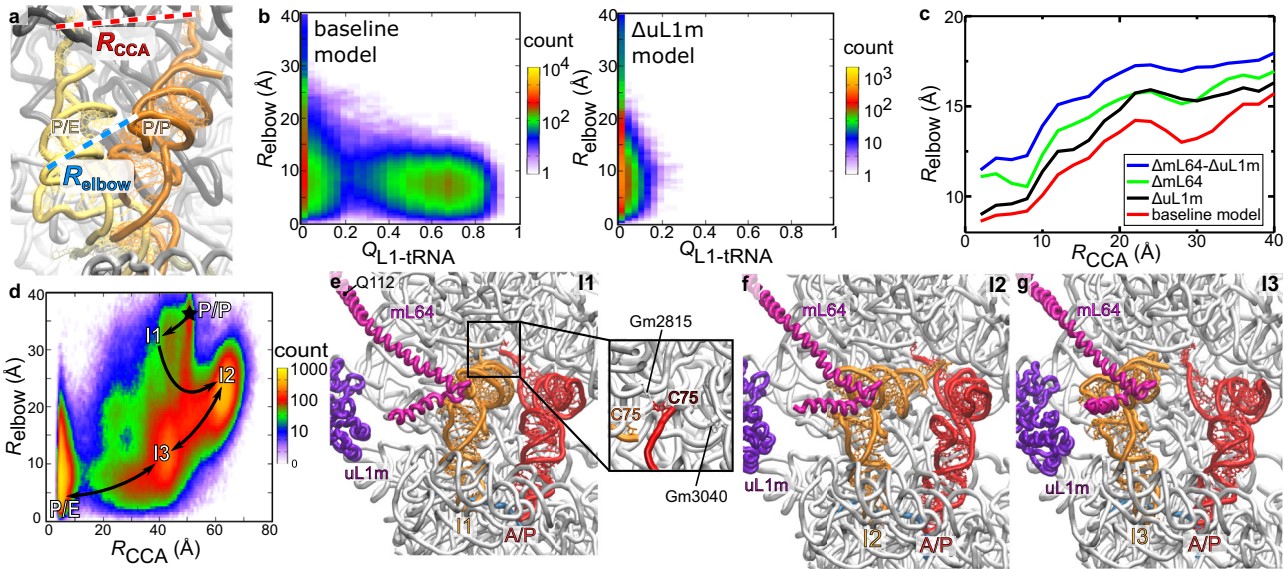

**Fig. 9 | Simulations of P/E formation reveal the influence of mitochondria-specific proteins. a** Distances $R_{elbow}$ and $R_{CCA}$ are used to describe the position of the tRNA during P/E formation. **b** Probability distribution as a function of P-site tRNA elbow position ($R_{elbow}$) and the fraction of uL1m-tRNA contacts ($Q_{L1-tRNA}$) relative to those formed in the E/E state, calculated from simulations that used the baseline model (left). When uL1m-P-tRNA contacts are attenuated (ΔuL1m model; right), $Q_{L1-tRNA}$ remains low, consistent with a lack of uL1m binding. **c** In simulations of P/E formation, perturbations to uL1m and/or mL64 (models ΔuL1m, ΔmL64, and ΔmL64-ΔuL1m models) lead to a wider range of tRNA elbow positions and longer timescales for P/E formation. The narrower range of pathways accessed with the complete/baseline model leads to a lower average value of $R_{elbow}$ as a function of the tRNA CCA position $R_{CCA}$, showing how the corridor favors a process where the elbow motion precedes CCA movement. **d** Probability distribution (327 simulated events) as a function of elbow and 3′-CCA position reveals distinct intermediates during P/E formation. Representative structural snapshots of I1-I3 are shown in panels (**e**–**g**). **e** Intermediate I1 is associated with a small displacement of the tRNA elbow and CCA tail toward the E site. **f** In order to reach I2, the tRNA elbow is displaced by an additional ~10 Å, where it it confined by contacts with mL64. **g** mL64 then guides the tRNA elbow towards the E site, allowing it to adopt I3. After reaching I3, a final ~40 Å displacement of the CCA tail results in adoption of the P/E conformation.

h24 loop that interacts with the P-tRNA anticodon stem at position 38 via A1078 (Fig. 7a, b).

In the A-loop of H92, there are two 2′-O-methyl modified nucleotides Um3039 and Gm3040. The 2′-O-methyl of Um3039 sterically positions the bases of Gm3040 and U3041 to form interactions with the nucleotides C75 and C74 of the A-tRNA, respectively (Fig. 7c, d). The 2′-O-methyl of Gm3040 present in the cytosolic but not bacterial ribosome and interacts with the base of A3069 to stabilize the conformation of Gm3040. Furthermore, the bacterial counterpart rRNA that interacts with A-tRNA (C75) is G:C, whereas in our structure it's A:U (3069:2994) that is less stable, and thus stabilized by the mitochondria-specific methylation of Gm3040 and base-pairing interactions with ψ3067 (Fig. 7d). In the LSU, mitochondria-specific m[1]A2617 positions U2614 into a wedged interaction (Fig. 7e). Surprisingly, there is only weak sugar puckering for m[1]A2617 (Fig. 2a). It bears a partial positive charge at N1 and interacts with U2649 of H61. The interactions mediated by m[1]A2617 potentially contribute to the structuring of H71 loop that participates in an intersubunit bridge shares an interface with the A-loop (Fig. 7e). The universally conserved Gm2815 in the P-loop forms Watson-Crick base pairing with C75 of P/P-tRNA, similar to the interactions of Gm3040 of the A-loop (Fig. 7f).

## Structure of the mitoribosomal L1 stalk reveals its unique protein features

To investigate whether mitoribosomal proteins might also have influence on tRNA rearrangement in the E-site, we aimed to perform molecular simulations of the mitochondrial L1 stalk. Therefore, we modeled the complete structure of the L1 stalk, which relies on mitochondria specific protein elements. In the structure, the E-tRNA is stabilized via mitochondria-specific protein elements of the C-terminal domain of uL9m and 33-residue N-terminal extension of uL1m (Fig. 8a, b). The orientation of the uL9m C-terminal domain adopts a

previously unseen folded conformation involving a ~75° rotation towards the E-tRNA (Fig. 8c). This unexpected conformational difference is facilitated by the C-terminal helix in the extension that reaches and contacts the non-base-pairing part and the distal short hairpin of the rRNA H76 (Fig. 8d). The resulting projection of uL9m towards the ribosomal core reduces the distance between the central helix and the C-terminus of uL1m from ~70 to only ~30 Å. This distance is spanned by the uL1m mitochondria-specific extension that forms direct contacts with the central helix of uL9m (Fig. 8d–f). Therefore, our model of the L1 stalk establishes a tertiary mitoribosomal complex uL9m:uL1m:rRNA-H76, where uL9m acquires a new functional role. From the opposite side, the mitochondria-specific mL64 approaches the T-arm through its long C-terminal helix extending to the tRNA binding sites.

## Recognition of tRNA by the L1 stalk and intersubunit communication

We next used an all-atom structure-based model[59,60] to simulate hundreds of transitions of the tRNA from the P site to P/E state (Fig. 9a, b; Supplementary Movie 2). To isolate the influence of uL1m and mL64, we compared the dynamics with four variants of the model (Fig. 9c; see Methods): 1) baseline model; 2) truncated mL64 (ΔmL64) with C-terminal region starting with Q112 being removed to prevent steric interactions between mL64 and EE-tRNA without disturbing the rest of the protein.; 3) modified uL1m (ΔuL1m) where stabilizing interactions between uL1m and P-site-tRNA are scaled down compared to the baseline model (values available in methods); 4) both ΔmL64 and ΔuL1m modifications are included (see methods section for details of the models). Using these models, we simulated 1066 spontaneous (non-targeted) transitions between the classical (A/A-P/P) and hybrid (A/P-P/E) states, which include a ~40 Å displacement of the P-tRNA and spontaneous rotation of the SSU (Supplementary Movie 2). A

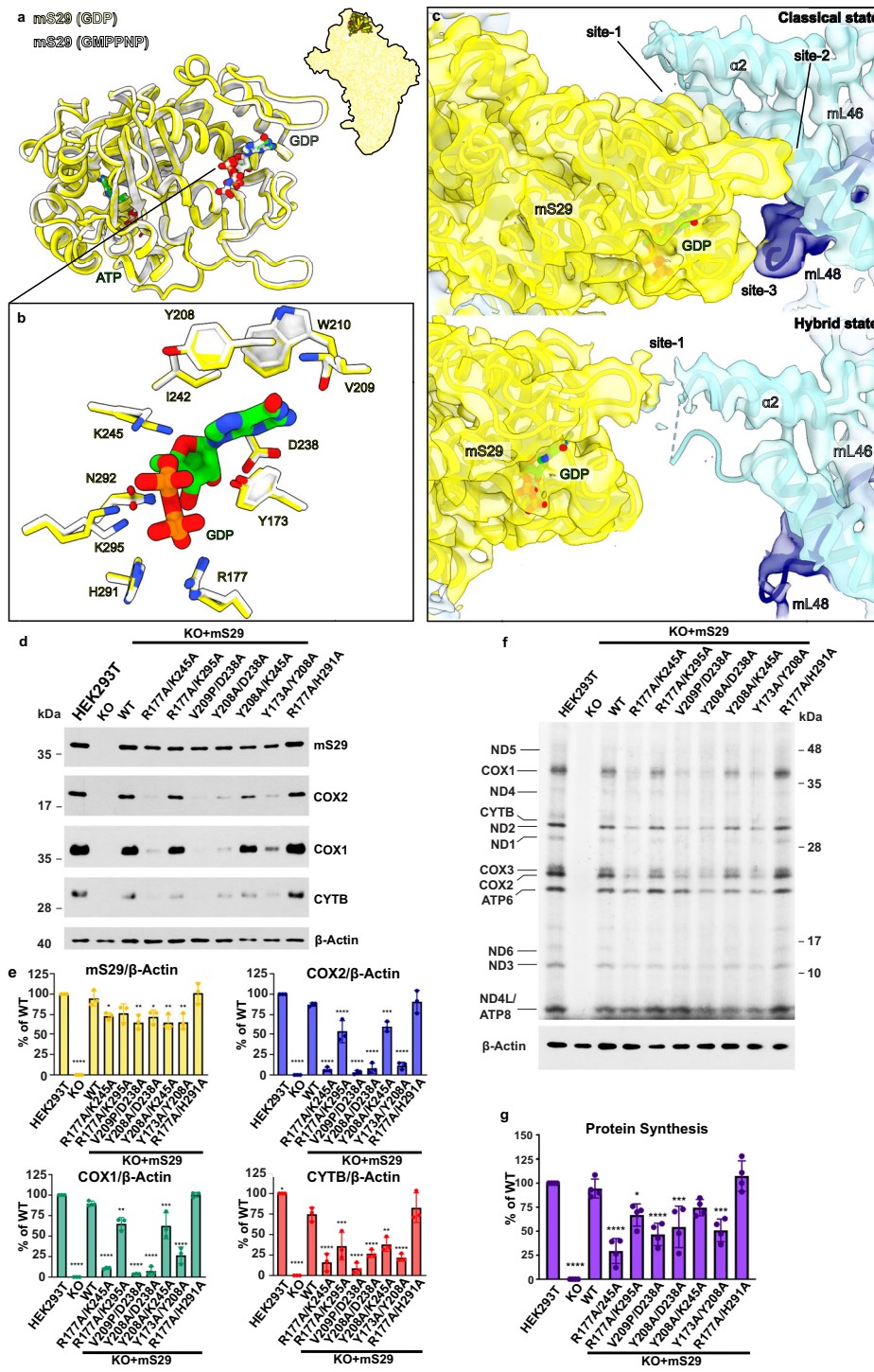

**Fig. 10 | Binding of GDP to mS29 and role in mitoribosome function.**
**a** Superposition of GDP-bound (yellow) and GMPPNP-bound (white) states.
**b** Close-up view illustrating side chains that coordinate nucleotide binding.
**c**, Intersubunit contacts in the classical and hybrid states mediated by mS29 (cryo-EM density map shown as colored surface). **d** Immunoblot analysis comparing steady-state levels of proteins using whole cell lysates from HEK293T (WT), *mS29*-KO (KO), and *mS29*-KO cells stably expressing either WT mS29 or double mutants targeting the GDP-mS29 β-hairpin binding site under the control of an attenuated CMV6 promoter (Δ5pCMV6). COX1, COX2, and CTYB were used as markers of mitoribosome function, and β-Actin was used as a loading control. **e** Images in panel (**d**) were digitalized, and densitometry was conducted using the histogram function in Adobe Photoshop. The bar graphs represent the mean ± SD from three independent repetitions (*n* = 3). Statistical differences were estimated using one-

way ANOVA with Dunnett's multiple comparisons test (two-tailed) in GraphPad Prism. \**p* < 0.05; \*\**p* < 0.01; \*\*\**p* < 0.001; \*\*\*\**p* < 0.0001. **f** Metabolic labeling of mitochondrially translated peptides in whole cells from the indicated cell lines using $^{35}$S-methionine for 15 min in the presence of emetine to inhibit cytosolic protein synthesis. Immunoblotting for β-ACTIN was used as a loading control. Newly synthesized peptides are identified on the left. **g** Images in panel (**d**) were digitalized, and densitometry was conducted using the histogram function in Adobe Photoshop. The bar graphs represent the mean ± SD from four independent repetitions (*n* = 4). Statistical differences were estimated using one-way ANOVA with Dunnett's multiple comparisons test (two-tailed) in GraphPad Prism. \**p* < 0.05; \*\**p* < 0.01; \*\*\**p* < 0.001; \*\*\*\**p* < 0.0001. The source data file contains all original uncropped and unprocessed immunoblots as well as densitometry values used to perform statistics and generate quantification graphs in panels (d–g).

mechanistic insight from these calculations is that mL64 imposes strict limits on the pathways that are accessible to tRNA, which thereby favors a specific sequence of conformational substeps during P/E formation (Fig. 9d–g). As a result, mL64 ensures that the tRNA is properly positioned and may reliably engage uL1m as the tRNA approaches the E site. The structure also shows that due to the different type of tRNA binding, no base pairs are formed between the E/E-tRNA and mRNA, unlike those found in bacteria[61] (Supplementary Fig. 10e). Instead, the protein uS7m undergoes a conformational change in which residues G164-165 directly contact the mRNA and stabilize its binding (Supplementary Fig. 10e, Supplementary Video 1). Thus, upon each step of translation, a tRNA is recognized in the E-site and escorted from the mitoribosome without involving rRNA, which is strikingly different from bacterial and cytosolic counterparts.

### Mitoribosome function is independent of mS29 GTPase activity, but GDP stabilizes intersubunit communication

For each translation step, the movement of mRNA:tRNA is also accompanied by the SSU head swiveling, but it has remained an open question whether intrinsic GTPase activity of mS29 (also known as death-associated protein 3, DAP3) is coupled to the process[62,63]. We now resolve it by analyzing the structure in two different conformations combined with structure-based mutagenesis and biochemical analysis (Fig. 10, Supplementary Fig. 12a–c). Particularly, we performed site-directed mutagenesis targeting the previously reported nucleotide-binding site[2,62] and found that the mutant K134A, which based on the structure would affect the nucleotide binding, appears to be dispensable for mitoribosomal function (Supplementary Fig. 12a, b). Recently, another putative binding site has been proposed, formed with residues 173, 177, 208, 209, 210, 238, 242, 245, 291, 292, 295[45,46] (Fig. 10a, b, Supplementary Fig. 12d), and our map at the local resolution of 2.65 Å suggests the presence of GDP in that site. To address the functional relevance of the GDP, we compared it with the SSU structure in the presence of a nonhydrolyzable analog GMPPNP[45] (Fig. 10a, b). It showed a very similar global architecture of mS29, implying that no net conformational change is involved upon GTP hydrolysis (Fig. 10a, b). Furthermore, there is no loss of interactions of mS29 with its binding partners in the SSU head (Fig. 10b). Together, the data suggest that the GTPase activity is not coupled to the translation cycle and the related SSU head swiveling.

We next used structural and biochemical analysis to identify the role of the GDP. We compared the interactions of mS29, which is the signature protein of the SSU head with the CP between our classical (2.6 Å resolution) and hybrid (3.0 Å resolution) structures. In the classical state, they are extensive and manifest through three contact sites. Site-1 and -2 are formed between GDP-associated region and the respective mL46 helices α2 and α3 extending from the CP. Contact site-3 is formed with mL48 β-turn protruding into the intersubunit space. Upon subunit ratcheting, in the hybrid state mS29 shifts resulting in disruption of contact site-3. Further, the mS29 β-hairpin 208-216 (stabilized by GDP) that formed contact site-2 in the classical state switches in the hybrid state and remains the only inter-subunit bridge in the dimeter of 50 Å (Fig. 10c, Supplementary Fig. 12e). To assess the role of GDP binding in translation, we generated cell lines stably expressing mS29 mutants to disrupt the GDP binding site (Fig. 10d–g, Supplementary Fig. 12a–c).

Expression of the double mutants R177A/K245A or R177A/K295A, which target residues interacting with the phosphate groups of GDP, led to reduced levels of mitochondrion-encoded proteins and cellular respiration (Fig. 10d–g, Supplementary Fig. 12c), suggesting a stabilizing role for GDP in this pocket. The R177A/K245A mutation shows only a tendency toward a more stringent effect on mitochondrial protein synthesis than the R177A/K295A mutation in our short 15 min pulses. However, the effect is significant when looking at the steady-state levels of mtDNA-encoded proteins. (Fig. 10d, e). Interestingly, the

double mutant R177A/H291A displayed a rate of mitochondrial protein synthesis comparable to WT (Fig. 10d–g). Although K245 and K295 stabilize the same beta phosphate on GDP, it is possible that H291 can compensate for the loss of K295, explaining the decreased stability of the R177A/K245A mutant compared to the R177A/K295A mutant. Change of nearby hydrophobic residues Y173A/Y208A yields a decrease in translation similar to the R177A/K245A implying their role in GDP stabilization (Fig. 10d–g). Lastly, double mutant pairs V209P/D238A and Y208A/D238A, which directly disrupt GDP base binding to the mS29 β-hairpin, showed drastic decreases in the synthesis and stability of mitochondrion-encoded proteins (Fig. 10d–g). Thus, the biochemical assays based on mutagenesis show that GDP binding to mS29 is required for efficient mitochondrial translation. Our structure-function correlation analyses suggest that GDP acts as a stabilizing agent for mS29 β-hairpin 208-216 to maintain the intersubunit communication as tRNA moves to the hybrid state.

## Discussion

This study illustrates the modifications and cofactors of the human mitoribosome (Figs. 1–4, Supplementary Figs. 3–5) and explains how mRNA and tRNAs are stabilized (Figs. 5–9, Supplementary movie 1). While the basic mechanism of translation and the rRNA in the decoding center are conserved, our structures attribute additional roles to mitoribosomal proteins in mRNA and tRNA binding.

For mRNA, previous work showed that the delivery of some mitochondrial transcripts for translation is facilitated by the factor LRPPRC[64,65]. We now demonstrate that once an mRNA is received by the mitoribosome, it involves six mitochondria-specific proteins that sequentially help to direct the mRNA to the decoding center, where it's aligned with a tRNA (Figs. 5,6, Supplementary Fig. 6,8,10). Most of these proteins compensate for the reduced rRNA through the mechanisms of structural patching[66] and protein ratcheting[67] that drive neutral evolution of protein complexity. The first protein to receive an mRNA is mS39, and we detected a co-evolution of mS39 with an mRNA motif Y-N-C-Y that is enriched in all 12 transcripts encoded in the heavy strand of the mitochondrial DNA. The only mRNA that is not enriched with the motif is ND6 (Fig. 5, Supplementary Fig. 7). Interestingly, *ND6* possesses a long 3′UTR and is the only transcript in human mitochondria lacking a polyA tail. This gene is also the only one in mitochondria of vertebrates that was found to be translocated to the control region through gene duplication[68]. This occurs in Antarctic fish species, and while the selectively advantageous duplicate is functionally neutral on the protein level[69], we found that there is a change in the Y-N-C-Y frequency in the mRNA sequence, which is increased by 2-fold (from 14 to 30 occurrences in 525 residues). It suggests a possible adaptive correlation between the mRNA stability on the mitoribosome and mt-DNA reorganization.

For tRNA, the structure demonstrates two cases with an involvement of mitoribosomal proteins. First, between the A- and P-site tRNAs, the N-terminus of mL40 extends from the CP to the phosphate backbone of mRNA. Second, the recognition of a tRNA in the E-site is achieved by a protein-made L1 stalk region and mL64. Both architectures are unique to mitochondria. Our molecular dynamics simulations further illustrate how the process of E-tRNA movement fundamentally relies on specific protein extensions and insertions of uL1m and uL9m without any involvement of rRNA (Figs. 8, 9, Supplementary movie 2). These adaptations of mitoribosomal proteins are particularly important in the context of different functional states, as previous structural studies of elongation[8], termination[10], and rescue[6] complexes reported unidentified densities associated with translational factors, which we now assign specific proteins. Since our data resolves those densities as mammalian protein extensions (Fig. 5, Supplementary Fig. 6), the structural information can now be used for reinterpretation of the functional complexes, thus completing previous works.

In addition to proteins, newly revealed structural details include polyamines as stabilizing intramolecular agents in human mitochondria. The visualization of polyamines in the human mitoribosome and the reduced occupancy in the presence of antibiotics further supports their stimulating effect on mitochondrial translation, which is essential for cell proliferation. In mitoribosomes from cell culture treated with antibiotics[45,70], the occupancy of the polyamines is reduced[45], and they are generally replaced with $Mg^{2+}$ or $K^+$, accompanied by some residue rearrangement (Supplementary Fig. 5). Those changes are associated with attenuation of translation, and thus our structure rationalizes previous report that the positive effect of polyamines could not be functionally replaced by the addition of $Mg^{2+}$ in a cell-free system[71]. Together, it links mitochondrial translation to reports assigning key regulatory effects to polyamines in mitochondria. This includes anti-inflammatory effects and inhibition of aging-associated pathologies upon consumption of polyamine in mice[31,32,72], the ability to boost mitochondrial respiratory capacity in a *Drosophila* aging model[32], restorative effects on oxidative functions of aged mitochondria[73,74], and antitumor immunity in aged mice[75]. Polyamines were also proposed to regulate mitochondrial metabolism via Src kinase signaling to preserve bioenergetics in the spinal cord[76]. Considering our results on the involvement of iron-sulphur clusters as protein-protein mediating agents[45], as well as in the mitoribosomal assembly[77], the new structural data provides a wealth of additional information on cofactors that can be further assessed in disease models and with respect to tissue and cell specificity, as well as ageing processes. The detection of disulfide bond in mS37 that was proposed to occur during the mitochondrial protein import pathway[47] further links translation with metabolic state and NADH/NAD$^+$ ratio in the intermembrane space and the matrix due to the special role of mS37 that links assembly to the initiation of translation[25].

On the rRNA level, through tailored quantitative mass-spectrometry, we describe a fully modified mitoribosome, including in the decoding center. While the critical modifications are conserved with bacteria, an unexpected feature of the structure is that the rRNA component CP-tRNA$^{Val}$ is modified, as well as aminoacylated. This suggests that this tRNA is recognized in the matrix by the elongation factor mtEF-Tu prior the assembly onto the CP. The involvement mtEF-Tu in the mitoribosome assembly has been previously reported, and direct interaction with the assembly factor GTPBP5 was visualized by cryo-EM of an LSU intermediate[78]. Thus, the regulation of the assembly of the mammalian mitoribosome relies on the availability of mtEF-Tu in active developmental stages of the organelle, which might be a potential early checkpoint for translation. A possible model for these observations is that accumulation of deacylated tRNAs might trigger starvation responses, and since tRNA$^{Val}$ is likely to be abundant at the onset of mitoribosomal assembly, its aminoacylation is essential, which requires the presence of mtEF-Tu. The feature is likely to be mammalian-specific, as representative from other lineages, such as fungi[46], ciliates[79], and algae[80,81] do not possess the CP-tRNA component in their mitoribosomes.

Finally, the data addresses a long-standing question of whether a putative GTPase and death-associated protein mS29 might provide the human mitoribosome with an intrinsic activity, while undergoing structural rearrangements. This protein is of a particular clinical interest, because several cancer types are associated with its aberrant expression[82]. We show that mS29-GDP does not change its conformation during translation but is a stable structural element that contributes to inter-subunit communication during transitions between classical and hybrid states (Fig. 10, Supplementary Fig. 12).

Overall, these data explain how the newly identified components modulate the functional and structurally distinct mitoribosome. With the continuous progress in improvement of resolution of cryo-EM reconstructions, and availability of newly developed methods for base editing of protein-coding genes in mammalian mitochondrial genome[83,84], the field of mitochondrial translation now steps into a phase, where it becomes possible to analyze fine regulatory mechanisms specific transcripts, and study how it's affected by modifications, cofactors, and mutants. Therefore, the structural approach has a potential to lead towards a more detailed description of the mitoribosome-associated pathologies, including mitochondrial disease and ageing, and the current study will provide a reference model of the human mitoribosome for future investigation.

## Methods

### Experimental model and culturing

HEK293S-derived cells (T501) were grown in Freestyle 293 Expression Medium containing 5% tetracycline-free fetal bovine serum (FBS) in vented shaking flasks at 37 °C, 5% CO$_2$ and 120 rpm (550 x g). Culture was scaled up sequentially, by inoculating at 1.5 ×10$^6$ cells/mL and subsequently splitting at a cell density of 3.0 × 10$^6$ cells/mL. Finally, a final volume of 2 L of cell culture at a cell density of 4.5 × 10$^6$ cells/mL was used for mitochondria isolation, as previously described[85].

*mS29*-KO cells were derived from HEK293T (CRL-3216) and cultured in complete DMEM supplemented with 20% FBS at 37 °C and 5% CO$_2$. HEK293T (CRL-3216) were cultured in high-glucose, sodium pyruvate, L-glutamine Dulbecco's modified Eagle medium (DMEM, Life Technologies) supplemented with 10% FBS, 100 µg/mL uridine, 3 mM sodium formate, and 1x MEM non-essential amino acids complete DMEM medium). Analysis of mycoplasma contamination was routinely performed.

For the labeling of cells with deuterated Uridine required for SILNAS analysis, TK6 cells deficient in uridine monophosphate synthase (*UMPS-/-*) were cultured at 37 °C in RPMI-1640 medium containing 10% horse serum, 2 mM L-glutamine, penicillin (50 U/ml), streptomycin (50 µg/ml) and 0.1 mM 5,6-D2-uridine.

### Generation of mS29 edited cell lines and plasmid transfections

To generate a stable human *mS29-KO* cell line in HEK293T cells (ATCC: CRL-3216), we used a pool of CRISPR/Cas9-mediated knockout cells generated by the genome editing company, Synthego (Synthego Corporation, Redwood City, CA). HEK293T cells were first tested to be negative for mycoplasma. Three guide RNAs targeting exon 5 of the *mS29* gene (DAP3-202 transcript ID ENST00000368336.10) were selected for high specificity and activity to create premature stop codons through frameshift mutations in the coding region via insertions and/or deletions (Indels). Targeting exon 5 ensured that all *mS29* transcript variants would be affected. Based on additional off-target analysis, the three specific guide RNAs selected (g1, g2, and g3) were: g1 5'-GUGAAGCUUGCCUGAUGGUA-3' [AGG]-PAM, g2 5'-UAUAGCUG-GAUAAGCAAAAC-3' [TGG]-PAM, and g3 5'-AUGCUUUCUCCCCU-CAAAUA-3" [AGG]-PAM. The pool of knockout cells was then generated by electroporation of ribonucleoproteins (RNPs) containing the Cas9 protein and synthetic chemically modified sgRNA (Synthego) into the cells using Synthego's optimized protocol. The editing efficiency was assessed upon recovery, 48 h post-electroporation. Genomic DNA is extracted from a portion of the cells, PCR-amplified, and sequenced using Sanger sequencing. The resulting chromatograms are processed using Synthego Inference of CRISPR Edits software (ice.synthego.com). The pooled *mS29-KO* was then plated into 96-well plates to screen for single clones in-house. Single clone candidates were screened by immunoblotting to determine the steady-state levels of mS29 and the mtDNA-encoded COX2 protein as a surrogate of mitochondrial protein synthesis capacity. Clones that had undetectable mS29 and attenuated COX2 levels were further analyzed by genotyping. For genotyping, the edited region of mS29 was amplified using oligonucleotides mS29 exon 5 seq (-164) Forward: 5'-GGATA-GATTTTCAAACTCAGTACCA-3' and mS29 exon 5 seq (+166) Reverse: 5'-TCCTGACTTCAGGCGATACG-3', and subsequently cloned into the TOPO-TA vector (Thermo Fisher) for sequencing.

To establish a stable cell line reconstituted with mS29, we obtained the WT *mS29* gene as a Myc/DDK-tagged ORF from Origene (Cat# RC223182). Using restriction sites *Asi*SI and *Pme*I, WT mS29-Myc/DDK was cloned into a mammalian vector with hygromycin as the selection marker. The vector used was pCMV6, in which mS29 gene expression was placed under the control of an attenuated version of the human cytomegalovirus (CMV) enhancer/promoter (Δ5) in which a deletion in the promoter sequence eliminates most transcription factor binding sites[86]. Furthermore, to ensure functionality of the recombinant mS29, the Myc/DDK tag was removed via the Q5 site-directed mutagenesis kit (New England Biolabs) using the oligonucleotides mS29 ΔMyc-DDK Forward: 5'-TAAACGGCCGGCCGCGGT-3' and mS29 Δmyc-DDK Reverse: 5'-GAGGTAGGCACAGTGCCGCTC-3'. Two µg of the construct containing *mS29* was transfected to the HEK293T *mS29*-KO cell line using 5 µL of EndoFectin™ Max (GeneCopoeia) pre-incubated in Opti-MEM (ThermoFisher). 72 h post-transfection, the medium was supplemented with 200 µg/mL hygromycin B for three weeks. We use the same protocol to establish stable cell lines reconstituted with mutant variants of mS29.

Mutant variants of the *mS29*-containing construct were generated using the Q5 site-directed mutagenesis kit (New England Biolabs). Mutagenesis primers were designed using the NEBaseChanger tool. Oligonucleotides used for mutagenesis include: mS29 K134A Forward: 5'-GGGAACAGGAGCAACCCTAAGTC-3'; mS29 K134A Reverse: 5'-TTCTCTCCATACAGAAGATATC-3'; mS29 Y173A Forward: 5'-GCAGTCCAGCGCCAACAAACAGC-3'; mS29 Y173A Reverse: 5'-AGAAGATCCCGACAATTTTTC-3'; mS29 R177A Forward: 5'-CAACAAACAGGCCTTTGATCAACCTTTAGAG-3'; mS29 R177A Reverse: 5'-TAGCTGGACTGCAGAAGA-3'; mS29 K245A Forward: 5'-AATTGTGCTGGCAGAGCTAAAGAGGC-3'; mS29 K245A Reverse: 5'-CCAACTGCATCTGTGGCG-3'; mS29 H291A Forward: 5'-AGCACTTGTTGCCAACTTGAGGAAAATGATG-3'; mS29 H291A Reverse: 5'-AATTCCTCGGGGGCAATC-3'; mS29 K295A Forward: 5'-CAACTTGAGGGCAATGATGAAAAATGATTG-3'; mS29 K295A Reverse: 5'-TGAACAAGTGCTAATTCC-3'; mS29 Y208A Forward: 5'-TCAAGAGAAGGCTGTCTGGAATAAGAG -3'; mS29 Y208A Reverse: 5'-ACTTTTATCTGGTTCAGG -3'; mS29 V209P Forward: 5'-AGAGAAGTATCCCTGGAATAAGAGAG -3'; mS29 V209P Reverse: 5'-TGAACTTTTATCTGGTTCAG -3'; mS29 D238A Forward: 5'-GAACGCCACAGCTGCAGTTGGAA -3'; mS29 D238A Reverse: 5'-CTCACCCGTGTTATGCCC -3. All mutant sequences were validated by Sanger sequencing using oligonucleotides Δ5pCMV6 Forward: 5'-CCTCTTCGCTATTACGCCAG-3', mS29 cDNA (308) Forward: 5'-AACCAGCCCTAGAACTTCTGC-3', and mS29 cDNA (829) Forward: 5'-GAAGATAAAAGCCCGATTGC-3'.

## Mitoribosome purification

Mitoribosome purification was carried out based on the previously developed protocol for rapid isolation[85]. HEK293S-derived cells were harvested from the 2 L culture when the cell density was $4.2 \times 10^6$ cells/mL by centrifugation at 1000 g for 7 min, 4 °C. The pellet was washed and resuspended in 200 mL Phosphate Buffered Saline (PBS). The washed cells were pelleted at 1000 g for 10 min at 4 °C. The resulting pellet was resuspended in 120 mL of MIB buffer (50 mM HEPES-KOH, pH 7.5, 10 mM KCl, 1.5 mM MgCl₂, 1 mM EDTA, 1 mM EGTA, 1 mM dithiothreitol, complete EDTA-free protease inhibitor cocktail (Roche) and allowed to swell in the buffer for 15 min in the cold room by gentle stirring. About 45 mL of SM4 buffer (840 mM mannitol, 280 mM sucrose, 50 mM HEPES-KOH, pH 7.5, 10 mM KCl, 1.5 mM MgCl₂, 1 mM EDTA, 1 mM EGTA, 1 mM DTT, 1X cOmplete EDTA-free protease inhibitor cocktail (Roche) was added to the cells in being stirred in MIB buffer and poured into a nitrogen cavitation device kept on ice. The cells were subjected to a pressure of 500 psi for 20 min before releasing the nitrogen from the chamber and collecting the lysate. The lysate was clarified by centrifugation at 800 x g and 4 °C, for 15 min, to separate the cell debris and nuclei. The supernatant was passed

through a cheesecloth into a beaker kept on ice. The pellet was resuspended in half the previous volume of MIBSM buffer (3 volumes MIB buffer + 1 volume SM4 buffer) and homogenized with a Teflon/glass Dounce homogenizer. After clarification as described before, the resulting lysate was pooled with the previous batch of the lysate and subjected to centrifugation at 1000 x g, 4 °C for 15 min to ensure complete removal of cell debris. The clarified and filtered supernatant was centrifuged at 10,000 x g and 4 °C for 15 min to pellet crude mitochondria. Crude mitochondria were resuspended in 10 mL MIBSM buffer and treated with 200 units of Rnase-free Dnase (Sigma-Aldrich) for 20 min in the cold room to remove contaminating genomic DNA. Crude mitochondria were again recovered by centrifugation at 10,000 g, 4 °C for 15 min and gently resuspended in 2 mL SEM buffer (250 mM sucrose, 20 mM HEPES-KOH, pH 7.5, 1 mM EDTA). Resuspended mitochondria were subjected to a sucrose density step-gradient (1.5 mL of 60% sucrose; 4 mL of 32% sucrose; 1.5 mL of 23% sucrose and 1.5 mL of 15% sucrose in 20 mM HEPES-KOH, pH 7.5, 1 mM EDTA) centrifugation in a Beckmann Coulter SW40 rotor at 28,000 rpm (139,000 x g) for 60 min. Mitochondria seen as a brown band at the interface of 32% and 60% sucrose layers were collected and snap-frozen using liquid nitrogen and transferred to −80 °C.

Frozen mitochondria were transferred on ice and allowed to thaw slowly. Lysis buffer (25 mM HEPES-KOH, pH 7.5, 50 mM KCl, 10 mM Mg(OAc)₂, 2% polyethylene glycol octylphenyl ether, 2 mM DTT, 1 mg/mL EDTA-free protease inhibitors (Sigma-Aldrich) was added to mitochondria and the tube was inverted several times to ensure mixing. A small Teflon/glass Dounce homogenizer was used to homogenize mitochondria for efficient lysis. After incubation on ice for 5–10 min, the lysate was clarified by centrifugation at 30,000 x g for 20 min, 4 °C. The clarified lysate was carefully collected. Centrifugation was repeated to ensure complete clarification. A volume of 1 mL of the mitochondrial lysate was applied on top of 0.4 mL of 1 M sucrose (v/v ratio of 2.5:1) in thick-walled TLS55 tubes. Centrifugation was carried out at 231,500 x g for 45 min in a TLA120.2 rotor at 4 °C. The pellets thus obtained were washed and sequentially resuspended in a total volume of 100 µl resuspension buffer (20 mM HEPES-KOH, pH 7.5, 50 mM KCl, 10 mM Mg(OAc)₂, 1% Triton X-100, 2 mM DTT). The sample was clarified twice by centrifugation at 18,000 g for 10 min at 4 °C. The sample was applied on to a linear 15–30% sucrose (20 mM HEPES-KOH, pH 7.5, 50 mM KCl, 10 mM Mg(OAc)₂, 0.05% n-dodecyl-β-D-maltopyranoside, 2 mM DTT) gradient and centrifuged in a TLS55 rotor at 213,600 x g for 120 min at 4 °C. The gradient was fractionated into 50 µL volume aliquots. The absorption for each aliquot at 260 nm was measured and fractions corresponding to the monosome peak were collected. The pooled fractions were subjected to buffer exchange with the resuspension buffer to dilute away sucrose.

## rRNA mass spectrometry

**Purification of mitochondrial rRNAs.** Mitochondrial rRNAs of the HEK293 cells were extracted from the purified ribosome using Isogen (Nippon Gene) and stored at −80 °C until use. TK6 cell culture (~1.0 × 10⁹ cells) were homogenized by Dounce homogenizer with 250 mM sucrose, 1 mM EDTA and 10 mM Tris−HCl (pH 8.0). The lysate was centrifuged twice at 1500 x g for 3 min at 4 °C, and the resulting supernatant was further centrifuged at 5000 x g for 10 min to obtain mitochondrial precipitate. From the precipitate, total mitochondrial RNA (~100 µg) was extracted using TRIzol Reagent (Thermo Fisher Scientific).

Each mitochondrial rRNA was purified by using reversed-phase LC through a PLRP-S 4000 Å column (4.6 × 150 mm, 10 µm, Agilent Technologies). After applying the mixture of rRNAs from the purified ribosome or total mitochondrial RNA to the column, the rRNAs were eluted with a 60-minute linear gradient of 11.5−13.5% (v/v) acetonitrile in 100 mM TEAA (pH 7.0) and 0.1 mM diammonium phosphate at a flow rate of 200 µl/min at 60 °C while monitoring the eluate at A260[87].

**LC-MS, MS/MS and MS/MS/MS analysis and database search of RNA fragments.** RNA was digested as described below. Nucleolytic RNA fragments were analyzed with a direct nanoflow LC-MS system as described[88]. The LC eluate was sprayed online at −1.3 kV with the aid of a spray-assisting device[88] to a Q Exactive Plus mass spectrometer (Thermo Fisher Scientific) in negative ion mode. Other settings for MS, MS/MS and MS/MS/MS were as described in ref. 89.

Ariadne[57] was used for database searches and assignment of MS/MS RNA spectra. The composite of human cytosolic and mitochondrial rRNA and tRNA sequences was used as a database. The following default search parameters for Ariadne were used: maximum number of missed cleavages, 1; variable modification parameters, two methylations per RNA fragment for any residue; RNA mass tolerance, ±5 ppm, and MS/MS tolerance, ±20 ppm. For assignment of Ψ residues using 5,6-D2-uridine labeled RNAs, the mass table and the variable modification parameters were altered from default values to "5,6D_CU" and "Ψ", respectively, because both C and U were labeled with the medium and the pseudouridylation reaction results in the exchange of the position 5 deuterium of 5,6-D2-uridine to the proton of solvent, providing a −1 Da mass shift[90].

**Internal standard RNAs and SILNAS-based quantitation of the stoichiometry of post-transcriptional modification.** To construct the plasmids for in vitro transcription of internal standard RNAs, DNAs encoding human 12 S and 16 S rRNAs were amplified by PCR from the mitochondrial DNA from TK6 cells. The PCR primers used are as follows: HindIII-12S-Forward TATAAAGCTTAATAGGTTTGGTCCTAGCC TTTCTATTAGC, 12S-*Bam*HI-reverse ATATGGATCCGTTCGTCCAAGTG CACTTTCCAGTACACTT, *Hind*III-16S-forward TATAAAGCTTGCTAAA CCTAGCCCCAAACCCACTCCACCT and 16S-*Xho*I-reverse ATATCTCGA GAAACCCTGTTCTTGGGTGGGTGTGGGTATA). The amplified DNAs were inserted into the multiple cloning sites of plasmid pCDNA3.1 (+) (Thermo Fisher Scientific). To synthesize RNA, 2 µg of template DNA was incubated and transcribed using Megascript T7 kit (Thermo Fisher Scientific). When RNA was synthesized, guanosine-$^{13}C_{10}$ 5′-triphosphate, cytidine-$^{13}C_9$ 5′-triphosphate, or uridine-$^{13}C_9$ 5′-triphosphate solution was used instead of the respective 5′-triphosphate reagent that contained carbons with a natural isotope distribution. The RNA was precipitated in ethanol, solubilized in nuclease-free water, and purified further by reversed-phase LC as described above.

SILNAS-based quantitation was performed as described in ref. 91. In brief, RNA ( ~100 fmol) from natural sources or cells grown in guanosine with natural isotope distribution was mixed with an equal amount of synthetic RNA transcribed in vitro with $^{13}$C-labeled guanosine and digested with RNAse T1 (2 ng/µl). For the RNA transcribed with $^{13}C_9$-labeled cytidine and uridine, Rnase A (0.5 ng/µl) was used as the digestion enzyme. Digestion was carried out in 100 mM TEAA (pH 7.0) at 37 °C. The 1:1 RNA mixing was performed based on the measurement of the absorbance at 260 nm and ensured later by a correction factor obtained experimentally. After obtaining the LC-MS spectrum of the digested RNA mixture, the stoichiometry of RNA modification at each site was estimated by Ariadne program designed for SILNAS[91]. The results were confirmed by manual inspection of the original MS spectrum to examine whether the estimates are based on 'uncontaminated' MS signals.

**Other procedures for RNA analysis.** The masses of RNA fragments and a-, c-, w-, and y-series ions were calculated with Ariadne (http://ariadne.riken.jp/). Sequence-specific Rnase H cleavage of rRNAs was performed as described in ref. 92, using O-methylated RNA/DNA hybrid oligonucleotide complementary to the 12 S and 16 S rRNAs, GmUmUmCmGmUmCm(CAAG)UmGmCmAmCmUmUmUmCmCm AmGmUmAmCmAmCm and (AmAmCmCmCmUmGm(TTCT)UmGm GmGmUmGmGmGmUm, respectively (where Nm refers to 2′-O-

methyl ribonucleotide and deoxyribonucleotides are indicated in parentheses).

## SDS-PAGE and immunoblotting analyses
Whole-cell protein extracts were obtained by solubilization in RIPA buffer (25 mM Tris-HCl pH 7.6, 150 mM NaCl, 1% NP-40, 1% sodium deoxycholate, and 0.1% SDS) with 1 mM PMSF and 1x mammalian protease inhibitor. Extracts were cleared by 5 min of centrifugation at 10,000 g at 4 °C. Protein concentrations were determined with the Folin phenol reagent. 40–60 µg of whole-cell protein extracts were separated by SDS-PAGE in the Laemmli buffer system. Proteins were then transferred to nitrocellulose membranes, subsequently blocked with 5% skim milk, and then incubated with antibodies against the indicated proteins followed by a second reaction with anti-mouse or anti-rabbit IgG conjugated to horseradish peroxidase. Signals were detected by chemiluminescence and exposure to X-ray films. For quantification, the signals in digitalized images were quantified by densitometry using the histogram function of Adobe Photoshop. Values were normalized to the signal of β-Actin and plotted as the percentage of WT using the Prism 8 software.

## Pulse labeling of mitochondrial translation products
Mitochondrial protein synthesis was determined by pulse-labeling of 60–70% confluent human wild-type (WT) HEK293T cells, *MRPS29*-KO HEK293T cells, and knockout cells stably reconstituted with mutant *MRPS29* variants. Cultures were grown in a medium without methionine and in the presence of 100 µg/mL emetine to inhibit cytoplasmic translation as described in refs. 93,94. Cells were labeled for 15 min at 37 °C with 100 µCi of $^{35}$S-methionine (PerkinElmer Life Sciences, Boston, MA). After incubation, the medium containing $^{35}$S-methionine was removed, the cells were washed once with PBS, and then the cells were incubated for 5 minutes in complete DMEM to allow recovery. After the short incubation, cells were washed once more with PBS, collected by trypsinization, and whole-cell protein extracts were obtained by solubilization with RIPA buffer. 40–50 µg of each sample was separated by SDS-PAGE on a 17.5% polyacrylamide gel, transferred to a nitrocellulose membrane, and exposed to X-ray film. Membranes were then probed by immunoblotting to assess the steady-state levels of β-ACTIN as a loading control. Values were normalized to the signal of β-ACTIN and plotted as the percentage of WT using the Prism 8 software. Potassium cyanide (KCN)-sensitive endogenous cell respiration was measured polarographically using a Clark electrode in the indicated cell lines. Cell respiration rates are presented as nmol of $O_2$ consumed per one million cells per minute.

## Cryo-EM data acquisition
Five cryo-EM datasets of mitoribosomal complexes were collected in total using EPU 1.9 (Supplementary Table 1). First, 3 µL of ~120 nM mitoribosome was applied onto a glow-discharged (20 mA for 30 s) holey-carbon grid (Quantifoil R2/2, copper, mesh 300) coated with continuous carbon (of ~3 nm thickness) and incubated for 30 s in a controlled environment of 100% humidity and 4 °C. The grids were blotted for 3 sec, followed by plunge-freezing in liquid ethane, using a Vitrobot MKIV (FEI/Thermofischer). For high resolution mitoribosome analysis, five separate datasets were collected on FEI Titan Krios (FEI/Thermofischer) transmission electron microscope operated at 300 keV, using C2 aperture of 70 µm and a slit width of 20 eV on a GIF quantum energy filter (Gatan). A K2 Summit detector (Gatan) was used at a pixel size of 0.83 Å (magnification of 165,000X) with a dose of 29-32 electrons/Å$^2$ fractionated over 20 frames. A defocus range of −0.6 to −2.8 µm was used. More detailed parameters for each one of the datasets are listed in Supplementary Table 1.

## Cryo-EM data processing

The complete workflows of the mitoribosomal complexes A/A-P/P-E/E, A/P-P/P-E, L1 stalk, monosome consensus map, SSU and their 8 masked regions are given in Supplementary Figs. 1 and 2. For all the datasets, beam-induced motion correction and per-frame B-factor weighting were performed for all data sets using RELION-3.0.2[36,37]. Motion-corrected micrographs were used for contrast transfer function (CTF) estimation with gctf[95]. Unusable micrographs were removed by manual inspection of the micrographs and their respective calculated CTF parameters. Particles were picked in RELION-3.0.2[36,37], using reference-free followed by reference-aided particle picking procedures. Reference-free 2D classification was carried out to sort useful particles from falsely picked objects, which were then subjected to 3D classification. 3D classes corresponding to unaligned particles and LSU were discarded and monosome particles were pooled and used for 3D auto-refinement yielding a map with an overall resolution of 2.9–3.4 Å for the five datasets. Resolution was estimated using a Fourier Shell Correlation cut-off of 0.143 between the two reconstructed half maps. Finally, the selected particles were subjected to per-particle defocus estimation, beam-tilt correction and per-particle astigmatism correction followed by Bayesian polishing. Bayesian polished particles were subjected to a second round per-particle defocus correction. A total of 994,919 particles from all datasets were then pooled and separated into 86 optics groups in RELION-3.1[38], based on acquisition areas and date of data collection. Beam-tilt, magnification anisotropy and higher-order (trefoil and fourth-order) aberrations were corrected in RELION-3.1[38]. Subsequently, 509,691 particles from datasets 2-4 were pooled together and subjected to 3D auto-refinement giving a final nominal resolution of 2.21 Å. Masked refinement was performed on the small subunit head, body, mS39 and tail to produce maps of 2.36 Å, 2.31 Å, 2.44 Å and 2.45 Å resolution, respectively and likewise for the SLU body, L10-L7/L12 stalk, CP and L1 stalk to yield maps of 2.08 Å, 2.38 Å, 2.36 Å and 2.89 Å resolution, respectively. To improve the quality of the L1-stalk map, 220,906 monosome particles with high occupancy of E/E-tRNA were pooled and subjected to partial signal subtraction using the mask that retains the E/E-tRNA and L1-stalk, followed by 3D auto-refinement to have a nominal resolution of 2.87 Å (Supplementary Fig. 1 and Supplementary Table 1).

For separating the A/A-P/P-E/E and A/P-P/E states, all 994,919 monosome particles were subjected to signal subtraction using the mask that retains the tRNA binding region. The tRNA binding sites displayed weak densities corresponding to a heterogeneous mixture of tRNAs bound with partial occupancy. Signal subtracted particles were 3D classified without alignment using a mask on the A-site. This yielded two main classes, one with density in the A-site (332,187 particles) and the other lacking it (629,597 particles). Particles with density in the A-site were selected and subjected to a second round of 3D-classification using a mask around the P- and E-sites. Two classes were selected, one with P/P- and E/E-tRNA density and other in hybrid state with P/E-tRNA density. All 3D classifications with signal subtracted particles were performed using a T-value of 400. The subtracted signal was reverted for these two classes, corresponding to the A/A-P/P-E/E state (82,522 particles) and A/P-P/E state (20,143 particles). Following 3D auto-refinement, the nominal resolution of the cosmplexes was 2.63 Å and 2.98 Å, respectively. Masked refinement for the classical state was performed using local masks on SSU head, body, tail, and mS39 to produce maps at 2.72 Å, 2.75 Å, 2.91 Å and 2.93 Å resolution, respectively. For the LSU body, L10-L7/L12, CP, and the L1 stalk regions, the resolution was 2.44 Å, 2.84 Å, 2.78 Å and 3.20 Å, respectively. Masked refinement for the hybrid state was performed using the same local masks as for the classical state but transformed to compensate for the structural differences. This produced maps for the SSU head, body, tail, and mS39 to produce maps of 3.03 Å, 3.07 Å, 3.39 Å and 3.47 Å resolution, respectively. For the LSU body, L10-L7/L12 region, CP, and the L1 stalk masks, the resolution was 2.84 Å, 3.29 Å, 3.25 Å and 3.53 Å, respectively. The maps were then subjected to modulation transfer function correction, automatic *B*-factor sharpening and local resolution filtering using RELION-3.1 (Supplementary Fig. 2, Supplementary Table 1).

## Model building and refinement

We started by building a model into the 2.21 Å resolution consensus map. The starting model for the LSU was taken from PDB ID: 6ZSG, and for the SSU from PDB ID: 6RW4. These models were rigid body fitted using UCSF Chimera[96]. Individual local-masked refined maps with local-resolution filtering and *B*-factor sharpening (RELION 3.1) superposed to the overall map were combined into a single composite map using Phenix.combine_focused_maps[97] for model building and refinement. *Coot* v0.9[98] with Ramachandran and torsion restraints was used for model building.

The L1-stalk model was built using the 2.87 Å resolution masked-refined map of L1 stalk from E/E-tRNA-containing particles. Initial model for uL1m was generated by homology-modeling with uL1 structure of *T. thermophilus* (PDB ID: 3U4M) as template using the Swiss-Model webserver[99]. The N-terminal extension of uL1m (residues 61–93), the C-terminal domain (residues 158–254) and the preceding loop (residues 148–157) of uL9m, and H76-77 (residues 2761–2786) of 16 S rRNA were modeled manually into the density. Mitochondria-specific C-terminal extension of uL9m was modeled as a helix based on the density and secondary structure prediction by PSIPRED[100]. The model was rigid body fitted into the respective local-masked refined maps for L1 stalk region in the A/A P/P E/E and A/P/E states followed by self-restrained real space refinement in *Coot* v0.9.

In the CP, the tRNA^Val could be completely modeled, owing to the improved resolution of the local masked-refined map (2.36 Å). The 3′-CCA terminus (C74–A76) and a valine residue linked to A76 could be placed on the density map. The N-terminus of mL40 was extended by 13-residues (47–59) modeled in a tubular wedged between the A- and P-site tRNAs to the phosphate backbone of mRNA residue 7. Similarly, the C-terminus of bL31m was extended by 31 residues (95–125) from the C-terminal helix onto SSU head. The local resolution was sufficient for accurate side-chain modeling for most residues and secondary-structure assignments. We modeled a disulfide linkage between C45-C76 of mS37. However, they might exist in both disulfide-linked, and reduced states as indicated by the density and a water mediated hydrogen bond with a neighbouring Arginine side chain (R240 from uS7m). The other pair C55-C66 appears to be reduced. However, as we had added 1 mM DTT to our ribosome purification buffers, this is not truly indicative of the native state of C45-C76 or C55-C66 pairs.

At the mRNA channel entrance, a more accurate and complete model of mS39 could be built with 29 residues added to the structure. Improved local resolution enabled unambiguous assignment of residues to the density which allowed us to address errors in the previous model. Finally, a total of 28 α-helices could be modeled in their correct register and orientation. Further, a 28-residue long N-terminal loop of mS31 (residues 247–275) along mS39 and mitochondria-specific N-terminal extension of uS9m (residues 53–70) approaching mRNA were modeled manually by fitting the loops into the density maps.

Polyamines were identified upon a further manual inspection of the density map for the presence of continuous tubular densities which corresponded to spermine, spermidine or putrescine in length. A total of one spermine, four spermidines and one putrescine were placed in the model, justified by the electrostatic and hydrogen-bonding interactions that each polyamine exhibits at its amine groups with its neighbors.

Metal ions, $Mg^{2+}$, $K^+$ and $Zn^{2+}$, were modeled into densities based on a manual inspection of small density blobs and the chemical environment, specifically, coordination geometry and strength of density in both unsharpened and local resolution filtered maps. The coordination distance of $Mg^{2+}$ is around 2 Å with an octahedral geometry, whereas that of $K^+$ is around 3 Å with more relaxed constraints on coordination geometry. Notably, a density that was modeled previously as $Zn^{2+}$ coordinated by residues D224, D240, D241 and H93 of uS2m was assigned as a magnesium ion, indicated by the octahedral geometry of coordination with the uS2m residues mentioned above and two water molecules.

A-, P- and E-site tRNAs, and mRNA were modeled manually in the density. The density for tRNAs and mRNAs comes from a natively-bound heterogeneous population. Hence, individual residues were modeled as one of the nucleotides, A, U, G, or C, based on the density and/or conservation. While the 70 residues of P-site tRNA could be modeled, fewer residues could be modeled for the A-site (22 residues) and E-site tRNAs (43 residues), including the acceptor arm and anticodon loop due to weaker respective densities. Along the mRNA channel, 34 residues of mRNA were built. In the tRNA binding region, codon-anticodon pairs were clearly resolved. Additionally, extra residues near the A-site (10-12, 14-16) and at the mRNA channel entrance (25, 27-29) were modeled accurately into density. For the remaining mRNA chain at a relatively lower resolution, correct overall chain trace and nucleotide orientation could be assigned. This allowed specific protein-nucleotide interactions to be identified for uS5m, uS7m, uS9m, uS12m, mS35, mS39 and mL40.

For modeling the classical A/A P/P E/E- and hybrid A/P P/E-states, the LSU and SSU from the consensus model were fitted into their composite maps, followed by manual revision. As described above, tRNAs and mRNA were modeled manually into the density.

The water molecules were automatically picked by *Coot* v0.9[61], followed by manual revision. Geometrical restraints of modified residues and ligands used for the refinement were calculated by Grade Web Server (http://grade.globalphasing.org) or obtained from CCP4 library[101]. Hydrogens were added to the models except for water molecules by REFMAC5[102] using the prepared geometrical restraint files. Charged α-amino-group hydrogens (H2 and H3) and a free-ribose hydrogen HO3′ were removed from non-terminal protein and RNA residues, respectively, while those at terminal residues were kept. Protonation of Histidine residues were judged based on the chemical environment and removed or kept their N-H hydrogens (HD1 and HE2) accordingly. Protonation/deprotonation of ligands and modified residues were also adjusted.

Final models were further subjected to refinement with Phenix.real_space_refine v1.13_2998[97], wherein three macro-cycles of global energy minimization with reference restraints (using the input model as the reference, sigma 5-7) and secondary structure restraints, rotamer restraints but without Ramachandran restraints were carried out for each model. The composite maps were used as the targets for the refinement. An 'edit' file that defines metal-coordination bonds was generated by ReadySet in the Phenix suite and used. Non-canonical covalent bonds, which are those between tRNA$^{Val}$ and Valine and between 1-methyl-isoaspartate 184 and G185 in uS11m, were also defined in the 'edit' file. Model refinement data are listed in Supplementary Table 1.

## Bioinformatic analysis of the Y-N-C-Y motif

The density and base-specific interactions of mRNA residues 25-28 with mS39 imply a sequence of pyrimidine, any nucleotide, cytosine, pyrimidine, leading to a consensus motif: Y-N-C-Y. We explored the possibility of increased frequency of this motif in mRNA sequences. The sequences for all 13 mRNAs were obtained from GenBank. The consensus motif was searched in each sequence and all occurrences (including ones that overlap) were recorded and used to calculate the percentage frequency of occurrence for each transcript and the average percentage frequency. To determine if there is an enrichment of this motif in mRNAs, we calculated its occurrence in a set of nuclear-encoded mRNAs. To create this set, we acquired sequences randomly from GenBank and subjected them to a 90% redundancy cut-off, resulting in a total of 1387 sequences. The percentage frequency of occurrence of Y-N-C-Y motif was determined for each of the 1387 sequences and used to calculate an average percentage frequency for the entire set. Next to determine the distribution of the Y-N-C-Y motif across the length of mRNAs, each occurrence was plotted against its position number in the mRNA sequence. This distribution for all 13 mRNA sequences was plotted together for comparison.

## Molecular dynamics simulations: modified force fields for simulating a truncated mL64 protein and modified interactions with protein uL1m

### Baseline force field used for simulations of P/E formation.
An all-atom multi-basin structure-based "SMOG" model[59,60] of the mitoribosome was used to simulate P/E formation. For this, we first constructed single-basin structure-based models for the pre- and post-translocation configurations (A/A-P/P and P/P-E/E). We then combined the two models, such that the composite model stabilizes both the pre- and post-translocation configurations. Here we describe the baseline model, which is described in the main text. In subsequent sections, we provide detailed descriptions of all modified force fields.

In a single-basin all-atom structure-based model, every non-hydrogen atom is represented by a bead of unit mass, and an experimentally-obtained structure is used to define the potential energy minimum. The functional form of the potential is given by equation (1):

$$U = \sum_{bonds} \frac{\epsilon_r}{2}(r_i - r_{i,0})^2 + \sum_{angles} \frac{\epsilon_\theta}{2}(\theta_i - \theta_{i,o})^2$$
$$+ \sum_{impropers} \frac{\epsilon_{\chi imp}}{2}(\chi_i - \chi_{i,0})^2 + \sum_{planars} \epsilon_{\chi planar}[1 - \cos(2\chi_i)]$$
$$+ \sum_{backbone\ dihedrals} \epsilon_{bb}F(\phi_i - \phi_{i,0}) + \sum_{sidechain\ dihedrals} \epsilon_{sc}F(\phi_i - \phi_{i,0})$$
$$+ \sum_{contacts} \epsilon_c \left[ \left(\frac{\sigma_{ij}}{r_{ij}}\right)^{12} - 2\left(\frac{\sigma_{ij}}{r_{ij}}\right)^6 \right] + \sum_{non-contacts} \epsilon_{nc} \left(\frac{\sigma_{nc}}{\sigma_{ij}}\right)^{12}$$

where

$$F(\phi) = [1 - \cos(\phi)] + \frac{1}{2}[1 - \cos(3\phi)]$$

$r_0$ and $\theta_0$ parameters are assigned values found in the Amber ff03 force field[103]. The dihedral parameters $\phi_0$ are defined by the experimental structure. $\chi_0$ non-planar improper dihedrals (around chiral centers) are also defined to have the values adopted in the experimental structures. Planar dihedral angles are given a cosine term with periodicity 2 and minima at 0 and 180 degrees. The energy scale is set to $\epsilon_r = 100 \frac{\epsilon}{A^2}$, $\epsilon_\theta = 80 \frac{\epsilon}{rad^2}$, $\epsilon_{\chi imp} = 10 \frac{\epsilon}{rad^2}$, $\epsilon_{\chi planar} = 40 \frac{\epsilon}{rad^2}$, where $\epsilon$ is the reduced energy scale. The dihedral and contact energy scales $\epsilon_{bb}, \epsilon_{sc}, \epsilon_C$ are normalized as in ref. 59. This combination of structure-based non-bonded terms and AMBER bonded terms is identical to the implementation of a previous SMOG-amber variant (called SBM_AA-amber-bonds[104] in the SMOG 2 force field repository (called SBM_AA-amber-bonds in the SMOG 2 force field repository at smog-server.org), though post-transcriptionally-modified residues were added for the current model. Upon publication, the SMOG 2 force field templates used in the current study will be available for download on the SMOG 2 force field repository (ID: AA_PTM_Hassan21.v1).

For the non-bonded interactions, each atomic contact that is found in the experimental structure (i.e. "native" contact) is given an attractive Lennard-Jones-like interaction that stabilizes a preassigned structure. Native contacts were defined based on the Shadow Contact Map algorithm with default parameters[105]. $\sigma_{ij}$ are the interatomic distances between contacts in the native structure, multiplied by 0.96. As employed previously in simulations of the ribosome[106,107] and viral capsids[104], this scaling of the contact distance was employed to avoid artificial expansion of the ribosome that results from thermal energy. That is, while the potential energy is defined such that the experimental structure is the global minimum, thermal energy leads to a free-energy minimum in which the ribosome is slightly expanded. By shortening the stabilizing interactions, the free-energy minimum is consistent with the experimental structure. For this comparison, the radius of gyration ($R_g$) was used as a measure of expansion. With this scaling of the contact distances, $R_g$ was found to match that of the cryo-EM model. All non-native contacts are given a repulsive term that ensures excluded-volume steric interactions are present. The parameter $\sigma_{nc}$ is given the value 2.5 Å, while $\epsilon_{nc} = 0.1\epsilon$.

Stabilizing intra-ribosome contacts and dihedrals were assigned the values found in the A/P-P/E (SSU rotated) hybrid structure, which is described in the main text. This ensures that the SSU rotates forward in each simulation. Each simulation is initiated from the A/A-P/P conformation. All intra-mt-tRNA and intra-mt-mRNA contacts were defined based on the A/A-P/P structure. Since stabilizing contacts in a structure-based model describe effective interactions[108,109], we rescaled the strengths of subsets of contacts within the model. Consistent with recent simulations of P/E formation in bacteria[106], interactions that stabilize the structure of the ribosome (i.e. intra-ribosomal) are given larger weights than contacts that are only transiently-formed (e.g. tRNA-ribosome contacts). Relative to intra-ribosome contacts, the energetic interactions between the mt-mRNA-tRNA complex and the SSU (in A/A-P/P and P/P-E/E structures) were rescaled by a factor of 0.3. While these interactions were weaker than intra-ribosomal contacts, since the simulations were not performed under conditions in which translocation will occur (e.g., binding of elongation factors or back-rotation, etc), the mt-mRNA maintained its initial position on the mRNA track of the SSU. P/P-E/E contacts between the mt-tRNA and the uL1m protein were rescaled by a factor of 0.8 and all other interactions between mt-mRNA-tRNA and the LSU were rescaled by a factor of 0.1. This rescaling was applied to account for the transient nature of mt-mRNA-tRNA binding to the ribosome. The strength of the uL1m-tRNA contacts was set such that the P/E configuration would stably form contacts with the uL1m protein. When contacts were scaled below a factor of -0.5, we found that uL1m could not form stable interactions with the mt-tRNA.

Interactions between the 3'-CCA ends of A-site and P-site mt-tRNAs with the A-site and the P-site were scaled by a factor of 0.2, while their interactions with P-site and E-site were scaled by a factor of 0.8. This leads to the mt-tRNA molecules favoring binding of their target sites (i.e., sites bound in the P/E configuration) on the ribosome. While the interactions with the subsequent binding site (i.e., A-site mt-tRNA with the P-site and P-site mt-tRNA with the E-site) were stronger, these contacts are short range (~6 Å). Accordingly, the stronger weights do not affect the dynamics of the mt-tRNA until it is very close to the binding site. Thus, the details of the final repositioning of the 3'-CCA tail of the P-site mt-tRNA should not be overinterpreted. While the contacts are short-range, the structure of the mitoribosome may introduce steric barriers that the mt-tRNA must navigate in order to adopt the P/E configuration[109]. With these considerations in mind, the current model is aimed at identifying structural elements that can impose steric limitations on mt-tRNA during P/E formation.

There were also two sets of harmonic potentials that were introduced. To mimic the presence of a nascent peptide chain, which would restrain the 3'-CCA end of the A-site tRNA to the PTC, we introduced weak harmonic restraints (weight of $1/nm^2$) between the 3'-CCA end of the A-site tRNA and the P site of the LSU. Distances were assigned based on the P/P conformation. These weak interactions ensured that the 3'-CCA end remains localized to the PTC. We also introduced harmonic potentials between base-pairing contacts in the mt-tRNA acceptor arm (weight $1000/nm^2$). Due to the details of the contact map, initial simulations exhibited partial unfolding/refolding of the acceptor arm during P/E formation. Since that was likely an artifact due to the simplicity of the model and the details of the contact map definition, these additional harmonic terms were introduced to ensure the tRNA acceptor arm remained folded throughout the transition.

**Modified force fields.** Multiple variants of the model (i.e., force field) were considered, in order to investigate the influence of steric interactions between the P-site mt-tRNA and C-terminal residues of mL64 and to investigate the influence of attractive interactions between uL1m and P-site mt-tRNA on P/E formation.

**Truncated-mL64 model: ΔmL64 force field.** The C-terminal region of mL64 is involved in steric interactions with the mt-tRNA. For ΔmL64 model, the C-terminal region of mL64 (beginning with Q112) was removed from the system. Q112 was chosen as truncating here guarantees no steric interactions between mL64 and the mt-tRNA and keeps the rest of the protein intact. Thus, all other interactions were unperturbed. This model is intended to mimic an ideal experiment, where one could delete the terminal region of mL64 without introducing indirect effects on the dynamics. In our application, this model allows us to ask what effect steric interactions between P-site mt-tRNA elbow and mL64 have on the dynamics of P/E formation. Deletion of this region from the model was achieved using the smog_extract tool of the SMOG 2 software package[59].

**Modified-uL1m model: ΔuL1m force field.** All stabilizing contacts between uL1m and the P-site mt-tRNA that are formed in the P/P-E/E state were scaled by a relative factor of 0.1, rather than 0.8 (baseline model, above). Thus, by weakening the attractive interactions, there is insufficient stabilizing energy between uL1m and the P/E tRNA, such that the tRNA elbow does not fully reach the P/E configuration.

**Combined truncated mL64 and modified uL1m model: ΔmL64-ΔuL1m force field.** All modifications applied in the truncated mL64 model and the modified-uL1m model were included in a single combined force field.

**Simulation details.** All simulations were initiated from the pre-translocation A/A-P/P configuration. All force field files were generated using the SMOG 2 software package[59]. Molecular dynamics simulations were performed using Gromacs (v5.1.4)[110,111]. Reduced units were used in all calculations. Energy minimization was first performed using steepest descent minimization. The system was then coupled to a heat bath using Langevin Dynamics protocols, with reference temperature of 0.5 $\epsilon/k_B$ reduced temperature units (60 Gromacs units). This temperature was chosen since it has been shown to produce structural fluctuations that are consistent with those inferred from anisotropic $B$-factors and explicit-solvent simulation[112]. The timestep was 0.002 reduced units and each simulation was continued until the distance of the 3'-CCA end from the LSU E-site $R_{CCA}$ reached a value less than 4 Å. Energy minimization was initially performed using steepest descent minimization. Each simulation was performed for a maximum of $10^8$ timesteps. For the primary/baseline model described in the main text (327 simulations) we obtained an aggregate simulated time of 2.27 $*10^6$ reduced units. For the truncated mL64 model (ΔmL64 model) (321 simulations), the aggregate simulation time was 1.87 $*10^6$ reduced units. For the attenuated uL1m contacts model (ΔuL1m model) (237 simulations), the aggregate simulation time was 1.76 $*10^6$ reduced

units. For the ΔmL64-ΔuL1m model (181 simulations), the aggregate simulation time was $1.04 * 10^6$. Previous comparisons of diffusion coefficients of tRNA molecules in the ribosome (explicit-solvent simulations vs. SMOG models), estimate that 1 reduced unit corresponds to an effective timescale of approximately 1 nanosecond[113]. Using this conversion factor, the presented simulations represent an aggregate effective simulate time of ~ 6.94 milliseconds.

**Structural metrics for describing simulated dynamics.** - A-site mt-tRNA, $R_{CCA}^P$: distance between the geometric centers of the side chains of C75 in the A-site mt-tRNA and Gm2815 in the LSU rRNA. These residues base pair in the A/P-P/E and P/P-E/E configurations.

- P-site mt-tRNA, $R_{CCA}^P$: distance between the geometric centers of the side chains of C75 in the P-site mt-tRNA and Gm2815 in the LSU rRNA. These residues base pair in the A/P-P/E configuration.

- $R_{CCA}$: distance between the geometric centers of the side chains of A76 in the P-site mt-tRNA and G2909 in the LSU rRNA. G2909 in the E-site forms a base stacking interaction with A76 of P-site mt-tRNA in the A/P-P/E configuration.

- $R_{elbow}$: distance between geometric centers of U54 of P-site mt-tRNA and its reference position in the A/P-P/E configuration after alignment of the LSU rRNA.

- $Q_{L1-tRNA}$: the fraction of P/P-E/E contacts between uL1m and P-site mt-tRNA. A contact is defined as formed if it is within a factor of 1.2 of the distance found in the P/P-E/E configuration.

**SSU rotation and tilt angles.** To follow the rotary motions of the SSU body with respect to the LSU, we calculated rotation and tilt angles for both the SSU body. For this, we first determined the corresponding *E. coli* numbering of the mitoribosome rRNA residues via STAMP alignment[114] of the mitoribosome to a reference *E. coli* structure (PDB ID: 4V9D, [https://doi.org/10.2210/pdb4V51/pdb]). Alignment was performed separately for the LSU, SSU head and SSU body. The angles were then calculated, using the assigned *E. coli* numbering.

**Structural model preparation for simulations.** Since the cryo-EM models did not resolve the exact same set of atoms, there was a need to perform some additional structural modeling steps prior to performing simulations. For example, to define the multi-basin structure-based model, it was necessary for each mt-tRNA molecule to have identical atoms in the pre- and post-translocation configurations. Additionally, to define the intra-ribosome interactions to stabilize the A/P-P/E conformation, it was necessary that all mt-rRNA and mitoribosomal proteins had identical atomic composition. While most of the resolved atoms are common to all reported models, we employed restraint-based modeling to construct models of the mitoribosome and mt-tRNA that have identical compositions. Below, we describe the steps required for modeling the mitoribosome and mt-tRNA.

**Structural modeling pre-processing steps for rRNA and proteins for simulations.** Since the structure-based model only included intra-ribosome interactions from the A/P-P/E structure, we first removed all mt-tRNA and mt-mRNA from the A/A-P/P and A/P-P/E models. We also removed ligands and water molecules (HOH, SPD, PUT, NAD, SPM). For the remaining atoms, we identified all atoms that were common to the A/A-P/P and A/P-P/E structures. We then generated a single-basin structure-based model of the A/A-P/P structure. Next, position restraints (harmonic terms, weight of 100 for each atom) were introduced for all common atoms, with positions corresponding to the A/P-P/E structure. The system was then energy minimized via steepest descent minimization. Then, the system was subject to simulated annealing. The system was initially simulated at 0.15 reduced temperature units for 25,000 timesteps. The temperature was then linearly decreased to 0 over the next 50,000 timesteps. An additional 25,000 steps were simulated at temperature 0. The final, annealed structural model was used for force field generation steps, as described above.

**Structural modeling pre-processing steps for mt-tRNA and mt-mRNA.** Since the cryo-EM reconstructions reported consensus/average sequences for each mt-tRNA, the composition of the A-, P- and E-site mt-tRNAs were not identical. To perform simulations, we homogenized the composition of the mt-tRNA molecules in the A/A-P/P-E/E structure and the mt-tRNA codons. For this, we started with the structural model of the A-site mt-tRNA (and the associated codon) from the A/A-P/P structure, and then used it to generate a structure-based model. We then introduced position restraints based on the positions of the A-site mt-tRNA in the A/A-P/P-E/E structure. All common atoms (*i.e.*, backbone atoms, or atoms in identical residues) were restrained to the A-site mt-tRNA/mRNA positions. For the A-site mt-tRNA, restraints were not imposed on residues 46–58 and 16–17. Cycles of minimization and annealing were performed, where the position restraints were given weights of 1, 10, 100 and then 1000. This process was repeated for the P-site and E-site mt-tRNAs, as well as for the upstream and downstream mt-mRNA. This process produced a model of the A/A-P/P-E/E configuration that has repeating codons and identical mt-tRNA composition. This structural model was used to define the pre-translocation and post-translocation structure-based models, which were used to construct the multi-basin model.

**Describing the sequence of rearrangements in simulations.** To describe the dynamics of the P-site mt-tRNA during P/E state formation, we have defined three states based on the 2D probability distribution of $R_{elbow}^E$ and $R_{CCA}^E$ distances (Fig. 8). These states are defined as all configurations of the P-site mt-tRNA in the baseline model that sample the following values of $R_{elbow}^E$ & $R_{CCA}^E$:

state $I_1$: disk centered at $(R_{elbow}^E, R_{CCA}^E) = (46,35)$ Å with radius r = 6 Å
state $I_2$: disk centered at $(R_{elbow}^E, R_{CCA}^E) = (63,23)$ Å with radius r = 8 Å
state $I_3$: disk centered at $(R_{elbow}^E, R_{CCA}^E) = (42,13)$ Å with radius r = 7 Å

Using this definition of the intermediate states, the most common sequence of transitions followed the order P/P => $I_1$ => $I_2$ => $I_3$ => P/E.

## Structure analysis and figures

Visualization and analysis of the models and maps was carried out using UCSF Chimera and ChimeraX[96,115]. For model and map comparisons, models were superposed in *Coot* v0.9[61] using the Secondary Structure Matching algorithm and maps downloaded from EMDB were resampled on our maps in UCSF Chimera. Simulated snapshots were visualized using Visual Molecular Dynamics[116].

## Statistics and reproducibility

All experiments were done in at least biological triplicate unless otherwise stated. GraphPad Prism 8 software was used for statistical analyses of the data. One-way analysis of variance (ANOVA) was performed, comparing the mean of each cell line with the mean of the mS29-KO cell line reconstituted with WT mS29, followed by Dunnett's multiple comparisons test. The data is presented as the mean ± SD; * $P \leq 0.05$, ** $P \leq 0.01$, *** $P \leq 0.001$, **** $P \leq 0.0001$. For immunoblotting, band density is quantified and normalized to ACTIN. For metabolic labeling of mitochondrially translated peptides in whole cells, the data presented represents the mean of each sample's $^{35}S$ signals.

## Reporting summary

Further information on research design is available in the Nature Portfolio Reporting Summary linked to this article.

## Data availability

The atomic coordinates were deposited in the RCSB Protein Data Bank, and EM maps have been deposited in the Electron Microscopy Data

bank under accession numbers: 7QI4 and EMD-13980 (consensus monosome), 7QI5 and EMD-13981 (classical state), 7QI6 and EMD-13982 (hybrid state). The atomic coordinates that were used in this study: 3U4M (uL1 from *T. thermophilus*), 4V51 [https://doi.org/10.2210/pdb4V51/pdb] (*T. thermophilus* ribosome with mRNA, tRNA), 4V9D [https://doi.org/10.2210/pdb4V9D/pdb] (*E. coli* ribosome), 4Y4P [https://doi.org/10.2210/pdb4Y4P/pdb] (*T. thermophilus* ribosome with A-, P-, E-site tRNAs), 6QNR [https://doi.org/10.2210/pdb6QNR/pdb] (*T. thermophilus* ribosome with experimentally assigned K$^+$ ions) 6RW4 [https://doi.org/10.2210/pdb6RW4/pdb] (mtSSU with mtIF3), 6ZM5 [https://doi.org/10.2210/pdb6ZM5/pdb] (mitoribosome from actinonin-treated cells) 6ZSG [https://doi.org/10.2210/pdb6ZSG/pdb] (mitoribosome with mRNA, tRNA), 6ZTJ [https://doi.org/10.2210/pdb6ZTJ/pdb] (*E.coli* expressome), 6ZTN [https://doi.org/10.2210/pdb6ZTN/pdb] (*E.coli* expressome), 7K00 [https://doi.org/10.2210/pdb7K00/pdb] (*E. coli* ribosome at 2 Å resolution). Source data are provided with this paper.

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

## Acknowledgements

We thank the EMBO Young Investigator Program, the Northeastern University Discovery cluster and Northeastern University Research Computing staff. Swedish Foundation for Strategic Research (FFL15:0325), European Research Council (ERC-2018-StG-805230), Knut and Alice Wallenberg Foundation (2018.0080), Japan Science and Technology Agency (19198049), NIH-R35 grant (GM118141 to A.B.), and MDA Research Grant (MDA-381828 to A.B.). V.S. is supported by the Horizon 2020 - Marie Sklodowska-Curie Innovative Training Network (721757). Y.I. was supported by H2020-MSCA-IF-2017 (799399-Itohribo). The SciLifeLab cryo-EM facility is funded by the Knut and Alice Wallenberg, Family Erling Persson, and Kempe foundations. P.C.W. and A.H. were supported by NSF grant MCB-1915843. Work at the Center for Theoretical Biological Physics was also supported by the NSF (Grant PHY-2019745). We would like to thank AMD for the donation of critical hardware and support resources from its HPC Fund that made this work possible. We extend our gratitude to Per Ljungdahl and Olli Kallioniemi for their contributions, despite the attempt to delay publication of our work.

## Author contributions

V.S., Y.I., S.A., and J.A. collected cryo-EM data. V.S. and Y.I. processed the data and built the models. V.S., Y.I., R.K.F., A.N., S.A. and A.A. carried out structural analysis. Y.N., K.I. and M.T. performed mass-spec characterization. S.L.D. and A.B. performed biochemical characterization. A.H. and P.C.W. performed molecular simulations. A.A. coordinated the manuscript writing with contributions from all authors.

## Competing interests

The authors declare no competing interests.
