## [Peer Review File · Nature Communications]

Mitoribosome structure with cofactors and modifications reveals mechanism of ligand binding and interactions with L1 stalkREVIEWER COMMENTS

Reviewer #1 (Remarks to the Author):

The authors use cryo-EM, Stable Isotope-Labeled RNA as an internal Standard (SILNAS), molecular simulation and functional validation to reveal the detailed interactions between the ligands of the ribosome (tRNA and mRNA) and the ribosome with unprecedented detail, suggesting key roles for modified nucleotides. The authors present high resolution models of the mitochondrial ribosome at high resolution (2.2. Å), based on cryo-EM reconstructions. At this resolution, they are able to identify modified nucleotides. The authors find that the modified nucleotides are positioned to help align the ligands (tRNA and mRNA) in the ribosome. Specifically, the high resolution map allowed of modeling of 13 rRNA modifications, 7 protein modifications, and 6 polyamines. The maps reveal the detailed interactions between the mRNA and proteins mS39, uS5m and mS35, suggestive of a gating mechanism upon mRNA delivery. The authors also provide structural and functional evidence showing that GDP binds to protein mS29 to effect conformational selection. Finally, their combined study of cryo-EM and molecular simulation shows that E-tRNA functions without rRNA interactions, in contract to bacterial and eukaryotic systems, with L9 acquiring a new functional role.

The study represents a strong effort and could be published, provided the authors make the following minor changes.

General comments: While the data is excellent and strong, the manuscript needs improvement. I recommend making the main story line clearer throughout the manuscript, rather than jumping from topic to topic. There is little to no discussion, so this should be expanded to explain the implications, challenges, previous work and future directions in much more detail.

Specific comments:

Title: title could be improved by describing the main findings of the study

Decoding center: the authors should include more discussion comparing their decoding center to the canonical bacterial decoding center (i.e., A1492, A1493, G530 and S12).

Figures

Figs. 1-2: Caption needs much more detail, including descriptions of all colors. Everything that appears in the figures needs to be described in the figure caption.

Fig. 1 panel c: here, the density of the modified nucleotides in isolation is shown. It was difficult to find density depicting interactions between the modified nucleotides and other part of the complex in the manuscript. Density depicting the connection of key modified nucleotides to their interaction partners should be shown, either in Fig.1 or in Fig.4.

Fig. 2, panel a: is the surface depicted based on the atomistic model or the cryo-EM density? This should be stated explicitly, so as to not mislead the reader. Ideally, the cryo-EM density should be colored with the electrostatic potential information (also in Fig. 3). The schematic is not described in the caption. Panel c, right hand side: what does the spacing of disks indicate? The relationship between d,e,f,g and h is loosely shown, but very difficult to understand. This should be explained explicitly in the figure caption. In addition, the appropriate individual residues should be labeled in the 'zoomed out' views to make clear relationships to the 'zoomed-in' views. These individual residues should be depicted in detail (e.g., bonds or sticks representations) in the 'zoomed out' views. The purple in panel (i) appears to be a surface rendering based on an atomistic model. If so, this should be depicted in a non-surface representation, so as not to confuse it with actual cryo-EM density.

Fig. 7: panel labels in the caption are mixed up. c and d are missing and a,b, e, f are mixed up. Panels h and i are referred to, but do not exist.

Reviewer #2 (Remarks to the Author):

See attached.

Singh et al. describes high resolution cryo-EM structure of human mitoribosome, and provides detailed information about roles of r-proteins in mRNA recognition, GDP-bound unique protein, characteristic L1 stalk, and cofactors including polyamines, and FeS clusters. This reviewer is impressed with comprehensive analysis of mitoribosome structure. However, I have a number of comments and concerns that should be addressed to improve this manuscript.

Since structural studies of mitoribosome have already been published in several papers, this reviewer does not figure out what are the new findings and progress provided in this study. Please explain this point clearly in introduction. In particular, I would like to know what part of protein density was not modeled in previous studies.

In abstract, “together with an experimental identification of the mitoribosomal RNA (rRNA) modifications” This reviewer does not find any new information about rRNA modifications. All these modifications have already been reported. They just confirmed previous studies. Please revise it with appropriate description.

I would suggest the authors briefly mention some details how they succeeded in getting the structure with higher resolution than previous structure.

L153, Fig e: In comparison to u5Sm density, mRNA densities are very sparse and almost undetectable except for G10-U11. This means mRNA is very flexible in these regions. I agree that u5Sm may support the correct path of mRNA toward the A site, but I think the authors cannot declare the contact between mRNA and u5Sm except for K239 and R237.

L155, 166 : R46 and R49 stabilize mRNA. It is necessary to put H-bonds between amino group and phosphates in Fig. 2g. Side chain density of Arg residue is visible?

L199: “The function of mL40 is to balance the charge between the A- and P-tRNAs.” This is intriguing, but it’s not clear what it means. Please discuss it more clearly. How do authors consider role of this basic peptide chain in translation?

L238-240: The authors should specifically point out the sites where GDP is bound. Regarding “newly identified GDP”, please explain why it was not observed in the previous structures and was observed in this structure.

L245-246: It is necessary to explain more clearly which process and what kind of structural change is specifically meant by “intersubunit dynamics” here.

L249-251: Switching of mS29 β -hairpin 208-216 should be indicated by the models (in classical state and hybrid state) based on actual densities instead of schematic depictions in fig.5c.

L253-256: Although they analyzed two double mutants (R177A/K245A and R177A/K295A), the results of R177A/K295A mutant were ignored in the main text. They should be added. Please explain why steady-state levels of COX1 and COX2 were significantly different between R177A/K245A and R177A/K295A, while the results from metabolic labeling were comparable.

L272-279: Distances and angle information written here can be put in the corresponding figures (Fig.6).

L285-288: Authors need to explain what exactly the delta-uL1m model is. Why Q112 was chosen for truncation?

L295-296: “The structure also shows that”, but the readers may not find any figure indicating this. They should prepare a figure showing that.

This reviewer was surprised to know that CP-tRNA is fully modified and also aminoacylated with Val. What is a role of aminoacylation for 39S assembly? Once tRNA is aminoacylated, aa-tRNA is tightly recognized by EF-Tu. The author might have some comments on CP-tRNA in ribosome assembly after aminoacylation.

All mito-rRNA modifications are visualized in this structure. This reviewer would like to know the structural effect and hypothetical role of mitochondria-specific rRNA modifications, especially for m⁵C1488, m⁵U1076 in 12S rRNA and m¹A2617 in 16S rRNA.

mRNA stabilization mechanism by uS12m, uS9m, uS7m are quite intriguing. Please compare roles of these r-Proteins in bacterial ribosome.

In Fig. 4a, the P-site codon is GUG, and anticodon is CAC? If this is tRNA^{Val}, its anticodon sequence is UAC. No mt-tRNA has CAC anticodon. Mitoribosomes contain mixture of endogenous mRNAs. Why mRNA sequence is so defined in this model?

There are 3 FeS clusters in mitoribosome. But, only one of them is shown in Fig. 1c Where can we find two others? What is the role of these FeS clusters? As 2Fe-2S cluster is very unstable, is there any density assignment or experimental verification of this

cluster?

“Polyamine and NAD functionally compensate” The authors do not mention NAD at all in the text.

Additional points

Fig. 1: No description for Ala2, Ser2, SPD and PUT nomenclatures in this figure’s legend. Also, the authors should add the explanation about the values of “level” and “res” in Fig.1c.

It is strange Fig. 1c is not mentioned in the text.

Fig. 3c: Need numbering of anticodon loop.

Fig. 3d: How do G164 and G165 stabilize mRNA?

Fig. 4h-j: Mention something from these figures.

Fig 7: labelling and legend are completely disordered. For example, g-i is shown in legend, but Fig 7h and 7i does not exist. Please check labeling and legends.

Fig. 8b: You should show the surrounding residues and interactions of these polyamines (or Mg, K ions in lower panel) in Figure. Add PDB ID for *E. coli* ribosome.

L117: 6 mitoribosome-specific modifications mean m⁵C1488, m⁵U1076 in 12S rRNA, m¹A 2617 in 16S rRNA, and 3 tRNA(Val) modifications? You should mention this in text or Fig1c.

L152: You should change Fig.2 to Fig.2a.

L185: Please show the position of ‘kink-2’ in Figure.

L188: Fig. 3f does not exist. Please check the labelling of Figure.

L194, Fig4a: In Fig 4a, G1485 is misspelled to C1485. And I think you should show the interaction between G1485 and mRNA in Fig 3a, because Fig 4a is focused on rRNA modifications around mRNA.

L195 : position 6 might be position 5?

L200, Fig3a: residues 51-55 are not shown in Figure. You should show these residues in Figure.

L214-L218: when referring to bacterial ribosome, showing the bacterial rRNA number is preferable (for example, U1563(m³U1498 in *E.coli*)).

L210-213: Unify the style of RNA modification. m⁴C1486 should be m⁴C1486. N⁴-methylated should be N⁴-methylated

L222 Fig4 typo m⁵1076 to m⁵U1076

Fig5e: the upper gel image is not clear. Please adjust the contrast to show S/N ratio is sufficient for analysis.

L282-L282: You should show this point in Figure.

Ex Fig 1: Labeling the PDB number in Figure is preferable. (for example, label ‘7QI5’ in E/E P/P A/A map)

Ex Fig5a: In figure, lane 4 is COX5B, but in legend, it is noted as uS3m. Please correct

it.

Reviewer #3 (Remarks to the Author):

The manuscript entitled 'The complete structure of human mitoribosome, roles of mito-specific protein elements, cofactors and rRNA modifications' by Singh et al. provides data on the structural details of human mitoribosome (rRNA and rProt modification, polyamines and ligand binding) explaining the mechanisms of mRNA/tRNA recognition. Despite the provided high-resolution structure (2.2 Å) most of the proposed mechanisms miss structural/biochemical evidence so I suggest that the manuscript as it stands is not suitable for publication in Nature communications and it should be further refined, including the models described therein. The key arguments why a publication cannot be accepted from this point on are as follows:

1) Despite the overall high quality of the map and the density present for most of the ligands proposed, there are plenty of errors in the model and ambiguous interpretation of the density. For example, SPD 1703 (chain AA) is not placed correctly in the density (figure 8b). Moreover, there are several examples where the same blob of density was treated as water and potassium ion (i.e. Water3857 (chain AA) and K3305 (chain A)). Due to the absence of additional evidence from other methods, we suggest that most potassium ions should be considered as magnesiums unless there is strict evidence provided. All the densities for the ligands and modifications should be revised and plotted at the same level, for example, Val101 in CP-tRNA is out of the density and the density of this region is very weak to consider it as Val or Glu (Figure 1). To sum up, all the models should be revised, hydrogens that are not visible in the density should be removed, as well as unmodelled blobs should be inspected and some parts of the model should be rebuilt, i.e. chain e, residues 246-251 which are not placed correctly in the map.

2) The hypothesis introduced in the paragraph concerning the mRNA recognitions is not consistent with the structural data observed. In the presented data, there is a strong density only for residues 4-10 and 27-29 out of 32 nts of provided mRNA sequence. It means that for 22 nts there is no density/weak density and it is not enough to make the statements. For example, as is shown in figure 2b there is no density for the 26th residue, and weak density for the 25th residue, so any conclusion could be done at a low level of confidence. Also, there is a question mark on the position of the protein mL40. According to the density, we can conclude that there is an interaction between the phosphate group of the 6th nucleotide of mRNA and the mL40 protein, but it seems ambiguous how the authors put the protein to the density because a big part of the protein in the intersubunit space is disordered (51-55 residues). Thus, it is not clear, how many residues are located in this gap and whether 47 residue (46? 48?) interacts with the mRNA. Moreover, there is also an interaction with the 6th nucleotide base, so the assumption that « mL40 contact represents a prominent feature that is independent of mRNA sequence » is not correct. I suggest omitting this part because there is not enough structural evidence for most of the suggestions proposed.

3) Concerning the GDP functions, the authors present structural data along with biochemical experiments. The role of GDP was elucidated by a comparison of the interaction between mS29 protein in presence of GDP or non-hydrolyzable analog GMPPNP. Authors are claiming that there are no structural changes occurred in mS39 protein, however from the density, there are different conformations for side chains of residues K245, H291, and K245 (figure 5b). Also, the authors suggest that there are « no changes in side chains engaged in intersubunit communication », however, there is no density for side chains to prove this hypothesis. Besides, the biochemical study does not seem fully

completed. Given the absence of influence of one residue mutation, the authors performed just 2 double mutations claiming that « GDP acts as a stabilizing agent for mS29 β -hairpin 208-216 to maintain the intersubunit communication ». To make this conclusion author based on the biochemical data of R177A/K245A mutation. To prove this hypothesis authors should investigate other mutants, for example, Y208/N238 to elucidate the pairs which keep the stability of GDP. Also, there is an error in the Y173 position, the side chain should be rotated to form a stacking interaction with GDP.

4) The mechanism of E-tRNA recognition which is performed by simulating approaches, is not supported by the structural data. To be confident that tRNA/L1 stalk motion follows the simulated pathway, intermediates should be captured proving the correlation between simulation and experiment. Also, there is an extra density for E-tRNA which is not built.

5) The paragraph on polyamine's action seems ambiguous for two reasons. Firstly, the authors make a comparison with the structure in presence of Streptomycin. In this structure (PDB 7P2E) the large subunit is missing, making it impossible to check the presence of polyamines substitution. Moreover, for SPD1703 in SSU, 4 molecules of water are missing (fig. 8b) which are present in the cryo-EM map of 7P2E). I assume that the whole density initially recognized as potassium and waters can be considered as a SPD molecule. Secondly, to date, all known studies show that polyamines are not interchangeable with any ions. Such a strong statement (335th row) should be at least double-checked. In this regard, the polyamine to ion substitution seems very ambiguous.

Based on these facts the manuscript in this edition could not be considered for publication. Authors should revise the manuscript/models and perform to make it ready for publication.

Reviewer #4 (Remarks to the Author):

The manuscript, "The complete structure of human mitoribosome, roles of mito-specific protein elements, cofactors and rRNA modifications", by Singh, Itoh, et al. is an outstanding contribution to our understanding of mitochondrial translation. Using a ~ 2.2 Angstrom cryoEM density map, these authors report the exciting discovery of a YNCY mito-mRNA sequence motif that engages mito-specific proteins and rRNA moieties that facilitate decoding. They further report on the role of GDP binding within the mito-specific protein mS29 and the importance of this site for motion coupling between the two ribosomal subunits. In addition to GDP, the authors highlight the structural roles played by polyamines like spermine, Fe-S clusters, and the metabolite NAD. Finally, the authors synthesized several tRNA structural states using molecular dynamics to simulate how tRNAs translocate from the P-site to the E-site. The structures and the dynamic simulation highlight unique mitochondrial features, especially of the protein mL64 and the L1 stalk that, by contrast with the cytoplasmic ribosome, govern translocation from the P- to E-sites and ejection of the spent tRNA. After considering the following suggestions, we recommend the manuscript be published without delay.

1. Regarding the maps and their interpretation:

Cys76-Cys45 have been reported to be redox active, cycling between disulfide and reduced forms, and a disulfide looks plausible in the map. Nearby, Arg AF:240, Cys A2:45, Met A2:49, Met A2:53, and Ile AF:156 all appear to fit poorly. With more attention to model building in this region, Cys45 appears deprotonated and in a stable interaction with Arg240. Arg240 also H-bonds to the Met49 sulfur. Please cite the prior work on the redox chemistry in this vicinity and comment on these aspects of the structure.

2. There appears to be overzealous protonation of histidines; too many are doubly protonated. This is a hindrance for future investigators launching simulations from the deposited coordinates.

3. Regarding the YNCY motif, we feel “selection” may be a better word than “recognition”. It appears that the mt-ribosome has co-evolved with mitochondrial mRNAs to utilize this feature, but it is not clear in what sense there is a recognition. Related, a suggestion for Figure 2C would be to include aggregate statistics in addition to the individual frequencies of the YNCY motif in mitochondrial genes. Next to the average for nuclear messages, we suggest the authors show the average and standard deviation of mitochondrial genes and a suitable test for statistical significance. For the text, the authors may want to exclude ND6—the only coding gene from the light strand—and add a comment about ND6’s uniqueness. For the averages and variances of mitochondrial mRNAs versus nuclear mRNAs, please show all the data points, a violin plot, a box plot, or something similar that conveys the distribution.

4. Regarding uS9m, we agree that the channel opens for mRNA, but it isn’t clear what the authors mean when they write that uS9m blocks the channel and represents “a gating mechanism.” Is it accurate to write that uS9m exists in a “closed” conformation in the absence of mRNA, and that upon message threading through the gate “a conformational change” in uS9m allows passage?

5. Regarding GDP and uS29, the authors may want to edit the last sentence on Page 11 to address two separable ideas in different sentences. For example, one sentence that describes the results of the biochemical assays based on mutagenesis, followed by a second sentence describing their proposed model for GDP+mS29’s role in governing the motions of the two subunits. In an ideal world, the authors would provide evidence that the loss-of-function mutant ribosomes have lost GDP binding activity.

6. The Amunt’s group previously reported the role of NAD with fungal mitoribosomes (<https://www.nature.com/articles/s41467-020-18830-w>). Perhaps this warrants a citation since NAD is also seen in the new human structure.

7. For Fig5c, in addition to the cartoon, would the authors present the actual structures of the classical state and the hybrid states that are represented by the cartoon? In Extended Data Fig5d, the authors do show the contacts for the hybrid state, but not the classical state.

8. On Page 8, Line 188, Fig. 3f... There appears to be a typo. Fig3 does not have f. This should be b?

9. On Page 15, in the captions for Fig7, the letters for panels appear to be mislabeled.

10. Line 118: it appears that "Fig. 1a" should be "Fig. 1b"

signed by Tristan Croll, Adam Frost, Kazuki Saito

We thank the Reviewers for taking the time to carefully read the manuscript, inspect the model and share their experience. We addressed all the requests and followed constructive suggestions on how to improve the study, its readability and presentation. The major changes in response to Reviewer # 1 include rearranging the manuscript, providing an additional analysis and figure on the decoding center, adding a discussion section, and changing the title. In response to Reviewer # 2, we expanded on the structural analysis and added the requested information to the text and figures. In response to Reviewer # 3, we improved the model, engineered, and analyzed five additional cell lines expressing mS29 variants to complete the biochemical assay. In response to Reviewer # 4, we followed the suggestions regarding the model, figures, and the text. Overall, in the revised version we provide more attention to the topics of the decoding center, YNCY motif, CP-tRNA, as well as additional experimental details. All the figures have been improved as suggested as well. For those aspects where the data is not in agreement with suggestions, we provide detailed explanations in the response letter with figures illustrating the discussed points.

Below is the point-by-point response with corresponding line and figure numbers indicated.

Reviewer #1 (Remarks to the Author):

The authors use cryo-EM, Stable Isotope-Labeled RNA as an internal Standard (SILNAS), molecular simulation and functional validation to reveal the detailed interactions between the ligands of the ribosome (tRNA and mRNA) and the ribosome with unprecedented detail, suggesting key roles for modified nucleotides. The authors present high resolution models of the mitochondrial ribosome at high resolution (2.2. Å), based on cryo-EM reconstructions. At this resolution, they are able to identify modified nucleotides. The authors find that the modified nucleotides are positioned to help align the ligands (tRNA and mRNA) in the ribosome. Specifically, the high resolution map allowed of modeling of 13 rRNA modifications, 7 protein modifications, and 6 polyamines. The maps reveal the detailed interactions between the mRNA and proteins mS39, uS5m and mS35, suggestive of a gating mechanism upon mRNA delivery. The authors also provide structural and functional evidence showing that GDP binds to protein mS29 to effect conformational selection. Finally, their combined study of cryo-EM and molecular simulation shows that E-tRNA functions without rRNA interactions, in contract to bacterial and eukaryotic systems, with L9 acquiring a new functional role.

The study represents a strong effort and could be published, provided the authors make the following minor changes.

General comments:

While the data is excellent and strong, the manuscript needs improvement. I recommend making the main story line clearer throughout the manuscript, rather than jumping from topic to topic.

Thank you. Following the suggestion to make the main story line clearer, we restructured the manuscript as follows:

1) The sections on aminoacylated tRNA^{Val} and polyamines have been brought forward to pages 4-5 right after the description of the structure determination, as examples of high resolution features, rather than individual topics,

2) This is followed by the mRNA theme that has now been expanded, as suggested by other reviewers, and broken down into two sections ‘*Initial binding of mRNA involves Y-N-C-Y motif coevolved with mS39*’ and ‘*mL40 N-terminal tail stabilizes the A- and P-site tRNAs and decoding interactions*’.

3) In addition, a new section has been added on pages 12-13 ‘*mRNA alignment in the decoding center and rRNA modifications*’. Together, the story line follows the progression of mRNA along the mitoribosome over three sections of the revised manuscript.

4) This is followed by the tRNA/uL1 theme that has also been expanded and divided into two sections: structural description, entitled ‘*Structure of the mitoribosomal L1 stalk reveals its unique protein features*’; and MD simulations ‘*Recognition of tRNA by the L1 stalk and intersubunit communication*’. The manuscript concludes with *Discussion*.

There is little to no discussion, so this should be expanded to explain the implications, challenges, previous work and future directions in much more detail.

We agree and now added a comprehensive *Discussion* section on pages 20-22 with relation to previous work. It is now a major part of the revised manuscript. In addition to the previous parts, the discussion consists of five new parts:

1) lines 500-516: mRNA binding by mitoribosomal proteins, including Y-N-C-Y motif and ND6.

2) lines 517-529: Adaptation of the L1 stalk for tRNA binding.

3) lines 530-552: Medical implications of the identified cofactors.

4) lines 553-567: Aminoacylated and modified CP-tRNA-Val, and potential involvement of mtEF-Tu in the mitoribosomal assembly.

5) lines 575-584: Future directions.

Title: title could be improved by describing the main findings of the study

To better describe the findings of the study, we changed the title to ‘*Structure of mitoribosome reveals mechanism of mRNA binding, tRNA interactions with L1 stalk, roles of cofactors and rRNA modifications*’.

Decoding center: the authors should include more discussion comparing their decoding center to the canonical bacterial decoding center (i.e., A1492, A1493, G530 and S12).

Thank you for pointing out that this important topic deserves more attention. In the revised version, we discuss the decoding center in a dedicated section on lines 300-338, and corresponding Supplementary Fig. 10 has been added. The decoding center is conserved, and it’s now described on lines 300-302: ‘*In the decoding center, the universally conserved G902 is in anti conformation, while A1557 and A1558 are flipped out from h44 to interact with the A-site codon:anti-codon helix in the canonical manner*^{55,56} (Supplementary Fig. 10).’ The figure shows modified nucleotides with cryo-EM density compared to the *E. coli* ribosome PDB ID 7K00. It highlights differences that are observed along the mRNA codons: m⁴C1486 (methylated m⁴Cm1402 in *E. coli*); m⁵C1488 (unmodified C1404 in *E. coli*), U1563 (N3-methylated m³U1498 in *E. coli*).

Regarding the protein uS12m, we now discuss it on lines 262-271 and new Supplementary Fig. 8: ‘*... in the A-site the nucleotide at position 9 is stabilized by uS12m* (Supplementary Fig. 8). Here, a

conserved K72 side chain forms a salt bridge with the phosphate of nucleotide 9 (last nucleotide in the A-site codon). At the same time, cis-proline P73 (Supplementary Fig. 8) coordinates two potassium ions, one of which directly coordinates 2' O of nucleotide 9 (Supplementary Fig. 8, Supplementary Video 1) and stabilizes the decoding center. The densities identified as two potassium ions have been observed previously, but modeled as Mg²⁺ (PDB 4V51)⁵² or waters (PDB 7K00)^{48,52}. However, they have been experimentally identified as K⁺ by long wavelength X-ray diffraction analysis (PDB 6QNR)⁵³, which agrees with our assignment (Supplementary Fig. 8).'

The figure shows interactions of uS12m with codon in the A-site in comparison to:

- 1) *T. thermophilus* ribosome (PDB ID 4V51) at 2.8 Å resolution, where the densities are assigned as Mg²⁺ and P73 is in *trans*;
- 2) *E. coli* ribosome (PDB ID 7K00) at 2.0 Å resolution, where two water molecules have been modeled, and P73 is in *cis*.
- 3) *T. thermophilus* ribosome (PDB ID 6QNR) at 3.1 Å resolution, where the densities were assigned as K⁺ based on long wavelength X-ray diffraction.

Figures

Figs. 1-2: Caption needs much more detail, including descriptions of all colors. Everything that appears in the figures needs to be described in the figure caption.

We revised the legends as suggested and added more details.

'Fig. 1: Structure of the human mitoribosome. a, Overview of the 2.2 Å resolution model with mRNA and tRNAs. Proteins are shown as cartoons colored in blue for LSU and yellow for SSU. The rRNA is shown as grey surface. Newly identified features are shown as spheres. b, The newly identified cofactors and modifications are indicated against background of translucent rRNA and proteins with tRNAs and mRNA shown in cartoon. We observe 13 rRNA modifications (green), 7 protein modifications (pink), 3 iron-sulfur clusters (red-yellow), guanosine diphosphate (GDP), adenosine triphosphate (ATP), NAD, SPM, 4 SPDs and PUT.'

'Fig. 2: High resolution features. Modified rRNA residues and disulfide-linked or reduced cysteine residues of mS37 in the SSU (yellow background). Modified rRNA and protein residues, and 2Fe-2S cluster in the LSU (cyan background). Their corresponding densities in the 2.2 Å map are shown as mesh at indicated threshold (bottom left) and local resolution (bottom right) levels.'

Fig. 1 panel c: here, the density of the modified nucleotides in isolation is shown. It was difficult to find density depicting interactions between the modified nucleotides and other part of the complex in the manuscript. Density depicting the connection of key modified nucleotides to their interaction partners should be shown, either in Fig.1 or in Fig.4.

We added this information on page 14, in Fig. 7: *'Post-transcriptional modifications in functional centers of the mitoribosome'*. It shows the following panels of the modified nucleotides with density:

- 1) m⁴C1486 and m⁵C1488 and the interactions at the P-site mRNA codon.
- 2) m⁶₂A1583, m⁶₂A1584, m⁵1076 and the interactions with mRNA and P-tRNA.
- 3) Um3039 and Gm3040 and the interactions with A-tRNA.
- 4) Gm3040 and Ψ3067 and the interface with the A-tRNA.

5) m¹A2617 and the inter-subunit bridge and the A-loop.

6) Gm2815 base-pairing with P-tRNA.

Fig. 2, panel a: is the surface depicted based on the atomistic model or the cryo-EM density? This should be stated explicitly, so as to not mislead the reader. Ideally, the cryo-EM density should be colored with the electrostatic potential information (also in Fig. 3). The schematic is not described in the caption. Panel c, right hand side: what does the spacing of disks indicate? The relationship between d,e,f,g and h is loosely shown, but very difficult to understand. This should be explained explicitly in the figure caption. In addition, the appropriate individual residues should be labeled in the ‘zoomed out’ views to make clear relationships to the ‘zoomed-in’ views. These individual residues should be depicted in detail (e.g., bonds or sticks representations) in the ‘zoomed out’ views. The purple in panel (i) appears to be a surface rendering based on an atomistic model. If so, this should be depicted in a non-surface representation, so as not to confuse it with actual cryo-EM density.

We indicated all the required information in the revised Fig. 5 and corresponding legends. The residues are now shown in the zoomed-out views in stick representation and links between figure panels have been clarified. The representation of uS9m has been changed to cartoon so as not to confuse with density. The revised legends read:

Fig. 5a: *‘Electrostatic coulomb potential surface representation of the model shows a positively charged groove (blue) on mS39 that accommodates the mRNA’.*

Fig. 5c: *‘Positions of the Y-N-C-Y motif in the mt-mRNAs are shown as discs (right) to depict its distribution along the mt-mRNA sequences. The total number of residues for each mt-mRNA is indicated.’*

Fig. 5d: *‘Overview of uS5m showing a region of contact with mRNA. The polybasic stretch 111-124 along the mRNA is colored by electrostatic potential. Interacting residues between uS5m and mRNA are shown with the local density map.’*

Fig. 5h: *‘uS9m N-terminus (purple cartoon) adopts alternative conformations that result in mRNA channel blocked or open states regulating mRNA access to the SSU (gold surface).’*

Fig. 7: panel labels in the caption are mixed up. c and d are missing and a,b, e, f are mixed up. Panels h and i are referred to, but do not exist.

Fixed, thank you for noticing.

Reviewer #2 (Remarks to the Author):

Singh et al. describes high resolution cryo-EM structure of human mitoribosome, and provides detailed information about roles of r-proteins in mRNA recognition, GDP-bound unique protein, characteristic L1 stalk, and cofactors including polyamines, and FeS clusters. This reviewer is impressed with comprehensive analysis of mitoribosome structure. However, I have a number of comments and concerns that should be addressed to improve this manuscript. Since structural studies of mitoribosome have already been published in several papers, this reviewer does not figure out what are the new findings and progress provided in this study. Please explain this point

clearly in introduction. In particular, I would like to know what part of protein density was not modeled in previous studies.

Thank you. We agree that it's important to define new findings from previous works. The finding reported in this study are new, and as suggested we outline the advances in two paragraphs on lines 121-131:

'The improved resolution allowed building the most complete available model of the mitoribosome, including several key protein extensions involved in tRNA binding (Fig. 1a). The presence of mRNA on the mitoribosome further enabled modeling 32 of its residues, which we then used to trace mitochondria-specific elements involved in mRNA binding and put the rRNA modifications in a functional context (Fig. 1b).

To provide a description of the conformational changes associated with tRNA movement, we next focused efforts on modelling the L1 stalk. Thus, we merged particles containing tRNA in the E-site and through partial signal subtraction³⁹ and obtained a 2.9 Å resolution map of the region, enabling model building of the L1 stalk (Supplementary Fig. 1). We then used this model to perform molecular simulations of the complete ribosome with tRNAs.'

In addition, following reviewer's comment, we composed a section on lines 134-171 that outlines new non-protein features of the model resolved due to the improvement in the resolution.

In abstract, "together with an experimental identification of the mitoribosomal RNA (rRNA) modifications" This reviewer does not find any new information about rRNA modifications. All these modifications have already been reported. They just confirmed previous studies. Please revise it with appropriate description.

We agree and changed to *'We present the structure of human mitoribosome together with validated mitoribosomal RNA (rRNA) modifications, including aminoacylated CP-tRNA^{Val}.*

I would suggest the authors briefly mention some details how they succeeded in getting the structure with higher resolution than previous structure.

Thank you for pointing this out. we added the requested information on lines 114-121: *'Selected monosome particles were subjected to per-particle defocus, beam-tilt and per-particle astigmatism correction, followed by Bayesian polishing^{36,37}. The data was then sorted into 86 optics groups based on acquisition areas followed by beam-tilt, magnification anisotropy and higher order aberration correction³⁸. A final set of 509,691 particles yielded a 2.2 Å map. The local resolution was further improved by performing masked refinements around different regions of the monosome. In addition, we obtained a 2.6 Å resolution map in the presence of mRNA and three tRNAs in the classical state, and 3.0 Å resolution map with two tRNAs in the hybrid state (Supplementary Figs. 1 and 2).'*

In addition, we also provide more details in the Methods section on pages 35-36.

L153, Fig e: In comparison to u5Sm density, mRNA densities are very sparse and almost undetectable except for G10-U11. This means mRNA is very flexible in these regions. I agree that u5Sm may support the correct path of mRNA toward the A site, but I think the authors cannot declare the contact between mRNA and u5Sm except for K239 and R237.

We agree with this assessment and revised Fig 5d that now shows R237 and K239 with mRNA and corresponding densities.

L155, 166:R46 and R49 stabilize mRNA. It is necessary to put H-bonds between amino group and phosphates in Fig. 2g. Side chain density of Arg residue is visible?

We added the density to the Fig 5f and clarified in the legend '*R46 and R48 of mS35 potentially interact with the mRNA backbone, and the density suggests flexibility of the side-chains*'.

L199: "The function of mL40 is to balance the charge between the A- and P-tRNAs." This is intriguing, but it's not clear what it means. Please discuss it more clearly. How do authors consider role of this basic peptide chain in translation?

We agree with the reviewer that this part is an overstatement, and there is not enough data to propose a mechanism for translation. Therefore, we toned down the claim on lines 283-284: '*The presence of four lysine residues 52-55 (Fig. 6a, Supplementary Fig. 9b), might suggest that mL40 contributes to balancing the charge between the A- and P-tRNAs.*'

In addition, Fig. 6b shows a comparison to bacterial and cytosolic ribosomes where a similar feature has been reported by Bruno Klaholz lab for uS19.

L238-240: The authors should specifically point out the sites where GDP is bound. Regarding "newly identified GDP", please explain why it was not observed in the previous structures and was observed in this structure.

We revised the text on lines 430-432: '*Recently, another putative binding site has been proposed, formed with residues 173, 177, 208, 209, 210, 238, 242, 245, 291, 292, 295^{45, 46} (Fig. 10a,b, Supplementary Fig. 11d), and our map at the local resolution of 2.65 Å suggests the presence of GDP in that site*'.

L245-246: It is necessary to explain more clearly which process and what kind of structural change is specifically meant by "intersubunit dynamics" here.

We revised the text, currently on lines 439-445: '*We compared the interactions of mS29, which is the signature protein of the SSU head with the CP between our classical (2.6 Å resolution) and hybrid (3.0 Å resolution) structures. In the classical state, they are extensive and manifest through three contact sites. Site-1 and -2 are formed between GDP-associated region and the respective mL46 helices α 2 and α 3 extending from the CP. Contact site-3 is formed with mL48 β -turn protruding into the intersubunit space. Upon subunit ratcheting, in the hybrid state mS29 shifts resulting in disruption of contact site-3*'.

L249-251: Switching of mS29 β -hairpin 208-216 should be indicated by the models (in classical state and hybrid state) based on actual densities instead of schematic depictions in fig.5c.

We agree and now show it in the revised version in Fig. 10c.

L253-256: Although they analyzed two double mutants (R177A/K245A and R177A/K295A), the results of R177A/K295A mutant were ignored in the main text. They should be added. Please explain why steady-state levels of COX1 and COX2 were significantly different between R177A/K245A and R177A/K295A, while the results from metabolic labeling were comparable.

We added a discussion on the effects of the two mutant pairs on lines 451-461. To assess the differences between the two mutants more accurately, we have now included the results of four biological repetitions of the metabolic labeling assay. The rate of mitochondrial protein synthesis for R177A/K245A is visibly more attenuated than for R177A/K295A, and the difference is significantly different, thus explaining the differences in the steady-state levels of COX1 and COX2. For the protein synthesis assays, we have done short pulses of 15 minutes to void potential degradation of newly synthesized proteins, which could contribute to the previous apparent lack of significant differences (Fig. 10d-g).

L272-279: Distances and angle information written here can be put in the corresponding figures (Fig.6).

Added to Fig. 8c and 8d, as suggested.

L285-288: Authors need to explain what exactly the delta-uL1m model is. Why Q112 was chosen for truncation?

The details regarding the models have been added to the methods section with the exact values used for each model, under the heading, '*Molecular Dynamics Simulation*'. The baseline model is on pages 39-42, followed by the modified models on page 42.

To elaborate further, in the Δ uL1m model, we examined the effect of the attractive interactions between uL1m protein and the mt-tRNA on P/E formation. This was done through weakening those attractive interactions compared to the baseline model and analysing the changed dynamics and kinetics. P/E formation in the Δ uL1m model showed less stable uL1m-tRNA interface formation as expected. This led to larger conformational entropy of the mt-tRNA elbow during P/E formation due to accessing a larger number of conformational states. This led to increasing the first passage time between the I3 intermediate state and the P/E state by 2-fold compared to the baseline model. We reason that uL1m stabilizes the mt-tRNA elbow in near its P/E state while the CCA end finds its way to dock into the E-site. This stabilization reduces the conformational entropy of the elbow and controls the rate of the transition. This shows that it is possible to control the rate of P/E formation through mutating some of the basic uL1m residues in the uL1m-tRNA interface.

The C-terminal region of mL64 is involved in steric interactions with the mt-tRNA. In the Δ mL64 model, we investigated the effect of mL64 steric interactions with the mt-tRNA through truncating the C-terminal region of mL64 and observing the changed dynamics and kinetics compared to the baseline model. Many sites for truncation of mL64 are possible that would remove the C-terminal region. We wanted to keep the rest of mL64 protein intact so as to not affect any other processes other than its interaction with the mt-tRNA during P/E formation. Q112 was chosen as truncating here guarantees no steric interactions between mL64 and the mt-tRNA and keeps the rest of the protein intact. This can be seen from examining Fig. 9e where the site of truncation at Q112 is indicated.

L295-296: “The structure also shows that”, but the readers may not find any figure indicating this. They should prepare a figure showing that.

We prepared Supplementary Fig. 9e showing that E-tRNA does not directly interact with mRNA in the mitoribosome, which contrasts with the bacterial ribosome PDB ID 4Y4P.

This reviewer was surprised to know that CP-tRNA is fully modified and also aminoacylated with Val. What is a role of aminoacylation for 39S assembly? Once tRNA is aminoacylated, aa-tRNA is tightly recognized by EF-Tu. The author might have some comments on CP-tRNA in ribosome assembly after aminoacylation.

We thank the reviewer for raising this important point. We added a discussion on this aspect on lines 554-565 with a reference for 39S assembly and GTPBP5 study by Rorbach and Hallberg labs:

‘While the critical modifications are conserved with bacteria, an unexpected feature of the structure is that the rRNA component CP-tRNA^{Val} is modified, as well as aminoacylated. This suggests that this tRNA is recognized in the matrix by the elongation factor mtEF-Tu prior the assembly onto the CP. The involvement mtEF-Tu in the mitoribosome assembly has been previously reported, and direct interaction with the assembly factor GTPBP5 was visualized by cryo-EM of an LSU intermediate⁷⁹. Thus, the regulation of the assembly of the mammalian mitoribosome relies on the availability of mtEF-Tu in active developmental stages of the organelle, which might be a potential early checkpoint for translation. A possible model for these observations is that accumulation of deacylated tRNAs might trigger starvation responses, and since tRNA^{Val} is likely to be abundant at the onset of mitoribosomal assembly, its aminoacylation is essential, which requires the presence of mtEF-Tu.

All mito-rRNA modifications are visualized in this structure. This reviewer would like to know the structural effect and hypothetical role of mitochondria-specific rRNA modifications, especially for m 5C1488, m5U1076 in 12S rRNA and m 1A2617 in 16S rRNA.

Thank you for the suggestion to expand on the structural analysis. As requested, we added a description on structural effect of rRNA modifications in a new section on pages 12-13. In addition, we revised the corresponding Fig. 7 and added cryo-EM density map. Finally, in Supplementary Fig. 10, we show mRNA-rRNA interactions in the mitoribosome compared to *E. coli* ribosome (PDB ID 7K00). As described in the legend, differences are observed along the mRNA codons: m⁴C1486 (methylated m⁴Cm1402 in *E. coli*); m⁵C1488 (unmodified C1404 in *E. coli*), U1563 (N3-methylated m³U1498 in *E. coli*). Conserved nucleotides at the decoding center G902, A1557, A1558 interact directly with A-site tRNA anti-codon, equivalent to the corresponding residues G530, A1492 and A1493 in *E. coli*.

mRNA stabilization mechanism by uS12m, uS9m, uS7m are quite intriguing. Please compare roles of these r-Proteins in bacterial ribosome.

It’s a good idea. We added Supplementary Fig. 6 showing comparison of protein elements involved in mRNA binding with *E. coli* ribosome (PDBID 6ZTJ). We also added Supplementary Fig. 9e showing that E-tRNA does not directly interact with mRNA in the mitoribosome in contrast to the bacterial ribosome (PDB ID 4Y4P) where E-tRNA nucleotide at position 34 stacks with mRNA

nucleotide base. Finally, the new Supplementary Fig. 8 compares interactions of uS12m with codon in the A-site.

In Fig. 4a, the P-site codon is GUG, and anticodon is CAC? If this is tRNA^{Val}, its anticodon sequence is UAC. No mt-tRNA has CAC anticodon. Mitoribosomes contain mixture of endogenous mRNAs. Why mRNA sequence is so defined in this model?

We thank the reviewer for pointing out the mistake in the tRNA model. We replaced the tRNA residue at position 36 from C to U, so the anticodon is now CAU. The corresponding codon residue has been changed from G, to A so the codon sequence is AUG. This fit the density better than CAC.

Only the codon:anti-codon pairs (residues 4-9) and Y-N-C-Y motif (residues 25-28) were best resolved and identification is highest confidence. For the remaining mRNA, the chain could be traced accurately, and base orientation could be discerned but the identification was not unambiguous. Therefore, the sequence of mRNA is an approximation based on density, interactions and conservation.

We added this information to Methods on page 38, lines 1228-1234: *‘Along the mRNA channel, 34 residues of mRNA were built. In the tRNA binding region, codon-anticodon pairs were clearly resolved. Additionally, extra residues near the A-site (10-12, 14-16) and at the mRNA channel entrance (25, 27-29) were modeled accurately into density. For the remaining mRNA chain at a relatively lower resolution, correct overall chain trace and nucleotide orientation could be assigned. This allowed specific protein-nucleotide interactions to be identified for uS5m, uS7m, uS9m, uS12m, mS35, mS39 and mL40.’*

There are 3 FeS clusters in mitoribosome. But, only one of them is shown in Fig. 1c Where can we find two others? What is the role of these FeS clusters? As 2Fe-2S cluster is very unstable, is there any density assignment or experimental verification of this cluster?

The current manuscript doesn't expand on the theme of FeS clusters, because it has been described and discussed in two recently published studies that are cited in the revised version: Itoh Y, *et al. Structure of the mitoribosomal small subunit with streptomycin reveals Fe-S clusters and physiological molecules. Elife* 11, (2022), ref 45; Ast T, *et al. METTL17 is an Fe-S cluster checkpoint for mitochondrial translation. bioRxiv*, 2022.2011.2024.517765 (2022), ref 78.

“Polyamine and NAD functionally compensate” The authors do not mention NAD at all in the text.
Removed.

Additional points

Fig. 1: No description for Ala2, Ser2, SPD and PUT nomenclatures in this figure's legend. Also, the authors should add the explanation about the values of “level” and “res” in Fig.1c. It is strange Fig. 1c is not mentioned in the text.

We added the missing details.

Fig. 3c: Need numbering of anticodon loop.

Figure3C is now Supplementary Fig. 9d. Anticodon numbers have been added.

Fig. 3d: How do G164 and G165 stabilize mRNA?

We added Supplementary Fig. 9e showing G164 and G165 of uS7m stacking against mRNA nucleotide base.

Fig. 4h-j: Mention something from these figures.

Figure 4h-j is now Fig. 3b-d, and it's mentioned on lines: 145-156.

Fig 7: labelling and legend are completely disordered. For example, g-i is shown in legend, but Fig 7h and 7i does not exist. Please check labeling and legends.

Fig. 7 is now Fig. 9, and the legends and labels have been corrected.

Fig. 8b: You should show the surrounding residues and interactions of these polyamines (or Mg, K ions in lower panel) in Figure. Add PDB ID for E. coli ribosome.

Added in the Supplementary Fig. 5 showing comparison of polyamine densities with antibiotic-treated cells (PDB ID 6ZM5).

L117: 6 mitoribosome-specific modifications mean m⁵C1488, m⁵U1076 in 12S rRNA, m¹A 2617 in 16S rRNA, and 3 tRNA(Val) modifications? You should mention this in text or Fig1c.

We added the relevant information on lines 137-139: *'The mitochondria specific modifications include base methylations m⁵U1076, m⁵C1488 in the SSU; m¹A2617 in the LSU; and m¹A9, m²G10 and ψ39 in the tRNA^{Val} incorporated in the central protuberance (CP)'*.

We also added the analysis of the interactions on lines 305-338: *'Close to the decoding center, the residue G899 was recently shown to interact with streptomycin in human mtSSU⁴⁵. Its methylated to m⁷G527 in E.coli (Supplementary Fig. 10) and this modification partly contributes to streptomycin sensitivity⁵⁹. In the P-site, two base-methylated nucleotides m⁴C1486 and m⁵C1488 flank the base triple C1487:A1564:U1563 by stabilizing the stacking interactions (Fig. 5a,b). The orientation of N⁴-methylated m⁴C1486 towards mRNA is conserved with respect to E. coli, where the corresponding residue is also 2'-O methylated (m⁴Cm1402) (Supplementary Fig. 10). m⁴C1486 together with C1487 interacts with the phosphate group of the last nucleotide in the P-site mRNA codon via hydrogen bonds (Fig. 5a,b). Further, phosphate group of m⁴C1486 coordinates a magnesium ion, which stabilizes a conserved kink of the mRNA between A- and P-sites (Fig. 7a). In contrast to the E. coli P-site, U1563 does not have an N³-methylation (m³U1498 in E. coli) (Supplementary Fig. 10). This lack of N³-methylation allows U1563 together with C1487 to form a water mediated interaction with the P-site codon (Fig. 7a,b). Further, the conserved dimethylated residues, m⁶₂A1583 and m⁶₂A1584 orient the nucleotide U1563 such that the ribose of U1563 forms a hydrogen bond to a phosphate group of the second nucleotide in the P-site mRNA codon, while the phosphate of U1563 forms a hydrogen bond with the ribose on the first codon nucleotide (Fig. 7b). The two dimethylated nucleotides facilitate A1080 positioning for the 2'-OH to hydrogen bond with N3 of m⁵U1076. This together with methylation of m⁵U1076 contributes to the stability of h24 loop that interacts with the P-tRNA anticodon stem at position 38 via A1078 (Fig. 7a,b).*

In the A-loop of H92, there are two 2'-O-methyl modified nucleotides Um3039 and Gm3040. The 2'-O-methyl of Um3039 sterically positions the bases of Gm3040 and U3041 to form interactions with the nucleotides C75 and C74 of the A-tRNA, respectively (Fig. 7c,d). The 2'-O-methyl of Gm3040 present in the cytosolic but not bacterial ribosome and interacts with the base of A3069 to stabilize the conformation of Gm3040. Furthermore, the bacterial counterpart rRNA that interacts with A-tRNA (C75) is G:C, whereas in our structure it's A:U (3069:2994) that is less stable, and thus stabilized by the mitochondria-specific methylation of Gm3040 and base-pairing interactions with ψ 3067 (Fig. 7d). In the LSU, mitochondria-specific m^1 A2617 positions U2614 into a wedged interaction (Fig. 7e). Surprisingly, there is only weak sugar puckering for m^1 A2617 (Fig. 2a). It bears a partial positive charge at N1 and interacts with U2649 of H61. The interactions mediated by m^1 A2617 potentially contribute to the structuring of H71 loop that participates in an intersubunit bridge shares an interface with the A-loop (Fig. 7e). The universally conserved Gm2815 in the P-loop forms Watson-Crick base pairing with C75 of P/P-tRNA, similar to the interactions of Gm3040 of the A-loop (Fig. 7f).'

L152: You should change Fig.2 to Fig.2a.

Changed.

L185: Please show the position of 'kink-2' in Figure.

Label has been added to Fig. 5E.

L188: Fig. 3f does not exist. Please check the labelling of Figure.

Fig. 3 is now split into: Fig. 6a showing interactions of mL40 with tRNAs and mRNA, Supplementary Fig. 8 showing interactions of uS12m with mRNA, and Fig. 9d,e showing interactions of uS9m and uS7m. The labeling and text references for the figure have been corrected.

L194, Fig4a: In Fig 4a, G1485 is misspelled to C1485. And I think you should show the interaction between G1485 and mRNA in Fig 3a, because Fig 4a is focused on rRNA modifications around mRNA.

Fig. 4 is now Fig. 7 showing G1485 phosphate and its coordination with Mg^{2+} in panel (a).

L195: position 6 might be position 5?

Corrected, thank you for noticing.

L200, Fig3a: residues 51-55 are not shown in Figure. You should show these residues in Figure.

We added Supplementary Fig. 9 showing the N-terminal region of mL40 lodged between A- and P-tRNA. Panel (b) illustrates basic residues 51-55 and how they interact with tRNA phosphate backbone. Particularly, R51 forms salt bridges with A1560 and C1561 phosphates in h44. In panel (c), we mutated the mRNA residue at position 6 to depict potential H-bond of S47 side chain regardless of base identity.

L214-L218: when referring to bacterial ribosome, showing the bacterial rRNA number is preferable (for example, U1563 (m3U1498 in E.coli)).

We added *E. coli* numbers in brackets in the text.

L210-213: Unify the style of RNA modification. m4C1486 should be m4C1486. N4-methylated should be N4-methylated

Corrected.

L222 Fig4 typo m51076 to m5U1076

Corrected.

Fig5e: the upper gel image is not clear. Please adjust the contrast to show S/N ratio is sufficient for analysis.

Done.

L282-L282: You should show this point in Figure.

Position of Q112 is now indicated in Fig. 8e.

Ex Fig 1: Labeling the PDB number in Figure is preferable. (for example, label '7QI5' in E/E P/P A/A map)

We indicated PDB IDs for all published structures used for analysis in the corresponding figure panels, listed them in the Data availability statement with descriptions and links to the database, and indicated the PDBIDs from the current work in Supplementary Table 1.

Ex Fig5a: In figure, lane 4 is COX5B, but in legend, it is noted as uS3m. Please correct it.

Corrected, it's now Supplementary Fig. 11.

Reviewer #3 (Remarks to the Author):

The manuscript entitled 'The complete structure of human mitoribosome, roles of mito-specific protein elements, cofactors and rRNA modifications' by Singh et al. provides data on the structural details of human mitoribosome (rRNA and rProt modification, polyamines and ligand binding) explaining the mechanisms of mRNA/tRNA recognition. Despite the provided high-resolution structure (2.2 Å) most of the proposed mechanisms miss structural/biochemical evidence so I suggest that the manuscript as it stands is not suitable for publication in Nature communications and it should be further refined, including the models described therein. The key arguments why a publication cannot be accepted from this point on are as follows:

1) Despite the overall high quality of the map and the density present for most of the ligands proposed, there are plenty of errors in the model and ambiguous interpretation of the density. For example, SPD 1703 (chain AA) is not placed correctly in the density (figure 8b). Moreover, there are several examples where the same blob of density was treated as water and potassium ion (i.e. Water3857 (chain AA) and K3305 (chain A)). Due to the absence of additional evidence from other methods, we suggest that most potassium ions should be considered as magnesiums unless there is strict evidence provided. All the densities for the ligands and modifications should be revised and plotted at the same level, for example, Val101 in CP-tRNA is out of the density and the density of this region is very weak to consider it as Val or Glu (Figure 1). To sum up, all the models should be revised, hydrogens that are not visible in the density should be removed, as well as unmodelled blobs should be inspected and some parts of the model should be rebuilt, i.e. chain e, residues 246-251 which are not placed correctly in the map.

Thank you very much for taking the time to carefully inspect the model. We appreciate that the Reviewer has generously shared their experience with respect to the model building and we agree that interpretations of the subunit interfaces and the cavities should be done very carefully. Based on the comments we made the following revisions in the model:

- SPD1703 has been removed from the figure and not discussed. We corrected the model as shown in the figure below, right panel:

- Chain e, residues 246-251 do not show a clear density and potentially exist in more than one conformation. We revised all three models to better accommodate into the map with improved stereochemical geometry, as shown in the figure below. Particularly, residues 246-251 were remodeled to better fit the density. Then, residues 244-252 were subjected to Molprobit analysis showing improvement in quality of the model (see table below).

Previous (Chain e:244-252)

Poor rotamers	0	0.00%	Goal: <0.3%
Favored rotamers	6	100.00%	Goal: >98%
Ramachandran outliers	0	0.00%	Goal: <0.05%
Ramachandran favored	4	57.14%	Goal: >98%
Rama distribution Z-score	-5.59 ± 1.88		Goal: abs(Z score) < 2
Cβ deviations >0.25Å	0	0.00%	Goal: 0
Bad bonds:	0 / 66	0.00%	Goal: 0%
Bad angles:	0 / 86	0.00%	Goal: <0.1%
Cis Prolines:	0 / 0	0%	Expected: ≤1 per chain, or ≤5%
CaBLAM outliers	1	20.0%	Goal: <1.0%
CA Geometry outliers	0	0.00%	Goal: <0.5%

Current (Chain e:244-252)

Poor rotamers	0	0.00%	Goal: <0.3%
Favored rotamers	6	100.00%	Goal: >98%
Ramachandran outliers	0	0.00%	Goal: <0.05%
Ramachandran favored	6	85.71%	Goal: >98%
Rama distribution Z-score	-0.36 ± 3.03		Goal: abs(Z score) < 2
Cβ deviations >0.25Å	0	0.00%	Goal: 0
Bad bonds:	0 / 66	0.00%	Goal: 0%
Bad angles:	0 / 86	0.00%	Goal: <0.1%
Cis Prolines:	0 / 0	0%	Expected: ≤1 per chain, or ≤5%
CaBLAM outliers	0	0.0%	Goal: <1.0%
CA Geometry outliers	0	0.00%	Goal: <0.5%
Chiral volume outliers	0/7		

- Histidines were manually inspected and hydrogens were removed from ND1 or NE2 atoms, so that now there are only 22 doubly protonated histidines in the structure.
- Partial E-tRNA (residues 15-21; 27-43 and 54-58 not modeled) including the 3'-CCA and the acceptor arm has been built into the consensus model which agrees with the density.
- In the consensus model, where A- and P-tRNA are in partial occupancy, density for mL40 N-terminal loop is weaker, and residues 52-59 have been removed.

Let us please explain the rationale for the panels showing densities in Fig. 2, as well as our considerations for the mentioned aspects of the model building. For the figures, due to the difference in the local resolution, it is more informative for a reader to assess adjusted levels, based on background noise which varies from region to region. For transparency, we indicated in Fig. 2 threshold and local resolution in each one of 20 panels.

For metal ion modeling, we rationalized the identity based on the following parameters in our 2.2 Å resolution map: 1) amino acid coordination; 2) distances from residues; 3) strength of density in both unsharpened and local resolution maps. If we understand correctly, the Reviewer mentions 'same blob of density' referring to the size, however we think it would be confusing to use this parameter alone as a guideline for model building and thus potentially lead to erroneous conclusions. To exemplify our approach, the coordination distance of magnesium is ~2 Å with an octahedral hexa-coordinated geometry, whereas that of potassium is ~3 Å with more relaxed constraints on coordination geometry. For waters, the distance is ~2.8 Å between counterparts, ~2 Å from magnesium, ~3 Å from potassium. We added these details in the Methods section (page 38, lines 1210-1217).

Figure: Examples of density and coordination geometry exhibited by Mg²⁺ and K⁺.

We made the following improvements in the model:

- K3750 (chain n; PDB ID 7QI4) which was interpreted as a water molecule (HOH6018, chain C) in AAPPEE state (PDB ID 7QI5) has been deleted.
- K2024 was replaced with Mg1910 (chain Am); K2025 replaced with HOH3405 (chain Ah); K2029 replaced with HOH3408 (chain Ah).

- K2030 was deleted and Mg1911 was added.

Regarding CP-tRNA, Val was identified using deep sequencing in Brown et al 2014. The extra density in the current study is now shown in Fig. 3d, and we refined the modelling of V77 to better fit the density, as shown below.

2) The hypothesis introduced in the paragraph concerning the mRNA recognitions is not consistent with the structural data observed. In the presented data, there is a strong density only for residues 4-10 and 27-29 out of 32 nts of provided mRNA sequence. It means that for 22 nts there is no density/weak density and it is not enough to make the statements. For example, as is shown in figure 2b there is no density for the 26th residue, and weak density for the 25th residue, so any conclusion could be done at a low level of confidence. Also, there is a question mark on the position of the protein mL40. According to the density, we can conclude that there is an interaction between the phosphate group of the 6th nucleotide of mRNA and the mL40 protein, but it seems ambiguous how the authors put the protein to the density because a big part of the protein in the intersubunit space is disordered (51-55 residues). Thus, it is not clear, how many residues are located in this gap and whether 47 residue (46? 48?) interacts with the mRNA. Moreover, there is also an interaction with the 6th nucleotide base, so the assumption that « mL40 contact represents a prominent feature that is independent of mRNA sequence » is not correct. I suggest omitting this part because there is not enough structural evidence for most of the suggestions proposed..

The binding of mRNA by the mitoribosome, and a role of mL40 are unique observations in this system. Thus, the Reviewer touches upon an intriguing question discussed in our work, which therefore deserve attention. For mRNA, based on the reviewer's comment, we removed 'recognition' from the text and made the following changes to the model:

For mRNA residues 11-24, the density is relatively poor. However, we could identify the chain trace and orientation of some of the bases. This allowed us to build a continuous mRNA chain as described in detail in the methods (Lines 1228-1234): *'Along the mRNA channel, 34 residues of mRNA were built. In the tRNA binding region, codon-anticodon pairs were clearly resolved. Additionally, extra residues near the A-site (10-12, 14-16) and at the mRNA channel entrance (25, 27-29) were modeled accurately into density. For the remaining mRNA chain at a relatively lower resolution, correct overall chain trace and nucleotide orientation could be assigned. This allowed specific protein-nucleotide interactions to be identified for uS5m, uS7m, uS9m, uS12m, mS35, mS39 and mL40.'*

To further address the reviewer's concerns, we have removed description of specific interactions between uS5m and mRNA (residues 13-24) and retained only interactions of R237 and K239 (with clear side chain densities) at mRNA residue U11.

Regarding mS35, we rephrased on lines 231-232 to '*... mS35 provides potential stabilizing electrostatic interactions to mRNA with N-terminal residues K46 and K49*'. Interactions of uS9m and mS35 described are justified with density for relevant mRNA residues shown in figure 5f,g.

For mL40, we added Supplementary Fig. 9a showing the continuous density for residues 47-59. Since S47 is the N-terminal residue of mL40 (1-46 is transit sequence), we placed it at the end of the density. This is now shown in Fig. 6a closeup panel. Overall, the number of modeled residues 47-59 is consistent with the length of the density. However, we agree with the reviewer's comment about the quality of the maps on this region, and therefore removed residues 52-59 from the consensus model, and kept it only in the classical state that is better resolved.

To address the comment regarding the interaction with mRNA, we added Supplementary Fig. 9c showing S47 H-bonds with each one of four possible nucleobases in position 6. It suggests H-bonding between S47 side chain and base regardless of its identity. The corresponding text on lines 279-280 was changed to '*Moreover, the base orientation allows any nucleotide to interact with S47*'.

3) Concerning the GDP functions, the authors present structural data along with biochemical experiments. The role of GDP was elucidated by a comparison of the interaction between mS29 protein in presence of GDP or non-hydrolyzable analog GMPPNP. Authors are claiming that there are no structural changes occurred in mS39 protein, however from the density, there are different conformations for side chains of residues K245, H291, and K245 (figure 5b). Also, the authors suggest that there are « no changes in side chains engaged in intersubunit communication », however, there is no density for side chains to prove this hypothesis. Besides, the biochemical study does not seem fully completed. Given the absence of influence of one residue mutation, the authors performed just 2 double mutations claiming that « GDP acts as a stabilizing agent for mS29 β -hairpin 208-216 to maintain the intersubunit communication ». To make this conclusion author based on the biochemical data of R177A/K245A mutation. To prove this hypothesis authors should investigate other mutants, for example, Y208/N238 to elucidate the pairs which keep the stability of GDP. Also, there is an error in the Y173 position, the side chain should be rotated to form a stacking interaction with GDP.

Thank you for this suggestion. It's a good point, and we now completed the biochemical study. As requested, we engineered five additional cell lines expressing mS29 variants carrying mutations that would identify residues that provide the stability for GDP: Y173A/Y208A, R177A/H291A, V209P/D238A, Y208A/D238A, Y208A/K245A, (Fig. 10). The results described on lines 448-466 confirm the original conclusion, We also adjusted the orientation of Y173, and Fig. 10 has been updated.

Lines 451-469: '*Expression of the double mutants R177A/K245A or R177A/K295A, which target residues interacting with the phosphate groups of GDP, led to reduced levels of mitochondrion-encoded proteins and cellular respiration (Fig. 10d-g, Supplementary Fig. 11c), suggesting a stabilizing role for GDP in this pocket. The R177A/K245A mutation shows only a tendency toward a more stringent effect on mitochondrial protein synthesis than the R177A/K295A mutation in our*

short 15 min pulses. However, the effect is significant when looking at the steady-state levels of mtDNA-encoded proteins. (Fig. 10d,e). Interestingly, the double mutant R177A/H291A displayed a rate of mitochondrial protein synthesis comparable to WT (Fig. 10d-g). Although K245 and K295 stabilize the same beta phosphate on GDP, it is possible that 91 can compensate for the loss of K295, explaining the decreased stability of the R177A/K245A mutant compared to the R177A/K295A mutant. Change of nearby hydrophobic residues Y173A/Y208A yields a decrease in translation similar to the R177A/K245A implying their role in GDP stabilization (Fig. 10d-g). Lastly, double mutant pairs V209P/D238A and Y208A/D238A, which directly disrupt GDP base binding to the mS29 β -hairpin, showed drastic decreases in the synthesis and stability of mitochondrion-encoded proteins (Fig. 10d-g). Thus, the biochemical assays based on mutagenesis show that GDP binding to mS29 is required for efficient mitochondrial translation. Our structure-function correlation analyses suggest that GDP acts as a stabilizing agent for mS29 β -hairpin 208-216 to maintain the intersubunit communication as tRNA moves to the hybrid state.'

4) The mechanism of E-tRNA recognition which is performed by simulating approaches, is not supported by the structural data. To be confident that tRNA/L1 stalk motion follows the simulated pathway, intermediates should be captured proving the correlation between simulation and experiment. Also, there is an extra density for E-tRNA which is not built.

This comment is aligned with our understanding, and it was our intention to capture intermediates for the E-tRNA recognition. However, since the structural information was not informative to derive conclusions, we bridged the gap by employing all-atom simulations to inspect hundreds of possible transitions of the tRNA from the P site to P/E state. We appreciate the suggestion to capture more intermediates, and indeed tried cryoSPARC's latest flexible refinement (3DFlex) algorithm on particles corresponding to classes A/A P/P and A/P P/E, but with no success. The particles were subjected to signal subtraction around the tRNA and L1-stalk region. And 3DVA was attempted with masks of varying tightness around the tRNA and L1-stalk region. But, after repeated attempts a clean classification of data could not be achieved. We also included particles from potential intermediate states with less resolved or fuzzy tRNA densities in the P- and E-site.

The simulation data is described on lines 378-418 and presented in Fig. 9, Supplementary Video 2. A mechanistic insight from these calculations is that mL64 imposes strict limits on the pathways that are accessible to tRNA, and thus ensures that the tRNA is properly positioned and may reliably engage uL1m as the tRNA approaches the E site. This is consistent with the structural data, and to further address the Reviewer's concern, we now added Supplementary Fig. 9e showing that due to the different type of tRNA binding, no base pairs are formed between the E-tRNA and mRNA, unlike those found in bacteria. Instead, the protein uS7m undergoes a conformational change in which residues G164-165 directly contact the mRNA and stabilize its binding. This is also shown in Supplementary Video 1.

Since the density for E-tRNA is relatively weak owing to partial occupancy and mixture of states, we have not included the anticodon stem (residues 27-43) in the consensus model. The complete E-tRNA model can be found in our classical state PDB ID 7QI5.

5) The paragraph on polyamine's action seems ambiguous for two reasons. Firstly, the authors make a comparison with the structure in presence of Streptomycin. In this structure (PDB 7P2E) the large subunit is missing, making it impossible to check the presence of polyamines substitution.

Moreover, for SPD1703 in SSU, 4 molecules of water are missing (fig. 8b) which are present in the cryo-EM map of 7P2E). I assume that the whole density initially recognized as potassium and waters can be considered as a SPD molecule. Secondly, to date, all known studies show that polyamines are not interchangeable with any ions. Such a strong statement (335th row) should be at least double-checked. In this regard, the polyamine to ion substitution seems very ambiguous. Based on these facts the manuscript in this edition could not be considered for publication. Authors should revise the manuscript/models and perform to make it ready for publication.

We agree with the Reviewer and made the following improvements in the text and the figure:

We removed the section on polyamines and now only mention it in the discussion. The text has been clarified to explicitly state what the Reviewer suggests on lines 536-538 *‘Those changes are associated with attenuation of translation, and thus our structure rationalizes previous report that the positive effect of polyamines could not be functionally replaced by the addition of Mg²⁺ in a cell-free system ⁷⁸’*.

In the figure, we now make a comparison with the monosome PDBID 6ZM5 purified from antibiotic treated cells. We improved the figure by presenting the models along the cryo-EM maps, indicating interactions with dashed lines, and reporting a conformational change upon loss of PUT3304. The controversial SPD1703 is not discussed. The figure has been moved to Supplementary.

Reviewer #4 (Remarks to the Author):

The manuscript, “The complete structure of human mitoribosome, roles of mito-specific protein elements, cofactors and rRNA modifications”, by Singh, Itoh, et al. is an outstanding contribution to our understanding of mitochondrial translation. Using a ~2.2 Angstrom cryoEM density map, these authors report the exciting discovery of a YNCY mito-mRNA sequence motif that engages mito-specific proteins and rRNA moieties that facilitate decoding. They further report on the role of GDP binding within the mito-specific protein mS29 and the importance of this site for motion coupling between the two ribosomal subunits. In addition to GDP, the authors highlight the structural roles played by polyamines like spermine, Fe-S clusters, and the metabolite NAD. Finally, the authors synthesized several tRNA structural states using molecular dynamics to simulate how tRNAs translocate from the P-site to the E-site. The structures and the dynamic simulation highlight unique mitochondrial features, especially of the protein mL64 and the L1 stalk that, by contrast with the cytoplasmic ribosome, govern translocation from the P- to E-sites and ejection of the spent tRNA. After considering the following suggestions, we recommend the manuscript be published without delay.

1. Regarding the maps and their interpretation:

Cys76-Cys45 have been reported to be redox active, cycling between disulfide and reduced forms, and a disulfide looks plausible in the map. Nearby, Arg AF:240, Cys A2:45, Met A2:49, Met A2:53, and Ile AF:156 all appear to fit poorly. With more attention to model building in this region, Cys45 appears deprotonated and in a stable interaction with Arg240. Arg240 also H-bonds to the Met49 sulfur. Please cite the prior work on the redox chemistry in this vicinity and comment on these aspects of the structure.

We thank the reviewer for suggesting this interesting point for discussion. Indeed, a recent publication from Riemer lab (*J Cell Biol* 222: e202210019) describes the significance of redox active cysteine pairs in mitochondria-imported proteins in the context of integration of bioenergetics with the activity of the mitochondrial disulfide relay system. In this context the link to the mitoribosome is of interest, especially given that the mitoribosomal protein mS37 (CHCHD1) plays a key role in transition between assembly and initiation. Although due to the usage of DTT in the buffer our insight is somewhat limited, we added this point to the results, figures and discussion. Particularly, we made the following changes in the manuscript:

- In the results section, added text on line 171-173: ‘*We also identified a disulfide bond in one of the two redox active cysteine pairs in mS37 (CHCHD1) (Fig. 2) that is oxidized in the intermembrane space during mitochondrial import, representing a quality control mechanism*⁴⁷’.
- Added two panels in Fig. 2 showing the cysteine residues of mS37 in the SSU.
- Added text in Discussion, lines 549-552: ‘*The detection of disulfide bond in mS37 that was proposed to occur during the mitochondrial protein import pathway*⁴⁷ further links translation with metabolic state and NADH/NAD⁺ ratio in the intermembrane space and the matrix due to the special role of mS37 that links assembly to the initiation of translation²⁵’.

We have also revised this region in the model as suggested. Specifically, rotamer for R240 (uS7m, chain AF) has been changed to better fit the density.

M49 (mS37, chain A2) potentially exists in two rotameric states as suggested by the density. We have currently modeled the more favored rotamer. We found R240 was not within H-bonding distance from M49 sulfur in any of the likely rotameric states for both residues. We could not find a direct interaction of R240 with C45 (mS37, Chain A2) either. However, C45 could potentially interact with R240 via a water molecule which we modeled into a distinct density blob bridging C45 sulfur and R240. This analysis is exemplified in the figure below.

Figure: R240 side-chain (uS7m; chain AF) in two most likely rotameric states appears to be too far away to have direct interactions with sulfur atoms of C45 (in disulfide-linked and reduced states) and M49 (in its second most likely rotamer) from mS37 (chain A2) are shown.

2. There appears to be overzealous protonation of histidines; too many are doubly protonated. This is a hindrance for future investigators launching simulations from the deposited coordinates.

All histidines have been revised and ND1/NE2 have been deprotonated based on chemical environment. We kept 22 protonated histidines in the revised model.

3. Regarding the YNCY motif, we feel “selection” may be a better word than “recognition”. It appears that the mt-ribosome has co-evolved with mitochondrial mRNAs to utilize this feature, but it is not clear in what sense there is a recognition. Related, a suggestion for Figure 2C would be to include aggregate statistics in addition to the individual frequencies of the YNCY motif in mitochondrial genes. Next to the average for nuclear messages, we suggest the authors show the average and standard deviation of mitochondrial genes and a suitable test for statistical significance. For the text, the authors may want to exclude ND6—the only coding gene from the light strand—and add a comment about ND6’s uniqueness. For the averages and variances of mitochondrial mRNAs versus nuclear mRNAs, please show all the data points, a violin plot, a box plot, or something similar that conveys the distribution.

We agree with your interpretation and removed ‘recognition’. We also followed the suggestion to include aggregate statistics and calculated average frequency of YNCY motif in mitochondrial vs nuclear transcripts, which is now presented in Fig. 5c.

Regarding the ND6 that appears to be unusual, we further developed this theme in the *Discussion* section. Particularly, in Antarctic fish species, where the gene is translocated to the control region through gene duplication and with no obvious advantage on the protein level, we found that there is a change in the Y-N-C-Y frequency of 2-fold.

The added description is on lines 509-516: *‘Interestingly, ND6 possesses a long 3’UTR and is the only transcript in human mitochondria lacking a polyA tail. This gene is also the only one in mitochondria of vertebrates that was found to be translocated to the control region through gene duplication⁶⁹. This occurs in Antarctic fish species, and while the selectively advantageous duplicate is functionally neutral on the protein level⁷⁰, we found that there is a change in the Y-N-C-Y frequency in the mRNA sequence, which is increased by 2-fold (from 14 to 30 occurrences in 525 residues). It suggests a possible adaptive correlation between the mRNA stability on the mitoribosome and mt-DNA reorganization.’*

4. Regarding uS9m, we agree that the channel opens for mRNA, but it isn’t clear what the authors mean when they write that uS9m blocks the channel and represents “a gating mechanism.” Is it accurate to write that uS9m exists in a “closed” conformation in the absence of mRNA, and that upon message threading through the gate “a conformational change” in uS9m allows passage? We have removed the phrase, ‘gating mechanism’ in the context of uS9m. The revised text on lines 237-239: *‘Thus, uS9m exists in a closed conformation in the absence of mRNA, and upon mRNA threading, a conformational change in uS9m allows the passage.’*

5. Regarding GDP and uS29, the authors may want to edit the last sentence on Page 11 to address two separable ideas in different sentences. For example, one sentence that describes the results of the biochemical assays based on mutagenesis, followed by a second sentence describing their proposed model for GDP+mS29’s role in governing the motions of the two subunits. In an ideal world, the authors would provide evidence that the loss-of-function mutant ribosomes have lost GDP binding activity.

Thank you for this suggestion. We rephrased the sentence on lines 465-469 to *‘Thus, the biochemical assays based on mutagenesis show that GDP binding to mS29 is required for efficient mitochondrial translation. Our structure-function correlation analyses suggest that GDP acts as a stabilizing agent for mS29 β -hairpin 208-216 to maintain the intersubunit communication as tRNA moves to the hybrid state.’*

6. The Amunt’s group previously reported the role of NAD with fungal mitoribosomes (<https://www.nature.com/>). Perhaps this warrants a citation since NAD is also seen in the new human structure.

Thank you for noticing. We added this information on lines 169-170: *‘... and a molecule of nicotinamide adenosine dinucleotide (NAD) reported previously ...’*.

7. For Fig5c, in addition to the cartoon, would the authors present the actual structures of the classical state and the hybrid states that are represented by the cartoon? In Extended Data Fig5d, the authors do show the contacts for the hybrid state, but not the classical state.

In the revised manuscript, we present the actual structures of the classical and the hybrid states in Fig. 10c.

8. On Page 8, Line 188, Fig. 3f... There appears to be a typo. Fig3 does not have f. This should be b?

Corrected.

9. On Page 15, in the captions for Fig7, the letters for panels appear to be mislabeled.

Corrected.

10. Line 118: it appears that "Fig. 1a" should be "Fig. 1b"

Corrected.

REVIEWERS' COMMENTS

Reviewer #1 (Remarks to the Author):

The authors took the comments seriously and did a thorough re-working of the manuscript, making changes throughout to improve the writing style and impact of the paper. The study is to be published without delay.

An additional note: it would be great to include in the manuscript or in the supplementary information section some of the density figures (with black backgrounds) made for the rebuttal for reviewer 3 (especially the nice figures with magnesium ions).

Reviewer #2 (Remarks to the Author):

This revised version faithfully addressed most of this reviewer's concerns and requests in full. I have no further comment.

Reviewer #3 (Remarks to the Author):

Singh et al.'s manuscript titled "Structure of mitoribosome reveals mechanism of mRNA binding, tRNA interactions with L1 stalk, roles of cofactors and rRNA modifications" presents valuable data on the structural characteristics of the human mitoribosome. The study covers topics such as rRNA and ribosomal protein modifications, polyamines, and ligand binding, aiming to explain the mechanisms behind mRNA and tRNA recognition. Despite the high-resolution structure provided (2.2 Å), several proposed mechanisms lack sufficient structural evidence. Although the authors have made some improvements to the paper, the work appears to be primarily incremental compared to existing knowledge and cannot be accepted for publication. The primary reasons for my inability to accept this publication are outlined below:

1. The primary finding presented by the authors concerns the mRNA pathway in the ribosome and the specific interaction between mRNA and ribosomal proteins. In my previous review, I expressed my concerns regarding the absence of strong density for a significant portion of the mRNA in the cryo-EM map. Unfortunately, the authors did not address these concerns adequately in their response, failing to provide a clear explanation for their interpretations. As previously mentioned, the cryo-EM map only shows a strong density for residues 4-10 and 27-29 of the provided mRNA sequence, leaving 22

nucleotides without clear density or with weak density. Consequently, it is not possible to draw conclusive statements with a high level of confidence. Figure 2b shows no density for the 26th residue and weak density for the 25th residue, undermining any firm conclusions. Additionally, there are uncertainties surrounding the position of protein mL40. While the density indicates an interaction between the phosphate group of the 6th nucleotide of mRNA and the mL40 protein, the authors' placement of the protein within the density is ambiguous due to disorder in a significant part of the protein in the intersubunit space (residues 51-55). Consequently, it remains unclear how many residues are present in this gap and whether residue 47 (46 or 48) interacts with the mRNA. In summary, the conclusions drawn by the authors in the paragraphs "Initial binding of mRNA involves Y-N-C-Y motif coevolved with mS39" and "mL40 N-terminal tail stabilizes the A- and P-site tRNAs and decoding interactions" lack support from the cryo-EM map.

2. Despite the authors' claim of having checked the assignment of ions in the map, there are still numerous instances where the interpretation of distances and coordination remains ambiguous. For example, K1778 and K1779 in chain AA have uncertain interpretations and can be easily assigned as Mg. Furthermore, some potential potassium ions are treated as water molecules, such as water molecule 454 in chain M, while others, like potassium 201 in chain 3, are treated differently. The authors have not adequately addressed the comments on the ion assignment, resulting in potentially incorrect distributions of Mg/K/water.

3. The authors' main idea that the mRNA involves a Y-N-C-Y motif lacks experimental support from the cryo-EM map. According to the density and possible interactions, the sequence should be C-N-C-Y-Y or C-N-N-C-Y-Y. It is crucial for the authors to thoroughly explore the possibilities before drawing ambiguous conclusions regarding mRNA binding motifs.

4. The mechanism of E-tRNA recognition, which relies on simulation approaches, lacks support from structural data. In the absence of such data, this part of the study should either be omitted or described in a more general manner.

5. The role of GTP hydrolysis in protein mS29 remains unclear since it does not appear to affect the structure of the domain. It would be beneficial to provide biochemical insights into the role of GTP hydrolysis.

Reviewer #4 (Remarks to the Author):

The authors have surpassed my expectations. I recommend publication.

Reviewer #1 (Remarks to the Author):

The authors took the comments seriously and did a thorough re-working of the manuscript, making changes throughout to improve the writing style and impact of the paper. The study is published without delay.

An additional note: it would be great to include in the manuscript or in the supplementary information section some of the density figures (with black backgrounds) made for the rebuttal for reviewer 3 (especially the nice figures with magnesium ions).

Thank you. As suggested we included such a figure. It's Supplementary Fig. 9: Examples of densities and coordination of ions K^+ and Mg^{2+}

Reviewer #2 (Remarks to the Author):

This revised version faithfully addressed most of this reviewer's concerns and requests in full. I have no further comment.

Reviewer #3 (Remarks to the Author):

Singh et al.'s manuscript titled "Structure of mitoribosome reveals mechanism of mRNA binding, tRNA interactions with L1 stalk, roles of cofactors and rRNA modifications" presents valuable data on the structural characteristics of the human mitoribosome. The study covers topics such as rRNA and ribosomal protein modifications, polyamines, and ligand binding, aiming to explain the mechanisms behind mRNA and tRNA recognition. Despite the high-resolution structure provided (2.2 Å), several proposed mechanisms lack sufficient structural evidence. Although the authors have made some improvements to the paper, the work appears to be primarily incremental compared to existing knowledge and cannot be accepted for publication. The primary reasons for my inability to accept this publication are outlined below:

1. The primary finding presented by the authors concerns the mRNA pathway in the ribosome and the specific interaction between mRNA and ribosomal proteins. In my previous review, I expressed my concerns regarding the absence of strong density for a significant portion of the mRNA in the cryo-EM map. Unfortunately, the authors did not address these concerns adequately in their response, failing to provide a clear explanation for their interpretations. As previously mentioned, the cryo-EM map only shows a strong density for residues 4-10 and 27-29 of the provided mRNA sequence, leaving 22 nucleotides without clear density or with weak density. Consequently, it is not possible to draw conclusive statements with a high level of confidence. Figure 2b shows no density for the 26th residue and weak density for the 25th residue, undermining any firm conclusions. Additionally, there are uncertainties surrounding the position of protein mL40. While the density indicates an interaction between the phosphate group of the 6th nucleotide of mRNA and the mL40 protein, the authors' placement of the protein within the density is ambiguous due to disorder in a significant part of the protein in the intersubunit space (residues 51-55). Consequently, it remains unclear how many residues are present in this gap and whether residue 47 (46 or 48) interacts with the mRNA. In summary, the conclusions drawn by the authors in the paragraphs "Initial binding of mRNA involves Y-N-C-Y motif coevolved with mS39" and "mL40 N-terminal tail stabilizes the A- and P-site tRNAs and decoding interactions" lack support from the cryo-EM map.

All the maps and models have been shared with the community. The manuscript presents the most plausible interpretation of the data, and the text has been carefully considered to provide clear and comprehensive explanations and avoid subjective descriptions.

2. Despite the authors' claim of having checked the assignment of ions in the map, there are still numerous instances where the interpretation of distances and coordination remains ambiguous. For example, K1778 and K1779 in chain AA have uncertain interpretations and can be easily assigned as Mg. Furthermore, some potential potassium ions are treated as water molecules, such as water molecule 454 in chain M, while others, like potassium 201 in chain 3, are treated differently. The authors have not adequately addressed the comments on the ion assignment, resulting in potentially incorrect distributions of Mg/K/water.

We added on line 268 that our assignment was based on distances from coordinating residues, and as suggested by Reviewer # 1 included Supplementary Fig. 9: Examples of densities and coordination of ions K^+ and Mg^{2+}

3. The authors' main idea that the mRNA involves a Y-N-C-Y motif lacks experimental support from the cryo-EM map. According to the density and possible interactions, the sequence should be C-N-C-Y-Y or C-N-N-C-Y-Y. It is crucial for the authors to thoroughly explore the possibilities before drawing ambiguous conclusions regarding mRNA binding motifs.

Our proposal of the existence of Y-N-C-Y (positions 25-28) motif is an approximation based on density of the individual bases, interactions and sequence conservation in mitochondrial transcripts.

First position (residue 25, Y-N-C-Y) is proposed as a pyrimidine (Y). Since the density and interactions observed can be explained by modeling both pyrimidines at this position, ie, U or C, it's clear that Y is a more appropriate placement, as illustrated below:

First position of Y-N-C-Y can be modeled as C (left) or U (right) given the density and the nature of interactions. Hydrogen bonds are indicated by arrows.

Second position (residue 26, Y-N-C-Y) potentially represents a non-specific base as the density is weak and not supported by base-specific interactions leading to promiscuity in base identity an orientation. The reviewer suggests an insertion here as in C-N-N-Y-Y. Despite the lack of clear density, there is not enough space between the well ordered residues 25 and 28 (ie, Y-N-C-Y) to

accommodate more than one nucleotide between them. An attempt to model an additional nucleotide results in poor accommodation in the available density and non-optimum backbone geometry as shown below.

The second position (N26) could not be identified due to poor density and lack of base specific interactions (upper panel). Modeling of two residues is possible, but leads to bad RNA backbone geometry and pushes phosphates out of the density.

In the last position, the suggestions C-N-C-Y_Y or C-N-N-C-Y_Y implies that the motif can extend to a fifth position including an additional Y at 3' terminal. But the corresponding 29th residue while being supported by strong density is not supported by any surrounding interactions, which is exemplified here:

The Last residue of Y-N-C-Y at position 28 can participate in base-specific interactions and shows a strong density. The following 3' residue N29 does have a strong density but evidence to ascertain identity is not available from surrounding interactions. Arrows show H-bonding interactions.

4. The mechanism of E-tRNA recognition, which relies on simulation approaches, lacks support from structural data. In the absence of such data, this part of the study should either be omitted or described in a more general manner.

This comment makes no sense. Since there was no structural data, we performed simulations to fill the gap and complete the investigation of the biological question.

5. The role of GTP hydrolysis in protein mS29 remains unclear since it does not appear to affect the structure of the domain. It would be beneficial to provide biochemical insights into the role of GTP hydrolysis.

There seems to be a misunderstanding of this section in the study as well. The conclusion of the analysis is that mS29-GDP does not change its conformation but is a stable structural element that contributes to inter-subunit communication during transitions between classical and hybrid states. This is stated in the Discussion on lines 568-570.

Reviewer #4 (Remarks to the Author):

The authors have surpassed my expectations. I recommend publication.

Thank you.